# PERMUTATION-BASED SGD: IS RANDOM OPTIMAL?

**Shashank Rajput**[*]           **Kangwook Lee**           **Dimitris Papailiopoulos**

University of Wisconsin-Madison

## ABSTRACT

A recent line of ground-breaking results for permutation-based SGD has corroborated a widely observed phenomenon: random permutations offer faster convergence than with-replacement sampling. However, is random optimal? We show that this depends heavily on what functions we are optimizing, and the convergence gap between optimal and random permutations can vary from exponential to nonexistent. We first show that for 1-dimensional strongly convex functions, with smooth second derivatives, there exist permutations that offer exponentially faster convergence compared to random. However, for general strongly convex functions, random permutations are optimal. Finally, we show that for quadratic, strongly-convex functions, there are easy-to-construct permutations that lead to accelerated convergence compared to random. Our results suggest that a general convergence characterization of optimal permutations cannot capture the nuances of individual function classes, and can mistakenly indicate that one cannot do much better than random.

## 1 INTRODUCTION

Finite sum optimization seeks to solve the following:

$$\min_{\boldsymbol{x}} F(\boldsymbol{x}) := \frac{1}{n} \sum_{i=1}^{n} f_i(\boldsymbol{x}). \tag{1}$$

Stochastic gradient descent (SGD) approximately solves finite sum problems, by iteratively updating the optimization variables according to the following rule:

$$\boldsymbol{x}_{t+1} := \boldsymbol{x}_t - \alpha \nabla f_{\sigma_t}(\boldsymbol{x}_t), \tag{2}$$

where $\alpha$ is the step size and $\sigma_t \in [n] = \{1, 2, \dots, n\}$ is the index of the function sampled at iteration $t$. There exist various ways of sampling $\sigma_t$, with the most common being with- and without-replacement sampling. In the former, $\sigma_t$ is uniformly chosen at random from $[n]$, and for the latter, $\sigma_t$ represents the $t$-th element of a random permutation of $[n]$. We henceforth refer to these two SGD variants as vanilla and permutation-based, respectively.

Although permutation-based SGD has been widely observed to perform better in practice (Bottou, 2009; Recht & Ré, 2012; 2013), the vanilla version has attracted the vast majority of theoretical analysis. This is because of the fact that at each iteration, in expectation the update is a scaled version of the true gradient, allowing for simple performance analyses of the algorithm, *e.g.*, see (Bubeck et al., 2015).

Permutation-based SGD has resisted a tight analysis for a long time. However, a recent line of breakthrough results provides the first tight convergence guarantees for several classes of convex functions $F$ (Nagaraj et al., 2019; Safran & Shamir, 2019; Rajput et al., 2020; Mishchenko et al., 2020; Ahn et al., 2020; Nguyen et al., 2020). These recent studies mainly focus on two variants of permutation-based SGD where (1) a new random permutation is sampled at each epoch (also known as RANDOM RESHUFFLE) (Nagaraj et al., 2019; Safran & Shamir, 2019; Rajput et al., 2020), and (2) a random permutation is sampled once and is reused throughout all SGD epochs (SINGLE SHUFFLE) (Safran & Shamir, 2019; Mishchenko et al., 2020; Ahn et al., 2020).

---

[*]Correspondence to Shashank Rajput ⟨rajput3@wisc.edu⟩

Perhaps interestingly, RANDOM RESHUFFLE and SINGLE SHUFFLE exhibit different convergence rates and a performance gap that varies across different function classes. In particular, when run for $K$ epochs, the convergence rate for strongly convex functions is $\widetilde{O}(1/nK^2)$ for both RANDOM RESHUFFLE and SINGLE SHUFFLE (Nagaraj et al., 2019; Ahn et al., 2020; Mishchenko et al., 2020). However, when run specifically on strongly convex quadratics, RANDOM RESHUFFLE experiences an acceleration of rates, whereas SINGLE SHUFFLE does not (Safran & Shamir, 2019; Rajput et al., 2020; Ahn et al., 2020; Mishchenko et al., 2020). All the above rates have been coupled by matching lower bounds, at least up to constants and sometimes log factors (Safran & Shamir, 2019; Rajput et al., 2020).

From the above we observe that reshuffling at the beginning of every epoch may not always help. But then there are cases where RANDOM RESHUFFLE is faster than SINGLE SHUFFLE, implying that certain ways of generating permutations are more suited for certain subfamilies of functions.

The goal of our paper is to take a first step into exploring the relationship between convergence rates and the particular choice of permutations. We are particularly interested in understanding if random permutations are as good as optimal, or if SGD can experience faster rates with carefully crafted permutations. As we see in the following, the answer the above is not straightforward, and depends heavily on the function class at hand.

---

**Algorithm 1** Permutation-based SGD variants

**Input:** Initialization $x_0^1$, step size $\alpha$, epochs $K$
1: $\sigma = $ a random permutation of $[n]$
2: **for** $k = 1, \ldots, K$ **do**
3:     **if** IGD **then**
4:         $\sigma^k = (1, 2, \ldots, n)$
5:     **else if** SINGLE SHUFFLE **then**
6:         $\sigma^k = \sigma$
7:     **else if** RANDOM RESHUFFLE **then**
8:         $\sigma^k = $ a random permutation of $[n]$
9:     **end if**

10:     **if** FLIPFLOP **and** $k$ is even **then**
11:         $\sigma^k = $ reverse of $\sigma^{k-1}$
12:     **end if**

13:     **for** $i = 1, \ldots, n$ **do** $\left.\rule{0cm}{1.2cm}\right\}$ Epoch $k$
14:         $x_i^k := x_{i-1}^k - \alpha \nabla f_{\sigma_i^k}(x_{i-1}^k)$
15:     **end for**
16:     $x_0^{k+1} := x_n^k$
17: **end for**

---

**Our Contributions:** We define as *permutation-based SGD* to be any variant of the iterates in (2), where a permutation of the $n$ functions, at the start of each epoch, can be generated deterministically, randomly, or with a combination of the two. For example, SINGLE SHUFFLE, RANDOM RESHUFFLE, and incremental gradient descent (IGD), are all permutation-based SGD variants (see Algorithm 1).

We first want to understand—even in the absence of computational constraints in picking the optimal permutations—what is the fastest rate one can get for permutation-based SGD? In other words, are there permutations that are better than random in the eyes of SGD?

Perhaps surprisingly, we show that there exist permutations that may offer up to exponentially faster conver-

| | Plain | with FLIPFLOP | |
|---|---|---|---|
| RR | $\Omega\left(\dfrac{1}{n^2 K^2} + \dfrac{1}{nK^3}\right)$ | $\widetilde{O}\left(\dfrac{1}{n^2 K^2} + \dfrac{1}{nK^5}\right)$ | Thm. 5 |
| SS | $\Omega\left(\dfrac{1}{nK^2}\right)$ | $\widetilde{O}\left(\dfrac{1}{n^2 K^2} + \dfrac{1}{nK^4}\right)$ | Thm. 4 |
| IGD | $\Omega\left(\dfrac{1}{K^2}\right)$ | $\widetilde{O}\left(\dfrac{1}{n^2 K^2} + \dfrac{1}{K^3}\right)$ | Thm. 6 |

Table 1: Convergence rates of RANDOM RESHUFFLE (RR), SINGLE SHUFFLE (SS) and INCREMENTAL GRADIENT DESCENT (IGD) on strongly convex quadratics: Plain vs. FLIPFLOP. Lower bounds for the "plain" versions are taken from (Safran & Shamir, 2019). When $n \gg K$, that is when the training set is much larger than the number of epochs, which arguably is the case in practice, the convergence rates of RANDOM RESHUFFLE, SINGLE SHUFFLE, and INCREMENTAL GRADIENT DESCENT are $\Omega(\frac{1}{nK^3})$, $\Omega(\frac{1}{nK^2})$, and $\Omega(\frac{1}{K^2})$ respectively. On the other hand, by combining these methods with FLIPFLOP the convergence rates become faster, *i.e.*, $\widetilde{O}(\frac{1}{nK^5})$, $\widetilde{O}(\frac{1}{nK^4})$, and $\widetilde{O}(\frac{1}{K^3})$, respectively.

gence than random permutations, but for a limited set of functions. Specifically, we show this for 1-dimensional functions (Theorem 1). However, such exponential improvement is no longer possible

in higher dimensions (Theorem 2), or for general strongly convex objectives (Theorem 3), where random is optimal. The above highlight that an analysis of how permutations affect convergence rates needs to be nuanced enough to account for the structure of functions at hand. Otherwise, in lieu of further assumptions, random permutations may just appear to be as good as optimal.

In this work, we further identify a subfamily of convex functions, where there exist easy-to-generate permutations that lead accelerated convergence. We specifically introduce a new technique, FLIPFLOP, which can be used in conjunction with existing permutation-based methods, *e.g.*, RANDOM RESHUFFLE, SINGLE SHUFFLE, or INCREMENTAL GRADIENT DESCENT, to provably improve their convergence on quadratic functions (Theorems 4, 5, and 6). The way that FLIPFLOP works is rather simple: every even epoch uses the *flipped* (or reversed) version of the previous epoch's permutation. The intuition behind why FLIPFLOP leads to faster convergence is as follows. Towards the end of an epoch, the contribution of earlier gradients gets attenuated. To counter this, we flip the permutation for the next epoch so that every function's contribution is diluted (approximately) equally over the course of two consecutive epochs. FLIPFLOP demonstrates that finding better permutations for specific classes of functions might be computationally easy. We summarize FLIPFLOP's convergence rates in Table 1 and report the results of numerical verification in Section 6.2.

Note that in this work, we focus on the dependence of the error on the number of iterations, and in particular, the number of epochs. However, we acknowledge that its dependence on other parameters like the condition number is also very important. We leave such analysis for future work.

**Notation:** We use lowercase for scalars ($a$), lower boldface for vectors ($\boldsymbol{a}$), and upper boldface for matrices ($\boldsymbol{A}$).

## 2 RELATED WORK

Gürbüzbalaban et al. (2019a;b) provided the first theoretical results establishing that RANDOM RESHUFFLE and INCREMENTAL GRADIENT DESCENT (and hence SINGLE SHUFFLE) were indeed faster than vanilla SGD, as they offered an asymptotic rate of $O\left(1/K^2\right)$ for strongly convex functions, which beats the convergence rate of $O\left(1/nK\right)$ for vanilla SGD when $K = \Omega(n)$. Shamir (2016) used techniques from online learning and transductive learning theory to prove an optimal convergence rate of $\widetilde{O}(1/n)$ for the first epoch of RANDOM RESHUFFLE (and hence SINGLE SHUFFLE). Later, Haochen & Sra (2019) also established a non-asymptotic convergence rate of $\widetilde{O}\left(\frac{1}{n^2 K^2} + \frac{1}{K^3}\right)$, when the objective function is quadratic, or has smooth Hessian.

Nagaraj et al. (2019) used a very interesting iterate coupling based approach to give a new upper bound on the error rate of RANDOM RESHUFFLE, thus proving for the first time that for general strongly convex smooth functions, it converges faster than vanilla SGD in all regimes of $n$ and $K$. This was followed by (Safran & Shamir, 2019), where the authors were able to establish the first lower bounds, in terms of both $n$ and $K$, for RANDOM RESHUFFLE. However, there was a gap in these upper and lower bounds. The gap in the convergence rates was closed by Rajput et al. (2020), who showed that the upper bound given by Nagaraj et al. (2019) and the lower bound given by Safran & Shamir (2019) were both tight up to logarithmic terms.

For SINGLE SHUFFLE, Mishchenko et al. (2020) and Ahn et al. (2020) showed an upper bound of $\widetilde{O}\left(\frac{1}{nK^2}\right)$, which matched the lower bound given earlier by (Safran & Shamir, 2019), up to logarithmic terms. Ahn et al. (2020) and Mishchenko et al. (2020) also proved tight upper bounds for RANDOM RESHUFFLE, with a simpler analysis and using more relaxed assumptions than (Nagaraj et al., 2019) and (Rajput et al., 2020). In particular, the results by Ahn et al. (2020) work under the PŁ condition and do not require individual component convexity.

INCREMENTAL GRADIENT DESCENT on strongly convex functions has also been studied well in literature (Nedić & Bertsekas, 2001; Bertsekas, 2011; Gürbüzbalaban et al., 2019a). More recently, Nguyen et al. (2020) provide a unified analysis for all permutation-based algorithms. The dependence of their convergence rates on the number of epochs $K$ is also optimal for INCREMENTAL GRADIENT DESCENT, SINGLE SHUFFLE and RANDOM RESHUFFLE.

There has also been some recent work on the analysis of RANDOM RESHUFFLE on non-strongly convex functions and non-convex functions. Specifically, Nguyen et al. (2020) and Mishchenko

et al. (2020) show that even there, RANDOM RESHUFFLE outperforms SGD under certain conditions. Mishchenko et al. (2020) show that RANDOM RESHUFFLE and SINGLE SHUFFLE beat vanilla SGD on non-strongly convex functions after $\Omega(n)$ epochs, and that RANDOM RESHUFFLE is faster than vanilla SGD on non-convex objectives if the desired error is $O(1/\sqrt{n})$.

Speeding up convergence by combining without replacement sampling with other techniques like variance reduction (Shamir, 2016; Ying et al., 2020) and momentum (Tran et al., 2020) has also received some attention. In this work, we solely focus on the power of "good permutations" to achieve fast convergence.

## 3 PRELIMINARIES

We will use combinations of the following assumptions:

**Assumption 1** (Component convexity). $f_i(\boldsymbol{x})$'s are convex.

**Assumption 2** (Component smoothness). $f_i(\boldsymbol{x})$'s are $L$-smooth, i.e.,
$$\forall \boldsymbol{x}, \boldsymbol{y} : \|\nabla f_i(\boldsymbol{x}) - \nabla f_i(\boldsymbol{y})\| \leq L\|\boldsymbol{x} - \boldsymbol{y}\|.$$

Note that Assumption 2 immediately implies that $F$ also has $L$-Lipschitz gradients:
$$\forall \boldsymbol{x}, \boldsymbol{y} : \|\nabla F(\boldsymbol{x}) - \nabla F(\boldsymbol{y})\| \leq L\|\boldsymbol{x} - \boldsymbol{y}\|.$$

**Assumption 3** (Objective strong convexity). $F$ is $\mu$-strongly convex, i.e.,
$$\forall \boldsymbol{x}, \boldsymbol{y} : F(\boldsymbol{y}) \geq F(\boldsymbol{x}) + \langle \nabla F(\boldsymbol{x}), \boldsymbol{y} - \boldsymbol{x} \rangle + \frac{1}{2}\mu\|\boldsymbol{y} - \boldsymbol{x}\|^2.$$

Note that Assumption 3 implies
$$\forall \boldsymbol{x}, \boldsymbol{y} : \langle \nabla F(\boldsymbol{x}) - \nabla F(\boldsymbol{y}), \boldsymbol{x} - \boldsymbol{y} \rangle \geq \mu\|\boldsymbol{y} - \boldsymbol{x}\|^2. \tag{3}$$

We denote the condition number by $\kappa$, which is defined as $\kappa = \frac{L}{\mu}$. It can be seen easily that $\kappa \geq 1$ always. Let $\boldsymbol{x}^*$ denote the minimizer of Eq. (1), that is, $\boldsymbol{x}^* = \arg\min_{\boldsymbol{x}} F(\boldsymbol{x})$.

We will study permutation-based algorithms in the *constant* step size regime, that is, the step size is chosen at the beginning of the algorithm, and then remains fixed throughout. We denote the iterate after the $i$-th iteration of the $k$-th epoch by $\boldsymbol{x}_i^k$. Hence, the initialization point is $\boldsymbol{x}_0^1$. Similarly, the permutation of $(1, \ldots, n)$ used in the $k$-th epoch is denoted by $\sigma^k$, and its $i$-th ordered element is denoted by $\sigma_i^k$. Note that if the ambient space is 1-dimensional, then we represent the iterates and the minimizer using non-bold characters, *i.e.* $x_i^k$ and $x^*$, to remain consistent with the notation.

In the following, due to lack of space, we only provide sketches of the full proofs, when possible. The full proofs of the lemmas and theorems are provided in the Appendix.

## 4 EXPONENTIAL CONVERGENCE IN 1-DIMENSION

In this section, we show that there exist permutations for Hessian-smooth 1-dimensional functions that lead to exponentially faster convergence compared to random.

**Assumption 4** (Component Hessian-smoothness). $f_i(x)$'s have $L_H$-smooth second derivatives, that is,
$$\forall x, y : |\nabla^2 f_i(x) - \nabla^2 f_i(y)| \leq L_H |x - y|.$$

We also define the following instance dependent constants: $G^* := \max_i \|\nabla f_i(\boldsymbol{x}^*)\|$, $D = \max\left\{\|\boldsymbol{x}_0^1 - \boldsymbol{x}^*\|, \frac{G^*}{2L}\right\}$, and $G = G^* + 2DL$.

**Theorem 1.** *Let Assumptions 1,2,3 and 4 hold. Let $D$ and $G$ be as defined above. If $\alpha = \frac{\mu}{8n(L^2 + L_H G)}$, then there exists a sequence of permutations $\sigma^1, \sigma^2, \ldots, \sigma^K$ such that using those permutations from any initialization point $x_0^1$ gives the error*
$$|x_n^K - x^*| \leq (D + 4n\alpha G)e^{-CK},$$
*where $C = \frac{\mu^2}{16(L^2 + L_H G)}$.*

An important thing to notice in the theorem statement is that the sequence of permutations $\sigma^1, \sigma^2, \ldots, \sigma^K$ only depends on the function, and not on the initialization point $x_0^1$. This implies that for any such function, there exists a sequence of permutations $\sigma^1, \sigma^2, \ldots, \sigma^K$, which gives exponentially fast convergence, unconditionally of the initialization. Note that the convergence rate is slower than Gradient Descent, for which the constant '$C$' would be larger. However, here we are purely interested in the convergence rates of the best permutations and their (asymptotic) dependence on $K$.

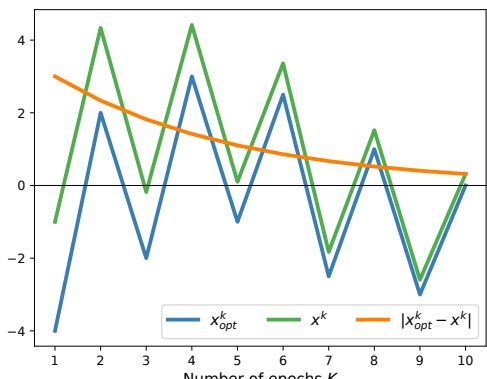

Figure 1: (A graphical depiction of Theorem 1's proof sketch.) Assume that the minimizer is at the origin. The proof of Theorem 1 first shows that there exists an initialization and a sequence of permutations, such that using those, we get to the exact minimizer. Let the sequence of iterates for this run be $x_{opt}^k$. Consider a parallel run, which uses the same sequence of permutations, but an arbitrary initialization point. Let this sequence be $x_k$. The figure shows how $x_{opt}^k$ converges to the exact optima, and the distance between $x_{opt}^k$ and $x_k$ decreases exponentially, leading to an exponential convergence for $x_k$.

**Proof sketch** The core idea is to establish that there exists an initialization point $x_0^1$ (close to the minimizer $x^*$), and a sequence of permutations such that that starting from $x_0^1$ and using that sequence of permutation leads us exactly to the minimizer. Once we have proved this, we show that if two parallel runs of the optimization process are initialized from two different iterates, and they are coupled so that they use the exact same permutations, then they approach each other at an exponential rate. Thus, if we use the same permutation from any initialization point, it will converge to the minimizer at an exponential rate. See Figure 1 for a graphical depiction of this sketch. We note that the figure is not the result of an actual run, but only serves to explain the proof sketch.

## 5 LOWER BOUNDS FOR PERMUTATION-BASED SGD

The result in the previous section leads us to wonder if exponentially fast convergence can be also achieved in higher dimensions. Unfortunately, for higher dimensions, there exist strongly convex quadratic functions for which there does not exist any sequence of permutations that lead to an exponential convergence rate. This is formalized in the following theorem.

**Theorem 2.** *For any $n \geq 4$ ($n$ must be even), there exists a $2n + 1$-dimensional strongly convex function $F$ which is the mean of $n$ convex quadratic functions, such that for every permutation-based algorithm with any step size,*

$$\|\boldsymbol{x}_n^{K-1} - \boldsymbol{x}^*\|^2 = \Omega\left(\frac{1}{n^3 K^2}\right).$$

This theorem shows that we cannot hope to develop constant step size algorithms that exhibit exponentially fast convergence in multiple dimensions.

**Proof sketch** Here we give a proof sketch for a simpler version of the theorem, which works in 2-Dimensions, for $n = 2$, and when the step size $\alpha = \Omega(1/K)$. Consider $F(x, y) = \frac{1}{2}f_1(x, y) + \frac{1}{2}f_2(x, y)$ such that

$$f_1(x, y) = \frac{x^2}{2} - x + y, \text{ and} f_2(x, y) = \frac{y^2}{2} - y + x.$$

Hence, $F = \frac{1}{4}(x^2 + y^2)$, and has minimizer at the origin. Each epoch has two possible permutations, $\sigma = (1, 2)$ or $\sigma = (2, 1)$. Working out the details manually, it can be seen that regardless of the permutation, $x_0^{k+1} + y_0^{k+1} > x_0^k + y_0^k$, that is, the sum of the co-ordinates keeps increasing. This can be used to get a bound on the error term $\|[x^K \quad y^K]^\top\|^2$.

Next, we show that even in 1-Dimension, individual function convexity might be necessary to obtain faster rates than RANDOM RESHUFFLE.

**Theorem 3.** *There exists a 1-Dimensional strongly convex function $F$ which is the mean of two quadratic functions $f_1$ and $f_2$, such that one of the functions is non-convex. Then, every permutation-based algorithm with constant step size $\alpha \leq \frac{1}{L}$ gives an error of at least*

$$\|\boldsymbol{x}_n^{K-1} - \boldsymbol{x}^*\|^2 = \Omega\left(\frac{1}{K^2}\right).$$

**Proof sketch** The idea behind the sketch is to have one of the two component functions as strongly concave. This gives it the advantage that the farther away from its maximum the iterate is, the more it pushes the iterate away. Hence, it essentially results in increasing the deviation in each epoch. This leads to a slow convergence rate.

In the setting where the individual $f_i$ may be non-convex, Nguyen et al. (2020) and Ahn et al. (2020) show that SINGLE SHUFFLE, RANDOM RESHUFFLE, and INCREMENTAL GRADIENT DESCENT achieve the error rate of $\widetilde{\mathcal{O}}(\frac{1}{K^2})$, for a fixed $n$. In particular, their results only need that the component functions be smooth and hence their results apply to the function $F$ from Theorem 3. The theorem above essentially shows that when $n = 2$, this is the best possible error rate, for any permutation-based algorithm - deterministic or random. Hence, at least for $n = 2$, the three algorithms are optimal when the component functions can possibly be non-convex. However, note that here we are only considering the dependence of the convergence rate on $K$. It is possible that these are not optimal, if we further take into account the dependence of the convergence rate on the combination of both $n$ and $K$. Indeed, if we consider the dependence on $n$ as well, INCREMENTAL GRADIENT DESCENT has a convergence rate of $\Omega(1/K^2)$ (Safran & Shamir, 2019), whereas the other two have a convergence rate of $\widetilde{O}(1/nK^2)$ (Ahn et al., 2020).

## 6 FLIPPING PERMUTATIONS FOR FASTER CONVERGENCE IN QUADRATICS

In this section, we introduce a new algorithm FLIPFLOP, that can improve the convergence rate of SINGLE SHUFFLE, RANDOM RESHUFFLE, and INCREMENTAL GRADIENT DESCENT on strongly convex quadratic functions.

The following theorem gives the convergence rate of FLIPFLOP WITH SINGLE SHUFFLE:

**Assumption 5.** $f_i(\boldsymbol{x})$'s are quadratic.

**Theorem 4.** *If Assumptions 1, 2, 3, and 5 hold, then running FLIPFLOP WITH SINGLE SHUFFLE for $K$ epochs, where $K \geq 80\kappa^{3/2}\log(nK)\max\left\{1, \frac{\sqrt{\kappa}}{n}\right\}$ is an even integer, with step size $\alpha = \frac{10\log(nK)}{\mu nK}$ gives the error*

$$\mathbb{E}\left[\left\|\boldsymbol{x}_n^{K-1} - \boldsymbol{x}^*\right\|^2\right] = \widetilde{\mathcal{O}}\left(\frac{1}{n^2K^2} + \frac{1}{nK^4}\right). \tag{4}$$

For comparison, Safran & Shamir (2019) give the following lower bound on the convergence rate of vanilla SINGLE SHUFFLE:

$$\mathbb{E}\left[\left\|\boldsymbol{x}_n^{K-1} - \boldsymbol{x}^*\right\|^2\right] = \Omega\left(\frac{1}{nK^2}\right), \tag{5}$$

Note that both the terms in Eq. (4) are smaller than the term in Eq. (5). In particular, when $n \gg K^2$ and $n$ is fixed as we vary $K$, the RHS of Eq. (5) decays as $\widetilde{\mathcal{O}}(\frac{1}{K^2})$, whereas the RHS of Eq. (4) decays as $\widetilde{\mathcal{O}}(\frac{1}{K^4})$. Otherwise, when $K^2 \gg n$ and $K$ is fixed as we vary $n$, the RHS of Eq. (5) decays as $\widetilde{\mathcal{O}}(\frac{1}{n})$, whereas the RHS of Eq. (4) decays as $\widetilde{\mathcal{O}}(\frac{1}{n^2})$. Hence, in both the cases, FLIPFLOP WITH SINGLE SHUFFLE outperforms SINGLE SHUFFLE.

The next theorem shows that FLIPFLOP improves the convergence rate of RANDOM RESHUFFLE:

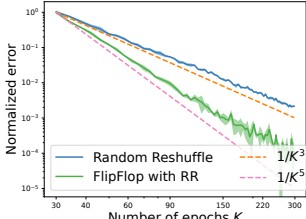 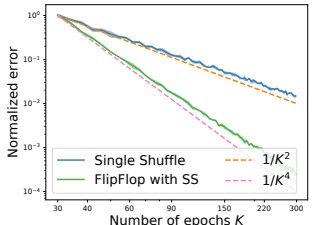 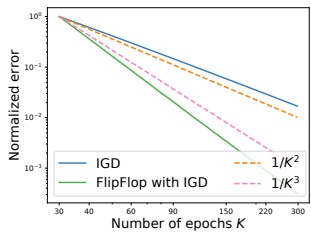

Figure 2: Dependence of convergence rates on the number of epochs $K$ for quadratic functions. The figures show the median and inter-quartile range after 10 runs of each algorithm, with random initializations and random permutation seeds (note that SS and IGD exhibit extremely small variance). We set $n = 800$, so that $n \gg K$ and hence the higher order terms of $K$ dominate the convergence rates. Note that both the axes are in logarithmic scale.

**Theorem 5.** *If Assumptions 1, 2, 3, and 5 hold, then running* FLIPFLOP WITH RANDOM RESHUFFLE *for $K$ epochs, where $K \geq 55\kappa \log(nK) \max\left\{1, \sqrt{\frac{n}{\kappa}}\right\}$ is an even integer, with step size $\alpha = \frac{10 \log(nK)}{\mu nK}$ gives the error*

$$\mathbb{E}\left[\left\|\boldsymbol{x}_n^{K-1} - \boldsymbol{x}^*\right\|^2\right] = \widetilde{\mathcal{O}}\left(\frac{1}{n^2 K^2} + \frac{1}{nK^5}\right).$$

For comparison, Safran & Shamir (2019) give the following lower bound on the convergence rate of vanilla RANDOM RESHUFFLE:

$$\mathbb{E}\left[\left\|\boldsymbol{x}_n^{K-1} - \boldsymbol{x}^*\right\|^2\right] = \Omega\left(\frac{1}{n^2 K^2} + \frac{1}{nK^3}\right).$$

Hence, we see that in the regime when $n \gg K$, which happens when the number of components in the finite sum of $F$ is much larger than the number of epochs, FLIPFLOP WITH RANDOM RESHUFFLE is much faster than vanilla RANDOM RESHUFFLE.

Note that the theorems above do not contradict Theorem 2, because for a fixed $n$, both the theorems above give a convergence rate of $\widetilde{\mathcal{O}}(1/K^2)$.

We also note here that the theorems above need the number of epochs to be much larger than $\kappa$, in which range Gradient Descent performs better than with- or without- replacement SGD, and hence GD should be preferred over SGD in that case. However, we think that this requirement on epochs is a limitation of our analysis, rather than that of the algorithms.

Finally, the next theorem shows that FLIPFLOP improves the convergence rate of INCREMENTAL GRADIENT DESCENT.

**Theorem 6.** *If Assumptions 1, 2, 3, and 5 hold, then running* FLIPFLOP WITH INCREMENTAL GD *for $K$ epochs, where $K \geq 36\kappa \log(nK)$ is an even integer, with step size $\alpha = \frac{6 \log nK}{\mu nK}$ gives the error*

$$\mathbb{E}\left[\left\|\boldsymbol{x}_n^{K-1} - \boldsymbol{x}^*\right\|^2\right] = \widetilde{\mathcal{O}}\left(\frac{1}{n^2 K^2} + \frac{1}{K^3}\right).$$

For comparison, Safran & Shamir (2019) give the following lower bound on the convergence rate of vanilla INCREMENTAL GRADIENT DESCENT:

$$\mathbb{E}\left[\left\|\boldsymbol{x}_n^{K-1} - \boldsymbol{x}^*\right\|^2\right] = \Omega\left(\frac{1}{K^2}\right),$$

In the next subsection, we give a short sketch of the proof of these theorems.

## 6.1 PROOF SKETCH

In the proof sketch, we consider scalar quadratic functions. The same intuition carries over to multi-dimensional quadratics, but requires a more involved analysis. Let $f_i(x) := \frac{a_i x^2}{2} + b_i x + c$.

Assume that $F(x) := \frac{1}{n} \sum_{i=1}^{n} f_i(x)$ has minimizer at 0. This assumption is valid because it can be achieved by a simple translation of the origin (see (Safran & Shamir, 2019) for a more detailed explanation). This implies that $\sum_{i=1}^{n} b_i = 0$.

For the sake of this sketch, assume $x_0^1 = 0$, that is, we are starting at the minimizer itself. Further, without loss of generality, assume that $\sigma = (1, 2, \dots, n)$. Then, for the last iteration of the first epoch,

$$
\begin{aligned}
x_n^1 &= x_{n-1}^1 - \alpha f_n'(x_{n-1}^1) \\
&= x_{n-1}^1 - \alpha(a_n x_{n-1}^1 + b_n) \\
&= (1 - \alpha a_n) x_{n-1}^1 - \alpha b_n.
\end{aligned}
$$

Applying this to all iterations of the first epoch, we get

$$
x_n^1 = \prod_{i=1}^{n}(1 - \alpha a_i) x_0^1 - \alpha \sum_{i=1}^{n} b_i \prod_{j=i+1}^{n}(1 - \alpha a_j). \tag{6}
$$

Substituting $x_0^1 = 0$, we get

$$
x_n^1 = -\alpha \sum_{i=1}^{n} b_i \prod_{j=i+1}^{n}(1 - \alpha a_j). \tag{7}
$$

Note that the sum above is not weighted uniformly: $b_1$ is multiplied by $\prod_{j=2}^{n}(1 - \alpha a_j)$, whereas $b_n$ is multiplied by 1. Because $(1 - \alpha a_j) < 1$, we see that $b_1$'s weight is much smaller than $b_n$. If the weights were all 1, then we would get $x_n^0 = -\alpha \sum_{i=1}^{n} b_i = 0$, i.e., we would not move away from the minimizer. Since we want to stay close to the minimizer, we want the weights of all the $b_i$ to be roughly equal.

The idea behind FLIPFLOP is to add something like $-\alpha \sum_{i=1}^{n} b_i \prod_{j=1}^{i}(1 - \alpha a_j)$ in the next epoch, to counteract the bias in Eq. (7). To achieve this, we simply take the permutation that the algorithm used in the previous epoch and flip it for the next epoch. Roughly speaking, in the next epoch $b_1$ will get multiplied by 1 whereas $b_n$ will get multiplied by $\prod_{j=1}^{n-1}(1 - \alpha a_j)$. Thus over two epochs, both get scaled approximately the same.

The main reason that the analysis for multidimensional quadratics is not as simple as the 1-dimensional analysis sketched above is because unlike scalar multiplication, matrix multiplication is not commutative, and the AM-GM inequality is not true in higher dimensions (Lai & Lim, 2020; De Sa, 2020). One way to bypass this inequality is by using the following inequality for small enough $\alpha$:

$$
\left\| \prod_{i=1}^{n}(\boldsymbol{I} - \alpha \boldsymbol{A}_i) \prod_{i=1}^{n}(\boldsymbol{I} - \alpha \boldsymbol{A}_{n-i+1}) \right\| \leq 1 - \alpha n \mu.
$$

Ahn et al. (2020) proved a stochastic version of this (see Lemma 6 in their paper). We prove a deterministic version in Lemma 3 (in the Appendix).

## 6.2 NUMERICAL VERIFICATION

We verify the theorems numerically by running RANDOM RESHUFFLE, SINGLE SHUFFLE and their FLIPFLOP versions on the task of mean computation. We randomly sample $n = 800$ points from a 100-dimensional sphere. Let the points be $\boldsymbol{x}_i$ for $i = 1, \dots, n$. Then, their mean is the solution to the following quadratic problem : $\arg \min_{\boldsymbol{x}} F(\boldsymbol{x}) = \frac{1}{n} \sum_{i=1}^{n} \|\boldsymbol{x} - \boldsymbol{x}_i\|^2$. We solve this problem by using the given algorithms. The results are reported in Figure 2. The results are plotted in a log–log graph, so that we get to see the dependence of error on the power of $K$.

Note that since the points are sampled randomly, INCREMENTAL GRADIENT DESCENT essentially becomes SINGLE SHUFFLE. Hence, to verify Theorem 6, we need 'hard' instances of INCREMENTAL GRADIENT DESCENT, and in particular we use the ones used in Theorem 3 in (Safran & Shamir, 2019). These results are also reported in a log–log graph in Figure 2.

We also tried FLIPFLOP in the training of deep neural networks, but unfortunately we did not see a big speedup there. We ran experiments on logistic regression for 1-Dimensional artificial data, the results for which are in Figure 3, and its details are in Appendix H. The code for all the experiments can be found at `https://github.com/shashankrajput/flipflop`.

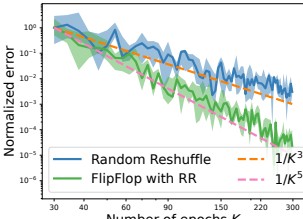 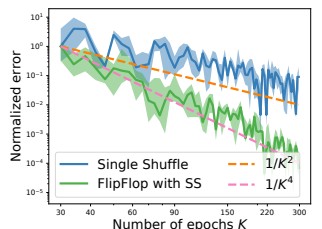 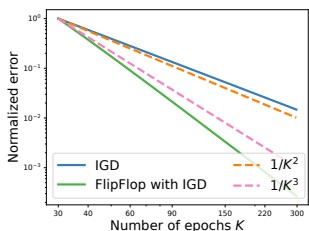

Figure 3: Dependence of convergence rates on the number of epochs $K$ for logistic regression. The figures show the median and inter-quartile range after 10 runs of each algorithm, with random initializations and random permutation seeds (note that IGD exhibits extremely small variance). We set $n = 800$, so that $n \gg K$ and hence the higher order terms of $K$ dominate the convergence rates. Note that both the axes are in logarithmic scale.

### 6.3 FASTER PERMUTATIONS FOR NON-QUADRATIC OBJECTIVES

The analysis of FlipFlop leverages the fact that the Hessians of quadratic functions are constant. We think that the analysis of FlipFlop might be extended to strongly convex functions or even PŁ functions (which are non-convex in general), under some assumptions on the Lipschitz continuity of the Hessians, similar to how Haochen & Sra (2019) extended their analysis of quadratic functions to more general classes. A key take-away from FlipFlop is that we had to understand how permutation based SGD works specifically for quadratic functions, that is, we did a white-box analysis. In general, we feel that depending on the specific class of non-convex functions (say deep neural networks), practitioners would have to think about permutation-based SGD in a white-box fashion, to come up with better heuristics for shuffling.

In a concurrent work by Lu et al. (2021), it is shown that by greedily sorting stale gradients, a permutation order can be found which converges faster for some deep learning tasks. Hence, there do exist better permutations than random, even for deep learning tasks.

## 7 CONCLUSION AND FUTURE WORK

In this paper, we explore the theoretical limits of permutation-based SGD for solving finite sum optimization problems. We focus on the power of good, carefully designed permutations and whether they can lead to a much better convergence rate than random. We prove that for 1-dimensional, strongly convex functions, indeed good sequences of permutations exist, which lead to a convergence rate which is exponentially faster than random permutations. We also show that unfortunately, this is not true for higher dimensions, and that for general strongly convex functions, random permutations might be optimal.

However, we think that for some subfamilies of strongly convex functions, good permutations might exist and may be easy to generate. Towards that end, we introduce a very simple technique, FLIPFLOP, to generate permutations that lead to faster convergence on strongly convex quadratics. This is a black box technique, that is, it does not look at the optimization problem to come up with the permutations; and can be implemented easily. This serves as an example that for other classes of functions, there can exist other techniques for coming up with good permutations. Finally, note that we only consider constant step sizes in this work for both upper and lower bounds. Exploring regimes in which the step size changes, *e.g.*, diminishing step sizes, is a very interesting open problem, which we leave for future work. We think that the upper and lower bounds in this paper give some important insights and can help in the development of better algorithms or heuristics. We strongly believe that under nice distributional assumptions on the component functions, there can exist good heuristics to generate good permutations, and this should also be investigated in future work.

ETHICS STATEMENT

This paper explores the theoretical limits of convergence rates of variants of popular SGD type algorithms. Thus, the authors do not think there are any ethical concerns regarding the paper.

REPRODUCIBILITY STATEMENT

The proof for all the theoretical claims are provided in the Appendix. The assumptions and notations are specified in the Preliminaries section (Section 3). The code for all the experiments can be found at `https://github.com/shashankrajput/flipflop`.

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

## A  DISCUSSION AND FUTURE WORK

Being the first paper (to the best of our knowledge) to theoretically analyze the optimality of random permutations, we limited our scope to a specific, but common theoretical setting - strongly convex functions with constant step size. We think that future work can generalize the results of this paper to settings of non-convexity, variable step sizes, as well as techniques like variance reduction, momentum, etc.

### A.1  LOWER BOUNDS FOR VARIABLE STEP SIZES

All the existing lower bounds (to the best of our knowledge) work in the constant step size regime (Safran & Shamir (2019); Rajput et al. (2020); Safran & Shamir (2021)). Thus, generalizing the lower bounds to variable step size algorithms would be a very important direction for future research.

However, the case when step sizes are not constant can be tricky to prove lower bounds, since the step size could potentially depend on the permutation, and the current iterate. A more reasonable setting to prove lower bounds could be the case when the step sizes follow a schedule over epochs, similar to what happens in practice.

### A.2  FLIPFLOP ON RANDOM COORDINATE DESCENT

A shuffling scheme similar to FlipFlop has been used in random coordinate descent for faster practical convergence (see page 231 in Nocedal & Wright (2006)). This should be further investigated empirically and theoretically in future work. Even though the current analysis of FlipFlop does not directly go through for random coordinate descent, we think the analysis can be adapted to work.

## B  PROOF OF THEOREM 1

In certain places in the proof, we would need that $\alpha \leq \frac{1}{4nL}$. To see how this is satisfied, note that we have assumed that $\alpha \leq \frac{\mu}{8n(L^2+L_H G)}$ in the theorem statement. Using the inequality $\mu \leq L$ in $\alpha \leq \frac{\mu}{4n(L^2+L_H G)}$ gives that $\alpha \leq \frac{1}{4n(L+(L_H G/\mu))} \leq \frac{1}{4nL}$.

In this proof, we assume that the minimizer of $F$ is at $0$ to make the analysis simpler. This assumption can be satisfied by simply translating the origin to the minimizer (Safran & Shamir, 2019).

There are three main components in the proof:

1. Say that an epoch starts off at the minimizer. We prove that there exists at least one pair of permutations such that if we do two separate runs of the epoch, the first using the first permutation and the second using the second, then at the end of that epoch, the iterates corresponding to the two runs end up on opposite sides of the minimizer.

2. There exists a sequence of permutations and a point in the neighborhood of the minimizer, such that intializing at that point and using these sequence of permutations, we converge exactly to the minimizer.

3. Starting from any other point, we can couple the iterates with the iterates which were shown in the previous component, to get that these two sequences of iterates come close to each other exponentially fast.

We prove the first and second components in the Subsections B.1 and B.2; and conclude the proof in Subsection B.3 where we also prove the third component.

### B.1  PERMUTATIONS IN ONE EPOCH

In this subsection, we prove that if $x_0, x_1, \ldots, x_n$ are the iterates in an epoch such that $x_0 = 0$, then there exists a permutation of functions such that $x_n \geq 0$. By the same logic, we show that there exists a permutation of functions such that $x_n \leq 0$. These will give us control over movement of iterates across epochs.

Order the gradients at the minimizer, $\nabla f_i(0)$, in decreasing order. WLOG assume that it is just $\nabla f_1, \nabla f_2, \dots, \nabla f_n$. We claim that this permutation leads to $x_n \geq 0$.

We will need the following intermediate result. Let $y_i, y_{i-1}$ be such that $y_i = y_{i-1} - \alpha \nabla f_i(y_{i-1})$. Assume $\alpha \leq 1/L$ and $y_{i-1} \geq x_{i-1}$. Then,

$$
\begin{aligned}
y_i - x_i &= y_{i-1} - x_{i-1} - \alpha(\nabla f_i(y_{i-1}) - \nabla f_i(x_{i-1})) \\
&\geq y_{i-1} - x_{i-1} - \alpha L(y_{i-1} - x_{i-1}) \\
&= (1 - \alpha L)(y_{i-1} - x_{i-1}) \\
&\geq 0,
\end{aligned}
\tag{8}
$$

that is, $y_{i-1} \geq x_{i-1} \implies y_i \geq x_i$.

Because $0$ is the minimizer, we know that $\sum_{i=1}^n \nabla f_i(0) = 0$. Also, recall that $x_0 = 0$. There can be two cases:

1. $\forall i \in [1, n] : x_i < 0$. This cannot be true because if $\forall i : 1 \leq i \leq n - 1 : x_i < 0$, then

$$
x_n = \sum_{i=1}^n -\alpha \nabla f_i(x_{i-1}) \geq -\alpha \sum_{i=1}^n \nabla f_i(0) \geq 0,
$$

where we used the fact that $\nabla f_i(x) \leq \nabla f_i(y)$ if $x \leq y$ and $f_i$ is convex.

2. Thus, $\exists i \in [1, n] : x_i \geq 0$. Now, consider the sequence $y_i, y_{i+1}, \dots, y_n$ such that $y_i = 0$ and for $j \geq i + 1$, $y_j = y_{j-1} - \alpha \nabla f_j(y_{j-1})$. Then because $\alpha \leq 1/L$ and $x_i \geq y_i = 0$, we get that $x_j \geq y_j$ for all $j \geq i$ (Using Ineq. (8)).

Hence, there is an $i \geq 1$ such that $x_i \geq y_i = 0$, and further $x_j \geq y_j$ for all $j \geq i$. Next, we repeat the process above for $y_i, \dots, y_n$. That is, there can be two cases:

1. $\forall j \in [i + 1, n] : y_j < 0$. This cannot be true because if $\forall j : i + 1 \leq j \leq n - 1 : y_j < 0$, then

$$
y_n = \sum_{j=i}^n -\alpha \nabla f_j(y_j) \geq -\alpha \sum_{j=i}^n \nabla f_j(0) \geq 0.
$$

2. Thus, $\exists j \in [i + 1, n] : y_j \geq 0$. Now, consider the sequence $z_j, z_{j+1}, \dots, z_n$ such that $z_j = 0$ and for $k \geq j + 1$, $z_k = z_{k-1} - \alpha \nabla f_k(z_{k-1})$. Then because $\alpha \leq 1/L$ and $y_j \geq z_j = 0$, we get that $y_k \geq z_k$ for all $k \geq j$ (Using Ineq. (8)).

Hence, there exists an integer $j > i > 0$ such that $y_j \geq 0$. We have already proved that $x_j \geq y_j$. Thus, we have that $x_j \geq 0$. We can continue repeating this process (apply the same two cases above for $z_j, z_{j+1}, \dots, z_n$, and so on), to get that $x_n \geq 0$. We define $p$ to be this non-negative value of $x_n$. Note that the following lemma gives us that the gradients are bounded by $G$

**Lemma 1.** *Define $G^* := \max_i \|\nabla f_i(\boldsymbol{x}^*)\|$ and $D = \max\left\{ \|\boldsymbol{x}_0^1 - \boldsymbol{x}^*\|, \frac{G^*}{2L} \right\}$. If Assumptions 2 and 3 hold, and $\alpha < \frac{1}{8\kappa n L}$, then for any permutation-based algorithm (deterministic or random), we have*

$$
\begin{aligned}
\forall i, j, k : \|\boldsymbol{x}_i^k - \boldsymbol{x}^*\| &\leq 2D, \quad \text{and} \\
\|\nabla f_j(\boldsymbol{x}_i^k)\| &\leq G^* + 2DL.
\end{aligned}
$$

Because the gradients are bounded by $G$, we get that $p \leq n\alpha G$.

Similarly, we can show that the reverse permutation leads to $x_n \leq 0$. We define $q$ to be this non-positive value of $x_n$. Because we have assumed that the gradients are bounded by $G$, we get that $q \geq -n\alpha G$.

## B.2 Exact convergence to the minimizer

In this section, we show that there exists a point such that if we initialize there and follow a specific permutation sequence, we land exactly at the minimizer.

In particular, we will show the following: There exists a point in $[4q, 4p]$ such that if we initialize there and follow a specific permutation sequence, then we land exactly at the minimizer.

We show this recursively. We will prove that there exists a point $m^K \in [4q, 4p]$ such that if the last epoch begins there, that is $x_0^K = m^K$, then we land at the minimizer at the end of the last epoch, that is, $x_n^K = 0$. Then, we will show that there exists a point $m^{K-1} \in [4q, 4p]$ such that if the $x_0^{K-1} = m^{K-1}$, then $x_0^K = x_n^{K-1} = m^K$. Repeating this for $K - 2, \ldots, 1$, we get that there exists a point $m^0 \in [4q, 4p]$ such that if we initialize the first epoch there, that is $x_0^1 = m^0$, then there is a permutation sequence such that ultimately $x_n^K = 0$.

We prove that any point $m^j \in [4q, 4p]$ can be reached at the end of an epoch by beginning the epoch at some point $m^{j-1} \in [4q, 4p]$, that is if $x_0^{j-1} = m^{j-1}$, then $x_n^{j-1} = m^j$.

- Case: $m^j \in [p, 4p]$. In this case, we show that $m^{j-1} \in [0, 4p]$. We have proved in the previous subsection that there exists a permutation $\sigma$ such that if $x_0^{j-1} = 0$ then $x_n^{j-1} = p$. Next, we have the following helpful lemma that we will also use later.

  **Lemma 2.** *Let $x_0, x_1, \ldots, x_n$ be a sequence of iterates in an epoch and $y_0, y_1, \ldots, y_n$ be another sequence of iterates in an epoch such that both use the same permutation of functions. If $\alpha \leq \frac{\mu}{2n(L^2 + L_H G)}$, then*

  $$(1 - n\alpha L)|y_0 - x_0| \leq (1 - L\alpha)^n |y_0 - x_0| \leq |y_n - x_n| \leq \left(1 - \frac{1}{2} n\mu\alpha\right)|y_0 - x_0|.$$

  If we set $x_0 = 4p, y_0 = 0$ in Lemma 2 and we follow the permutation $\sigma$, then we get that

  $$x_n - y_n \in (x_0 - y_0)\left[1 - \alpha n L, 1 - \frac{\alpha n \mu}{2}\right]$$
  $$\implies x_n - p \in (4p - 0)\left[1 - \alpha n L, 1 - \frac{\alpha n \mu}{2}\right]$$
  $$\text{(Since using } \sigma \text{ and } y_0 = 0 \text{ results in } y_n = p.)$$
  $$\implies x_n \geq 4p,$$

  where we used the fact that $\alpha \leq \frac{1}{4nL}$ is the last step.

  Thus, if $x_0^{j-1} = 4p$ and we follow the permutation $\sigma$, then $x_n^{j-1} \geq 4p$.

  Next, note that

  $$x_1^{j-1} = x_0^{j-1} - \alpha \nabla f_{\sigma(0)}(x_0^{j-1}).$$
  $$x_2^{j-1} = x_1^{j-1} - \alpha \nabla f_{\sigma(1)}(x_1^{j-1})$$
  $$\vdots$$
  $$x_n^{j-1} = x_{n-1}^{j-1} - \alpha \nabla f_{\sigma(n-1)}(x_{n-1}^{j-1})$$

  Looking above, we see that $x_1^{j-1}$ is a continuous function of $x_0^{j-1}$; $x_2^{j-1}$ is a continuous function of $x_1^{j-1}$; and so on. Thus, using the fact that composition of continuous functions is continuous, we get that $x_n^{j-1}$ is also a continuous function of $x_0^{j-1}$. We have shown that if $x_0^{j-1} = 0$, then $x_n^{j-1} = p$ and if $x_0^{j-1} = 4p$, then $x_n^{j-1} \geq 4p$. Thus, using the fact that that $x_n^{j-1}$ is a continuous function of $x_0^{j-1}$, we get that for any point $m^j \in [p, 4p]$, there is at least one point $m^{j-1} \in [0, 4p]$, such that $x_0^{j-1} = m^{j-1}$ leads to $x_n^{j-1} = m^j$.

- Case: $m_j \in [4q, q]$. We can apply the same logic as above to show that there is at least one point $m^{j-1} \in [4q, 0]$, such that $x_0^{j-1} = m^{j-1} \implies x_n^{j-1} = m^j$.

- Case: $m_j \in [q, p]$. WLOG assume that $|q| < |p|$. Let $\sigma_q$ be the permutation such that if $x_0^{j-1} = 0$ and the epoch uses this permutation, then we end up at $x_n^{j-1} = q$.

  If we set $x_0 = 4p, y_0 = 0$ in Lemma 2 and we follow the permutation $\sigma_q$, then we get that

  $$x_n - y_n \in (x_0 - y_0)\left[1 - \alpha n L, 1 - \frac{\alpha n \mu}{2}\right]$$

$$\implies x_n - q \in (4p - 0)\left[1 - \alpha nL, 1 - \frac{\alpha n\mu}{2}\right]$$

$$\text{(Since using } \sigma_q \text{ and } y_0 = 0 \text{ results in } y_n = q.)$$

$$\implies x_n \geq q + 4p(1 - \alpha nL) \geq q + 3p \geq 2p,$$

where we used the fact that $\alpha \leq \frac{1}{4nL}$ is the last step.

Thus, if $x_0^{j-1} = 4p$ and we follow the permutation $\sigma_q$, then $x_n^{j-1} \geq 2p$.

Thus, using similar argument of continuity as the first case, we know that there is a point $m^{j-1} \in [0, 4p]$, such that $x_0^{j-1} = m^{j-1}$ leads to $x_n^{j-1} = m^j$ when we use the permutation $\sigma_q$.

## B.3 SAME SEQUENCE PERMUTATIONS GET CLOSER

In the previous subsection, we have shown that there exists a point $m^0 \in [4q, 4p]$ and a sequence of permutations $\sigma^1, \sigma^2, \ldots, \sigma^K$ such that if $x_0^1 = m^0$ and epoch $j$ uses permutation $\sigma^j$, then $x_n^K = 0$. In this subsection, we will show that if $x_0^1$ is initialized at any other point such that $|x_0^1| \leq D$, then using the same permutations $\sigma^1, \sigma^2, \ldots, \sigma^K$ gives us that $|x_n^K| \leq (D + n\alpha G)e^{-K}$. For this we will repeatedly apply Lemma 2 on all the $K$ epochs. Assume that $x_0^1 = \nu^0$ with $|\nu^0| \leq D$.

Let $y_i^j$ be the sequence of iterates such that $y_0^1 = m^0$ and uses the permutation sequence $\sigma^1, \sigma^2, \ldots, \sigma^K$. Then, we know that $y_n^K = 0$. Let $x_i^j$ be the sequence of iterates such that $x_0^1 = \nu^0$ and uses the same permutation sequence $\sigma^1, \sigma^2, \ldots, \sigma^K$.

Then, using Lemma 2 gives us that $|y_n^1 - x_n^1| \leq |\nu^0 - m^0|(1 - \frac{\mu\alpha n}{2})$. Thus, we get that $|y_0^2 - x_0^2| \leq |\nu^0 - m^0|(1 - \frac{\mu\alpha n}{2})$. Again applying Lemma 2 gives us that $|y_n^2 - x_n^2| \leq |\nu^0 - m^0|(1 - \frac{\mu\alpha n}{2})^2$. Therefore, after applying it $K$ times, we get

$$|y_n^K - x_n^K| \leq |\nu^0 - m^0|\left(1 - \frac{\mu\alpha n}{2}\right)^K.$$

Note that $|x_0^1| = |\nu^0| \leq D$, and $|y_0^1| = |m^0|$, with $m^0 \in [4q, 4p]$. We showed earlier in Subsection B.1 that $|p| \leq n\alpha G$ and $|q| \leq n\alpha G$. Therefore,

$$|y_n^K - x_n^K| \leq |D + 4n\alpha G|\left(1 - \frac{\mu\alpha n}{2}\right)^K.$$

Further, we know that $y_n^K = 0$. Thus,

$$|x_n^K| \leq |D + 4n\alpha G|\left(1 - \frac{\mu\alpha n}{2}\right)^K$$
$$\leq |D + 4n\alpha G|e^{-\frac{\mu\alpha n}{2}K}.$$

Substituting the value of $\alpha$ completes the proof. Next we prove the lemmas used in this proof.

## B.4 PROOF OF LEMMA 1

We restate the lemma below.

**Lemma.** *Define* $G^* := \max_i \|\nabla f_i(\boldsymbol{x}^*)\|$ *and* $D = \max\left\{\|\boldsymbol{x}_0^1 - \boldsymbol{x}^*\|, \frac{G^*}{2L}\right\}$. *If Assumptions 2 and 3 hold, and* $\alpha < \frac{1}{8\kappa nL}$, *then for any permutation-based algorithm (deterministic or random), we have*

$$\forall i, j, k : \|\boldsymbol{x}_i^k - \boldsymbol{x}^*\| \leq 2D, \quad and$$
$$\|\nabla f_j(\boldsymbol{x}_i^k)\| \leq G^* + 2DL.$$

This lemma says that for any permutation-based algorithm, the domain of iterates and the norm of the gradient stays bounded during the optimization. This means that we can assume bounds on norm of iterates and gradients, which is not true in general for unconstrained SGD. This makes analyzing such algorithms much easier, and hence this lemma can be of independent interest for proving future results for permutation-based algorithms.

**Remark 1.** *This lemma does not hold in general for vanilla SGD where sampling is done with replacement. Consider the example with two functions $f_1(x) = x^2 - x$, and $f_2(x) = x$; and $F(x) = f_1(x) + f_2(x)$. This satisfies Assumptions 2 and 3, but one may choose $f_2$ consecutively for arbitrary many iterations, which will lead the iterates a proportional distance away from the minimizer. This kind of situation can never happen for permutation-based SGD because we see every function exactly once in every epoch and hence no particular function can attract the iterates too much towards its minimizer, and by the end of the epochs most of the noise gets cancelled out.*

Note that the requirement $\alpha < \frac{1}{8\kappa n L}$ is stricter than the usual requirement $\alpha = O\left(\frac{1}{nL}\right)$, but we believe the lemma should also hold in that regime. For the current paper, however, this stronger requirement suffices.

*Proof.* To prove this lemma, we show two facts: If for some epoch $k$, $\|\boldsymbol{x}_0^k - \boldsymbol{x}^*\| \leq D$, then a) $\forall i : \|\boldsymbol{x}_i^k - \boldsymbol{x}^*\| \leq 2D$ and b) $\|\boldsymbol{x}_0^{k+1} - \boldsymbol{x}^*\| = \|\boldsymbol{x}_n^k - \boldsymbol{x}^*\| \leq D$. To see how a) and b) are sufficient to prove the lemma, assume that they are true. Then, since the first epoch begins inside the bounded region $\|\boldsymbol{x}_0^1 - \boldsymbol{x}^*\| \leq D$, we see using b) that every subsequent epoch begins inside the same bounded region, that is $\|\boldsymbol{x}_0^k - \boldsymbol{x}^*\| \leq D$ as well. Hence using a) we get that during these epochs, the iterates satisfy $\|\boldsymbol{x}_i^k - \boldsymbol{x}^*\| \leq 2D$, which is the first part of the lemma. Further, this bound together with the gradient Lipshitzness directly gives the upper bound $G^* + 2DL$ on the gradients. Thus, all we need to do to prove this lemma is to prove a) and b), which we do next.

We will prove a) and b) for $D = \max\{\|\boldsymbol{x}_0^1 - \boldsymbol{x}^*\|, \frac{2\kappa n\alpha G^*}{1-4\kappa n\alpha L}\}$. Once we do this, using $\alpha < \frac{1}{8\kappa n L}$ will give us the exact statement of the lemma.

Let $|\boldsymbol{x}_0^k - \boldsymbol{x}^*| \leq D$ for some epoch $k$. Then, we try to find the minimum number of iterations $i$ needed so that $|\boldsymbol{x}_i^k - \boldsymbol{x}^*| \geq 2D$. Within this region, the gradient is bounded by $G^* + 2DL$. Thus, the minimum number of iterations needed are $\frac{2D-D}{\alpha(G^*+2DL)}$. However,

$$
\begin{aligned}
\frac{2D - D}{\alpha(G^* + 2DL)} &= \frac{1}{\alpha(\frac{G^*}{D} + 2L)} \\
&\geq \frac{1}{\alpha(G^* \frac{1-4\kappa n\alpha L}{2\kappa n\alpha G^*} + 2L)} && \left(\text{Using the fact that } D \geq \frac{2\kappa n\alpha G^*}{1-4\kappa n\alpha L}.\right) \\
&= \frac{1}{\alpha(\frac{1-4\kappa n\alpha L}{2\kappa n\alpha} + 2L)} \\
&= 2\kappa n \\
&\geq 2n.
\end{aligned}
$$

Thus, the minimum number iterations needed to go outside the bound $|\boldsymbol{x}_i^k - \boldsymbol{x}^*| \geq 2D$ is more than the length of the epoch. This implies that within the epoch, $\|\boldsymbol{x}_i^k - \boldsymbol{x}^*\| \leq 2D$, which proves a).

We prove b) next:

$$
\begin{aligned}
\|\boldsymbol{x}_n^k - \boldsymbol{x}^*\| &= \left\| \left( \boldsymbol{x}_0^k - \alpha \sum_{i=0}^n \nabla f_{\sigma_i^k}(\boldsymbol{x}_i^k) \right) - \boldsymbol{x}^* \right\| \\
&= \left\| \boldsymbol{x}_0^k - \boldsymbol{x}^* - \alpha \sum_{i=0}^n \nabla f_{\sigma_i^k}(\boldsymbol{x}_0^k) + \alpha \sum_{i=0}^n (\nabla f_{\sigma_i^k}(\boldsymbol{x}_0^k) - \nabla f_{\sigma_i^k}(\boldsymbol{x}_i^k)) \right\|
\end{aligned}
$$

Note that $\sum_{i=0}^n \nabla f_{\sigma_i^k}(\boldsymbol{x}_0^k)$ is just the sum of all component gradients at $\boldsymbol{x}_0^k$, that is $\sum_{i=0}^n \nabla f_{\sigma_i^k}(\boldsymbol{x}_0^k) = n\nabla F(\boldsymbol{x}_0^k)$. Using this, we get

$$
\begin{aligned}
\|\boldsymbol{x}_n^k - \boldsymbol{x}^*\| &= \left\| \boldsymbol{x}_0^k - \boldsymbol{x}^* - n\alpha\nabla F(\boldsymbol{x}_0^k) + \alpha \sum_{i=0}^n (\nabla f_{\sigma_i^k}(\boldsymbol{x}_0^k) - \nabla f_{\sigma_i^k}(\boldsymbol{x}_i^k)) \right\| \\
&\leq \left\| \boldsymbol{x}_0^k - \boldsymbol{x}^* - n\alpha\nabla F(\boldsymbol{x}_0^k) \right\| + \alpha \sum_{i=0}^n \left\| \nabla f_{\sigma_i^k}(\boldsymbol{x}_0^k) - \nabla f_{\sigma_i^k}(\boldsymbol{x}_i^k) \right\|
\end{aligned}
$$

(Triangle inequality.)

$$\leq \left\| \boldsymbol{x}_0^k - \boldsymbol{x}^* - n\alpha\nabla F(\boldsymbol{x}_0^k) \right\| + \alpha L \sum_{i=0}^n \left\| \boldsymbol{x}_0^k - \boldsymbol{x}_i^k \right\|, \tag{9}$$

where we used gradient Lipschitzness (Assumption 2) in the last step.

To bound the first term above, we use the standard analysis of gradient descent on smooth, strongly convex functions as follows

$$
\begin{aligned}
\left\| \boldsymbol{x}_0^k - \boldsymbol{x}^* - n\alpha\nabla F(\boldsymbol{x}_0^k) \right\|^2 &= \left\| \boldsymbol{x}_0^k - \boldsymbol{x}^* \right\|^2 - 2n\alpha\langle \boldsymbol{x}_0^k - \boldsymbol{x}^*, \nabla F(\boldsymbol{x}_0^k) \rangle + n^2\alpha^2 \| \nabla F(\boldsymbol{x}_0^k) \|^2 \\
&\leq \left\| \boldsymbol{x}_0^k - \boldsymbol{x}^* \right\|^2 - 2n\alpha\mu \| \boldsymbol{x}_0^k - \boldsymbol{x}^* \|^2 + n^2\alpha^2 \| \nabla F(\boldsymbol{x}_0^k) \|^2 \\
&\qquad\qquad\qquad\qquad\qquad\qquad\qquad\qquad\qquad\text{(Using Ineq. (3))} \\
&= (1 - n\alpha\mu) \left\| \boldsymbol{x}_0^k - \boldsymbol{x}^* \right\|^2 + n\alpha(n\alpha\| \nabla F(\boldsymbol{x}_0^k) \|^2 - \mu\| \boldsymbol{x}_0^k - \boldsymbol{x}^* \|^2) \\
&\leq (1 - n\alpha\mu) \left\| \boldsymbol{x}_0^k - \boldsymbol{x}^* \right\|^2 + n\alpha(n\alpha L^2 \| \boldsymbol{x}_0^k - \boldsymbol{x}^* \|^2 - \mu\| \boldsymbol{x}_0^k - \boldsymbol{x}^* \|^2) \\
&\qquad\qquad\qquad\qquad\qquad\qquad\qquad\text{(Using gradient Lipschitzness)} \\
&= (1 - n\alpha\mu) \left\| \boldsymbol{x}_0^k - \boldsymbol{x}^* \right\|^2 + n\alpha(n\alpha L^2 - \mu)\| \boldsymbol{x}_0^k - \boldsymbol{x}^* \|^2 \\
&\leq (1 - n\alpha\mu) \left\| \boldsymbol{x}_0^k - \boldsymbol{x}^* \right\|^2,
\end{aligned}
$$

where in the last step we used that $\alpha \leq \frac{\mu}{nL^2}$ since $\alpha \leq \frac{1}{8\kappa nL}$. Substituting this inequality in Ineq. (9), we get

$$
\begin{aligned}
\left\| \boldsymbol{x}_n^k - \boldsymbol{x}^* \right\| &\leq \sqrt{1 - n\alpha\mu} \left\| \boldsymbol{x}_0^k - \boldsymbol{x}^* \right\| + \alpha L \sum_{i=0}^n \left\| \boldsymbol{x}_0^k - \boldsymbol{x}_i^k \right\| \\
&\leq \left( 1 - \frac{1}{2}n\alpha\mu \right) \left\| \boldsymbol{x}_0^k - \boldsymbol{x}^* \right\| + \alpha L \sum_{i=0}^n \left\| \boldsymbol{x}_0^k - \boldsymbol{x}_i^k \right\|.
\end{aligned}
$$

We have already proven a) that says that the iterates $\boldsymbol{x}_i^k$ satisfy $\| \boldsymbol{x}_i^k - \boldsymbol{x}^* \| \leq 2D$. Using gradient Lipschitzness, this implies that the gradient norms stay bounded by $G^* + 2DL$. Hence, $\left\| \boldsymbol{x}_0^k - \boldsymbol{x}_i^k \right\| \leq \alpha i(G^* + 2DL)$. Using this,

$$
\begin{aligned}
\left\| \boldsymbol{x}_n^k - \boldsymbol{x}^* \right\| &\leq \left( 1 - \frac{1}{2}n\alpha\mu \right) \left\| \boldsymbol{x}_0^k - \boldsymbol{x}^* \right\| + \alpha L \sum_{i=0}^n \alpha i(G^* + 2DL) \\
&\leq \left( 1 - \frac{1}{2}n\alpha\mu \right) D + \alpha L \sum_{i=0}^n \alpha i(G^* + 2DL) \\
&\leq \left( 1 - \frac{1}{2}n\alpha\mu \right) D + n^2\alpha^2 L(G^* + 2DL) \\
&\leq D,
\end{aligned}
$$

where we used the fact that $D \geq \frac{2\kappa n\alpha G^*}{1 - 4\kappa n\alpha L}$ in the last step. $\qquad\square$

## B.5 Proof of Lemma 2

Without loss of generality, let $\sigma = (1, 2, 3, \ldots, n)$. This is only done for ease of notation. The analysis goes through for any other permutation $\sigma$ too.

First we show the lower bound. WLOG assume $y_0 > x_0$. Because $\alpha < 1/L$, we have that $\forall i, y_i > x_i$ by induction (see the equations below). Then,

$$
\begin{aligned}
y_i - x_i &= y_{i-1} - x_{i-1} - \alpha(\nabla f_i(y_{i-1}) - \nabla f_i(x_{i-1})) \\
&\geq y_{i-1} - x_{i-1} - \alpha L(y_{i-1} - x_{i-1}) \\
&= (1 - \alpha L)(y_{i-1} - x_{i-1}) \\
&\qquad\vdots \\
&= (1 - \alpha L)^i(y_0 - x_0)
\end{aligned}
$$

$$\geq (1 - i\alpha L)(y_0 - x_0). \tag{10}$$

Next we show the upper bound

$$y_n - x_n = y_0 - x_0 - \alpha \sum_{i=1}^{n} (\nabla f_i(y_{i-1}) - \nabla f_i(x_{i-1}))$$

$$= y_0 - x_0 - \alpha \sum_{i=1}^{n} (\nabla f_i(y_0) - \nabla f_i(x_0)) + \alpha \sum_{i=1}^{n} (\nabla f_i(y_0) - \nabla f_i(x_0) - \nabla f_i(y_{i-1}) + \nabla f_i(x_{i-1}))$$

$$= y_0 - x_0 - n\alpha(\nabla F(y_0) - \nabla F(x_0)) + \alpha \sum_{i=1}^{n} (\nabla f_i(y_0) - \nabla f_i(x_0) - \nabla f_i(y_{i-1}) + \nabla f_i(x_{i-1}))$$

$$\leq (1 - n\alpha\mu)(y_0 - x_0) + \alpha \sum_{i=1}^{n} (\nabla f_i(y_0) - \nabla f_i(x_0) - \nabla f_i(y_{i-1}) + \nabla f_i(x_{i-1})).$$

(Using strong convexity)

We use the fact that the function is twice differentiable:

$$y_n - x_n = (1 - n\alpha\mu)(y_0 - x_0) + \alpha \sum_{i=1}^{n} \left( \int_{x_0}^{y_0} \nabla^2 f_i(t) \mathrm{d}t - \int_{x_{i-1}}^{y_{i-1}} \nabla^2 f_i(t) \mathrm{d}t \right)$$

$$= (1 - n\alpha\mu)(y_0 - x_0)$$
$$+ \alpha \sum_{i=1}^{n} \left( \int_{x_0}^{y_0} \nabla^2 f_i(t) \mathrm{d}t - \int_{x_{i-1}}^{x_{i-1}+(y_0-x_0)} \nabla^2 f_i(t) \mathrm{d}t - \int_{x_{i-1}+(y_0-x_0)}^{y_{i-1}} \nabla^2 f_i(t) \mathrm{d}t \right)$$

$$= (1 - n\alpha\mu)(y_0 - x_0)$$
$$+ \alpha \sum_{i=1}^{n} \left( \int_{x_0}^{y_0} (\nabla^2 f_i(t) - \nabla^2 f_i(x_{i-1} - x_0 + t)) \mathrm{d}t - \int_{x_{i-1}+(y_0-x_0)}^{y_{i-1}} \nabla^2 f_i(t) \mathrm{d}t \right).$$

In the above, we used the convention that $\int_a^b f(x)dx$ is the same as $-\int_b^a f(x)dx$ if $a > b$.

Now, we can use the Hessian Lipschitzness to bound the term as follows

$$y_n - x_n \leq (1 - n\alpha\mu)(y_0 - x_0) + \alpha \sum_{i=1}^{n} \left( \int_{x_0}^{y_0} L_H |x_{i-1} - x_0| \mathrm{d}t - \int_{x_{i-1}+(y_0-x_0)}^{y_{i-1}} \nabla^2 f_i(t) \mathrm{d}t \right)$$

$$\leq (1 - n\alpha\mu)(y_0 - x_0) + \alpha \sum_{i=1}^{n} \left( \int_{x_0}^{y_0} L_H G\alpha n \mathrm{d}t - \int_{x_{i-1}+(y_0-x_0)}^{y_{i-1}} \nabla^2 f_i(t) \mathrm{d}t \right)$$

$$= (1 - n\alpha\mu)(y_0 - x_0) + L_H G\alpha^2 n^2 (x_0 - y_0) - \alpha \sum_{i=1}^{n} \int_{x_{i-1}+(y_0-x_0)}^{y_{i-1}} \nabla^2 f_i(t) \mathrm{d}t$$

$$\leq (1 - n\alpha\mu)(y_0 - x_0) + L_H G\alpha^2 n^2 (x_0 - y_0) + \alpha \sum_{i=1}^{n} L((y_0 - x_0) - (y_i - x_i))$$

$$\leq (1 - n\alpha\mu)(y_0 - x_0) + L_H G\alpha^2 n^2 (x_0 - y_0) + \alpha \sum_{i=1}^{n} L(i\alpha L)(y_0 - x_0)$$

(Using Ineq. (10).)

$$\leq (1 - n\alpha\mu)(y_0 - x_0) + L_H G\alpha^2 n^2 (x_0 - y_0) + \alpha^2 n^2 L^2 (y_0 - x_0).$$

Thus, if we have $\alpha \leq \frac{\mu}{2n(L^2 + L_H G)}$, then

$$y_n - x_n \leq \left( 1 - \frac{\mu n\alpha}{2} \right)(y_0 - x_0).$$

## C  PROOF OF THEOREM 2

To prove this theorem, we consider three step-size ranges and do a case by case analysis for each of them. We construct functions for each range such that the convergence of any permutation-based

algorithm is "slow" for the functions on their corresponding step-size regime. The final lower bound is the minimum among the lower bounds obtained for the three regimes.

In this proof, we will use different notation from the rest of the paper because we work with scalars in the proof, and hence the superscript will denote the scalar power, and not the epoch number. We will use $\boldsymbol{x}_{k,i}$ to denote the $i$-th iterate of the $k$-th epoch.

We will construct three functions $F_1(\boldsymbol{x})$, $F_2(\boldsymbol{y})$, and $F_3(z)$, each the means of $n$ component functions, such that

- Any permutation-based algorithm on $F_1(\boldsymbol{x})$ with $\alpha \in [\frac{1}{2(n-1)KL}, \frac{3}{nL}]$ and initialization $\boldsymbol{x}_{1,0} = \boldsymbol{0}$ results in

$$\|\boldsymbol{x}_{K,n}\|^2 = \Omega\left(\frac{1}{n^3 K^2}\right).$$

  $F_1$ will be an $n$-dimensional function, that is $\boldsymbol{x} \in \mathbb{R}^n$. This function will have minimizer at $\boldsymbol{0} \in \mathbb{R}^n$. NOTE: This is the 'key' step-size range, and the proof sketch explained in the main paper corresponds to this function's construction.

- Any permutation-based algorithm on $F_2(\boldsymbol{y})$ with $\alpha \in [\frac{3}{nL}, \frac{1}{L}]$ and initialization $\boldsymbol{y}_{1,0} = \boldsymbol{0}$ results in

$$\|\boldsymbol{y}_{K,n}\|^2 = \Omega\left(\frac{1}{n^2}\right)$$

  $F_2$ will be an $n$-dimensional function, that is $\boldsymbol{y} \in \mathbb{R}^n$. This function will have minimizer at $\boldsymbol{0} \in \mathbb{R}^n$. The construction for this is also inspired by the construction for $F_1$, but this is constructed significantly differently due to the different step-size range.

- Any permutation-based algorithm on $F_3(z)$ with $\alpha \notin [\frac{1}{2(n-1)KL}, \frac{1}{L}]$ and initialization $z_{1,0} = 1$ results in

$$|z_{K,n}|^2 = \Omega(1)$$

  $F_3$ will be an 1-dimensional function, that is $z \in \mathbb{R}$. This function will have minimizer at $0$.

Then, the $2n + 1$-dimensional function $F([\boldsymbol{x}^\top, \boldsymbol{y}^\top, z]^\top) = F_1(\boldsymbol{x}) + F_2(\boldsymbol{y}) + F_3(z)$ will show bad convergence in any step-size regime. This function will have minimizer at $\boldsymbol{0} \in \mathbb{R}^{2n+1}$. Furthermore,

$$\frac{n-1}{n}L\boldsymbol{I} \preceq \nabla^2 F_1, \nabla^2 F_2, \nabla^2 F_3, \nabla^2 F \preceq 2L\boldsymbol{I},$$

that is, $F_1$, $F_2$, $F_3$, and $F$ are all $\frac{n-1}{n}L$-strongly convex and $2L$-smooth.

In the subsequent subsections, we prove the lower bounds for $F_1$, $F_2$, and $F_3$ separately.

**NOTE:** We note that above, we have used a specific initialization. However, in Appendix C.4, we discuss how the lower bound is actually invariant to initialization.

## C.1 Lower bound for $F_1$, $\alpha \in [\frac{1}{2(n-1)KL}, \frac{3}{nL}]$

We will work in $n$-dimensions ($n$ is even) and represent a vector in the space as $\boldsymbol{z} = [x_1, y_1, \ldots, x_{n/2}, y_{n/2}]$. These $x_i$ and $y_i$ are not related to the vectors used by $F_2$ and $F_3$ later, we only use $x_i$ and $y_i$ to make the proof for this part easier to understand.

We start off by defining the $n$ component functions: For $i \in [n/2]$, define

$$f_i(\boldsymbol{z}) := \frac{L}{2}x_i^2 - x_i + y_i + \sum_{j \neq i}\left(\frac{L}{2}x_j^2 + \frac{L}{2}y_j^2\right), \text{ and}$$

$$g_i(\boldsymbol{z}) := \frac{L}{2}y_i^2 - y_i + x_i + \sum_{j \neq i}\left(\frac{L}{2}x_j^2 + \frac{L}{2}y_j^2\right).$$

Thus, $F(\boldsymbol{z}) := \frac{1}{n}\left(\sum_{i=1}^{n/2} f_i + \sum_{i=1}^{n/2} g_i\right) = \left(\frac{n-1}{n}\right)\frac{L}{2}\|\boldsymbol{z}\|^2$. This function has minimizer at $\boldsymbol{z}^* = \boldsymbol{0}$.

Let $z_{k,j}$ denote $z$ at the $j$-th iteration in $k$-th epoch. We initialize at $z_{1,0} = 0$. For the majority of this proof, we will work inside a given epoch, so we will skip the subscript denoting the epoch. We denote the $j$-th iterate within the epoch as $z_j$ and the coordinates as $z_j = [x_{j,1}, y_{j,1}, \ldots, x_{j,n/2}, y_{j,n/2}]$

In the current epoch, let $\sigma$ be the permutation of $\{f_1, \ldots, f_{n/2}, g_1, \ldots, g_{n/2}\}$ used.

For any $i \in [n/2]$, let $p$ and $q$ be indices such that $\sigma_p = f_i$ and $\sigma_q = g_i$. Let us consider the case that $p < q$ (the case when $p > q$ will be analogous). Then, it can be seen that

$$x_{n,i} = (1 - \alpha L)^{n-1} x_{0,i} + (1 - \alpha L)^{n-p-1}\alpha - (1 - \alpha L)^{n-q}\alpha, \text{ and}$$
$$y_{n,i} = (1 - \alpha L)^{n-1} y_{0,i} - (1 - \alpha L)^{n-p}\alpha + (1 - \alpha L)^{n-q}\alpha. \tag{11}$$

Hence,

$$x_{n,i} + y_{n,i} = (1 - \alpha L)^{n-1}(x_{0,i} + y_{0,i}) + 2(1 - \alpha L)^{n-p-1}\alpha^2 L$$
$$\geq (1 - \alpha L)^{n-1}(x_{0,i} + y_{0,i}) + 2(1 - \alpha L)^{n-1}\alpha^2 L.$$

In the other case, when $p > q$, we will get the same inequality. Let $x_{K,n,i}$ and $y_{K,n,i}$ denote the value of $x_{n,i}$ and $y_{n,i}$ at the $K$-th epoch. Then, recalling that $z$ was initialized to $0$, we use the inequality above to get

$$x_{K,n,i} + y_{K,n,i} \geq (1 - \alpha L)^{(n-1)K} \cdot 0 + 2(1 - \alpha L)^{n-1}\alpha^2 L \frac{1 - (1 - \alpha L)^{(n-1)K}}{1 - (1 - \alpha L)^{n-1}}$$
$$= 2(1 - \alpha L)^{n-1}\alpha^2 L \frac{1 - (1 - \alpha L)^{(n-1)K}}{1 - (1 - \alpha L)^{n-1}} \tag{12}$$

Since this inequality is valid for all $i$, we get that

$$\|z_{K,n}\|^2 = \sum_{i=1}^{n/2}(x_{K,n,i}^2 + y_{K,n,i}^2)$$
$$\geq \sum_{i=1}^{n/2}\frac{1}{2}(x_{K,n,i} + y_{K,n,i})^2$$
$$\geq \sum_{i=1}^{n/2}\frac{1}{2}\left(2(1 - \alpha L)^{n-1}\alpha^2 L \frac{1 - (1 - \alpha L)^{(n-1)K}}{1 - (1 - \alpha L)^{n-1}}\right)^2$$
$$= n\left((1 - \alpha L)^{n-1}\alpha^2 L \frac{1 - (1 - \alpha L)^{(n-1)K}}{1 - (1 - \alpha L)^{n-1}}\right)^2. \tag{13}$$

Note that if $\alpha \leq \frac{3}{nL}$ and $n \geq 4$, then $(1 - \alpha L)^{n-1} \geq \frac{1}{8}$. Using this in (13), we get

$$\|z_{K,n}\|^2 \geq n\left((1 - \alpha L)^{n-1}\alpha^2 L \frac{1 - (1 - \alpha L)^{(n-1)K}}{1 - (1 - \alpha L)^{n-1}}\right)^2$$
$$\geq \frac{n}{128}\alpha^4 L^2\left(\frac{1 - (1 - \alpha L)^{(n-1)K}}{1 - (1 - \alpha L)^{n-1}}\right)^2$$
$$= \frac{n}{128 L^2}\left(\frac{(\alpha L)^2}{1 - (1 - \alpha L)^{n-1}}\right)^2\left(1 - (1 - \alpha L)^{(n-1)K}\right)^2$$

We consider two cases:

1. **Case A**, $\alpha L \leq \frac{1}{2(n+2)}$: It can be verified that $\frac{(\alpha L)^2}{1-(1-\alpha L)^{n-1}}$ is an increasing function of $\alpha$ when $\alpha L \leq \frac{1}{2(n+2)}$. Noting that we are working in the range when $\alpha \geq \frac{1}{2(n-1)KL}$, then

$$\|z_{K,n}\|^2 \geq \frac{n}{128 L^2}\left(\frac{\left(\frac{1}{2(n-1)K}\right)^2}{1 - \left(1 - \frac{1}{2(n-1)K}\right)^{n-1}}\right)^2\left(1 - (1 - \alpha L)^{(n-1)K}\right)^2$$

$$\geq \frac{n}{128L^2} \left( \frac{\left( \frac{1}{2(n-1)K} \right)^2}{\frac{1}{2(n-1)K}(n-1)} \right)^2 \left( 1 - (1-\alpha L)^{(n-1)K} \right)^2$$

$$= \frac{n}{512(n-1)^4 K^2 L^2} \left( 1 - (1-\alpha L)^{(n-1)K} \right)^2$$

$$\geq \frac{n}{512(n-1)^4 K^2 L^2} \left( 1 - e^{-\alpha L(n-1)K} \right)^2$$

$$\geq \frac{n}{512(n-1)^4 K^2 L^2} \left( 1 - e^{-1/2} \right)^2$$

$$= \Omega \left( \frac{1}{n^3 K^2 L^2} \right).$$

2. **Case B**, $\alpha L > \frac{1}{2(n+2)}$: In this case,

$$\|\boldsymbol{z}_{K,n}\|^2 \geq \frac{n}{128L^2} \left( \frac{\left( \frac{1}{2(n+2)} \right)^2}{1} \right)^2 \left( 1 - (1-\alpha L)^{(n-1)K} \right)^2$$

$$\geq \frac{n}{128L^2} \left( \frac{\left( \frac{1}{2(n+2)} \right)^2}{1} \right)^2 \left( 1 - e^{-\alpha L(n-1)K} \right)^2$$

$$\geq \frac{n}{128L^2} \left( \frac{\left( \frac{1}{2(n+2)} \right)^2}{1} \right)^2 \left( 1 - e^{-\frac{3(n-1)K}{2(n+2)}} \right)^2$$

$$= \Omega \left( \frac{1}{n^3 L^2} \right).$$

## C.2 LOWER BOUND FOR $F_2$, $\alpha \in [\frac{3}{nL}, \frac{1}{L}]$

We will work in $n$-dimensions and represent a vector in the space as $\boldsymbol{y} = [y_1, \ldots, y_n]$.

We start off by defining the $n$ component functions: For $i \in [n]$, define

$$f_i(\boldsymbol{y}) := -y_i + \sum_{j \neq i} \left( \frac{L y_j^2}{2} + \frac{y_j}{n-1} \right)$$

Thus, $F(\boldsymbol{y}) := \frac{1}{n} \sum_{i=1}^{n} f_i = \left( \frac{n-1}{n} \right) \frac{L}{2} \|\boldsymbol{y}\|^2$. This function has minimizer at $\boldsymbol{y}^* = \boldsymbol{0}$.

Let $\boldsymbol{y}_{k,j}$ denote $\boldsymbol{y}$ at the $j$-th iteration in $k$-th epoch. We initialize at $\boldsymbol{y}_{1,0} = \boldsymbol{0}$. For the majority of this proof, we will work inside a given epoch. We denote the $j$-th iterate within the epoch as $\boldsymbol{y}_j$ and the coordinates as $\boldsymbol{y}_j = [y_{j,1}, \ldots, y_{j,n}]$

In the current epoch, let $\sigma$ be the permutation of $\{f_1, \ldots, f_n\}$ used. Let $i$ be the index such that $\sigma_n = i$, that is, $i$ is the last element of the permutation $\sigma$. Then at the end of the epoch,

$$y_{n,i} = (1-\alpha L)^{n-1} y_{0,i} + \alpha - \frac{\alpha}{n-1} \sum_{j=0}^{n-2} (1-\alpha L)^j$$

$$= (1-\alpha L)^{n-1} y_{0,i} + \alpha - \frac{(1 - (1-\alpha L)^{n-1})}{(n-1)L}. \tag{14}$$

For some $j \in [n]$, let $s$ be the integer such that $\sigma_s = j$, that is $j$ is the $s$-th element in the permutation $\sigma$. Then for any $j$ and any epoch,

$$y_{n,j} = (1-\alpha L)^{n-1} y_{0,j} + \alpha (1-\alpha L)^{n-s} - \frac{\alpha}{n-1} \sum_{j=0}^{n-2} (1-\alpha L)^j.$$

Then,

$$y_{n,j} \geq (1 - \alpha L)^{n-1} y_{0,j} - \frac{\alpha}{n-1} \sum_{j=0}^{n-2} (1 - \alpha L)^j$$

$$= (1 - \alpha L)^{n-1} y_{0,j} - \frac{(1 - (1 - \alpha L)^{n-1})}{(n-1)L}.$$

Note that the above is independent of $\sigma$, and hence applicable for all epochs. Applying it recursively and noting that we initialized $\boldsymbol{y}_{1,0} = \boldsymbol{0}$, we get

$$y_{n,j} \geq -\frac{(1 - (1 - \alpha L)^{n-1})}{(n-1)L} \sum_{k=1}^{K} (1 - \alpha L)^n$$

$$\geq -\frac{1}{(n-1)L}.$$

Note that $y_{0,i}$ is just $y_{n,i}$ from the previous epoch. Hence we can substitute the inequality above for $y_{0,i}$ in (14). Thus,

$$y_{n,i} \geq -\frac{(1 - \alpha L)^{n-1}}{(n-1)L} + \alpha - \frac{(1 - (1 - \alpha L)^{n-1})}{(n-1)L}$$

$$= \alpha - \frac{1}{(n-1)L}$$

$$\geq \frac{3}{nL} - \frac{1}{(n-1)L}$$

$$= \Omega\left(\frac{1}{nL}\right).$$

This gives us that $\|\boldsymbol{y}_{k,n}\|^2 = \Omega\left(\frac{1}{n^2 L^2}\right)$ for any $k$.

## C.3 Lower bound for $F_3$, $\alpha \notin \left[\frac{1}{2(n-1)KL}, \frac{1}{L}\right]$

Consider the function $F(z) = \frac{1}{n} \sum_{i=1}^{n} f_i(z)$, where $f_i(z) = Lz^2$ for all $i$. Note that the gradient of any function at $z$ is just $-2Lz$. Hence, regardless of the permutation, if we start the shuffling based gradient descent method at $z_{1,0} = 1$, we get that

$$z_{K,n} = (1 - 2\alpha L)^{nK} z_{1,0} = (1 - 2\alpha L)^{nK}.$$

In the case when $\alpha \leq \frac{1}{2(n-1)KL}$, we see that

$$z_{K,n} \geq \left(1 - 2\frac{1}{2(n-1)KL}L\right)^{nK}$$

$$\geq \left(1 - \frac{1}{(n-1)K}\right)^{nK}$$

$$= \Omega(1),$$

for $n, K \geq 2$. Finally, in the case when $\alpha \geq \frac{1}{L}$, we see that

$$|z_{K,n}| = |1 - 2\alpha L|^{nK}$$

$$\geq \left|1 - 2\frac{1}{L}L\right|^{nK}$$

$$\geq 1^{nK}$$

$$= \Omega(1).$$

### C.4 DISCUSSION ABOUT INITIALIZATION

The lower bound partitions the step size in three ranges -

- In the step size ranges $\alpha \in [\frac{1}{2(n-1)KL}, \frac{3}{nL}]$ and $\alpha \in [\frac{3}{nL}, \frac{1}{L}]$, the initializations are done at the minimizer and it is shown that any permutation-based algorithm will still move away from the minimizer. The choice of initializing at the minimizer was solely for the convenience of analysis and calculations, and the proof should work for any other initialization as well.

  Furthermore, the effect of initializing at any arbitrary (non-zero) point will decay exponentially fast with epochs anyway. To see how, note that every epoch can be treated as $n$ steps of full gradient descent and some noise, and hence the full gradient descent part will essentially keep decreasing the effect of initialization exponentially, and what we would be left with is the noise in each epoch. Thus, it was more convenient for us to just assume initialization at the minimizer and only focus on the noise in each epoch.

- The step size range $\alpha \notin [\frac{1}{2(n-1)KL}, \frac{1}{L}]$ can be divided into two parts, $\alpha \in [0, \frac{1}{2(n-1)KL}]$ and $\alpha \in [\frac{1}{L}, \infty)$.

  For the range $\alpha \in [0, \frac{1}{2(n-1)KL}]$, we essentially show that the step size is too small to make any meaningful progress towards the minimizer. Hence, instead of initializing at 1, initializing at any other arbitrary (non-zero) point will also give the same slow convergence rate.

  For the range $\alpha \in [\frac{1}{L}, \infty)$, we show that the optimization algorithm will in fact diverge since the step size is too large. Hence, even here, any other arbitrary (non-zero) point of initialize will also give divergence.

## D PROOF OF THEOREM 3

Define $f_1(x) := \frac{L}{2}x^2 - x$ and $f_2(x, y) := -\frac{L}{4}x^2 + x$. Thus, $F(x, y) = \frac{L}{8}x^2$. This function has minimizer at $x^* = 0$. For this proof, we will use the convention that $x_{i,j}$ is the iterate after the $i$-th iteration of the $j$-th epoch. Further, the number in the superscript will be the scalar power. For example $x_{i,j}^2 = x_{i,j} \cdot x_{i,j}$.

Initialize at $x_{0,0} = \frac{1}{L}$. Then at epoch $k$, there are two possible permutations: $\sigma = (1, 2)$ and $\sigma = (2, 1)$. In the first case , $\sigma = (1, 2)$, we get that after the first iteration of the epoch,

$$x_{1,k} = x_{0,k} - \alpha \nabla f_1(x_{0,k}, y_{0,k})$$
$$= (1 - \alpha L)x_{0,k} + \alpha,$$

Continuing on, in the second iteration, we get

$$x_{2,k} = x_{1,k} - \alpha \nabla f_2(x_{1,k}, y_{1,k})$$
$$= \left(1 + \frac{1}{2}\alpha L\right) x_{1,k} - \alpha$$
$$= \left(1 + \frac{1}{2}\alpha L\right) ((1 - \alpha L)x_{0,k} + \alpha) - \alpha$$
$$= \left(1 + \frac{1}{2}\alpha L\right) (1 - \alpha L)x_{0,k} + \frac{1}{2}\alpha^2 L.$$

Note that $x_{0,k+1} = x_{2,k}$. Thus, $x_{0,k+1} = \left(1 + \frac{1}{2}\alpha L\right) (1 - \alpha L)x_{0,k} + \frac{1}{2}\alpha^2 L$.

Similarly, for the other possible permutation, $\sigma = (2, 1)$, we get $x_{0,k+1} = \left(1 + \frac{1}{2}\alpha L\right) (1 - \alpha L)x_{0,k} + \alpha^2 L$. Thus, regardless of what permutation we use, we get that

$$x_{0,k+1} \geq \left(1 + \frac{1}{2}\alpha L\right) (1 - \alpha L)x_{0,k} + \frac{1}{2}\alpha^2 L \geq (1 - \alpha L)x_{0,k} + \frac{1}{2}\alpha^2 L.$$

Hence, recalling that we initialized at $x_{0,0} = \frac{1}{L}$, we get

$$x_{n,K} = x_{0,K+1}$$

$$\geq (1 - \alpha L)^K \frac{1}{L} + \frac{1}{2} \alpha^2 L \sum_{i=0}^{K-1} (1 - \alpha L)^i$$

$$= \frac{1}{L} (1 - \alpha L)^K + \frac{1}{2} \alpha^2 L \frac{1 - (1 - \alpha L)^K}{1 - (1 - \alpha L)}$$

$$= \frac{1}{L} (1 - \alpha L)^K + \frac{1}{2} \alpha \left( 1 - (1 - \alpha L)^K \right). \tag{15}$$

Now, if $\alpha \geq \frac{1}{KL}$, then

$$\frac{1}{2} \alpha \left( 1 - (1 - \alpha L)^K \right) \geq \frac{\alpha}{8} = \Omega \left( \frac{1}{KL} \right),$$

and hence, $x_{n,K}^2 = \Omega(\frac{1}{K^2 L^2})$. Otherwise, if $\alpha \leq \frac{1}{KL}$, then continuing on from (15),

$$x_{n,K} \geq \frac{1}{L} (1 - \alpha L)^K + \frac{1}{2} \alpha \left( 1 - (1 - \alpha L)^K \right)$$

$$\geq \frac{1}{L} (1 - \alpha L)^K$$

$$\geq \frac{1}{L} e^{-\alpha L K}$$

$$\geq \frac{1}{L} e^{-1}$$

$$= \Omega(1/L),$$

and hence in this case, $x_{n,K}^2 = \Omega(\frac{1}{L^2})$.

## E    PROOF OF THEOREM 4

Let $F(\boldsymbol{x}) := \frac{1}{n} \sum_{i=1}^{n} f_i(\boldsymbol{x})$ be such that its minimizer it at the origin. This can be assumed without loss of generality because we can shift the coordinates appropriately, similar to (Safran & Shamir, 2019). Since the $f_i$ are convex quadratics, we can write them as $f_i(\boldsymbol{x}) = \frac{1}{2} \boldsymbol{x}^\top \boldsymbol{A}_i \boldsymbol{x} - \boldsymbol{b}_i^\top \boldsymbol{x} + c_i$, where $\boldsymbol{A}_i$ are symmetric, positive-semidefinite matrices. We can omit the constants $c_i$ because they do not affect the minimizer or the gradients. Because we assume that the minimizer of $F(\boldsymbol{x})$ is at the origin, we get that

$$\sum_{i=1}^{n} \boldsymbol{b}_i = \boldsymbol{0}. \tag{16}$$

Let $\sigma = (\sigma_1, \sigma_2, \ldots, \sigma_n)$ be the random permutation of $(1, 2, \ldots, n)$ generated at the beginning of the algorithm. Then for $k \in (1, 2, \ldots, K/2)$, epoch $2k - 1$ sees the $n$ functions in the following sequence:

$$\left( \frac{1}{2} \boldsymbol{x}^\top \boldsymbol{A}_{\sigma_1} \boldsymbol{x} - \boldsymbol{b}_{\sigma_1}^\top \boldsymbol{x}, \frac{1}{2} \boldsymbol{x}^\top \boldsymbol{A}_{\sigma_2} \boldsymbol{x} - \boldsymbol{b}_{\sigma_2}^\top \boldsymbol{x}, \ldots, \frac{1}{2} \boldsymbol{x}^\top \boldsymbol{A}_{\sigma_n} \boldsymbol{x} - \boldsymbol{b}_{\sigma_n}^\top \boldsymbol{x} \right),$$

whereas epoch $2k$ sees the $n$ functions in the reverse sequence:

$$\left( \frac{1}{2} \boldsymbol{x}^\top \boldsymbol{A}_{\sigma_n} \boldsymbol{x} - \boldsymbol{b}_{\sigma_n}^\top \boldsymbol{x}, \frac{1}{2} \boldsymbol{x}^\top \boldsymbol{A}_{\sigma_{n-1}} \boldsymbol{x} - \boldsymbol{b}_{\sigma_{n-1}}^\top \boldsymbol{x}, \ldots, \frac{1}{2} \boldsymbol{x}^\top \boldsymbol{A}_{\sigma_1} \boldsymbol{x} - \boldsymbol{b}_{\sigma_1}^\top \boldsymbol{x} \right).$$

We define $\boldsymbol{S}_i := \alpha \boldsymbol{A}_{\sigma_i}$ and $\boldsymbol{t}_i = \alpha \boldsymbol{b}_{\sigma_i}$ for convenience of notation. We start off by computing the progress made during an even indexed epoch. Since the even epochs use the reverse permutation, we get

$$\boldsymbol{x}_0^{2k+1} = \boldsymbol{x}_n^{2k}$$

$$= \boldsymbol{x}_{n-1}^{2k} - \alpha \left( \boldsymbol{A}_{\sigma_1} \boldsymbol{x}_{n-1}^{2k} - \boldsymbol{b}_{\sigma_1} \right) \qquad (f_{\sigma_1} \text{ is used at the last iteration of even epochs.})$$

$$= \left( \boldsymbol{I} - \alpha \boldsymbol{A}_{\sigma_1} \right) \boldsymbol{x}_{n-1}^{2k} + \alpha \boldsymbol{b}_{\sigma_1}$$

$$= (I - S_1)x_{n-1}^{2k} + t_1.$$

We recursively apply the same procedure as above to the whole epoch to get the following

$$
\begin{aligned}
x_0^{2k+1} &= (I - S_1)x_{n-1}^{2k} + t_1 \\
&= (I - S_1)\left((I - S_2)x_{n-2}^{2k} + t_2\right) + t_1 \\
&= (I - S_1)(I - S_2)x_{n-2}^{2k} + (I - S_1)t_2 + t_1 \\
&\vdots \\
&= \left(\prod_{i=1}^{n}(I - S_i)\right)x_0^{2k} + \sum_{i=1}^{n}\left(\prod_{j=1}^{n-i}(I - S_j)\right)t_{n+1-i},
\end{aligned} \tag{17}
$$

where the product of matrices $\{M_i\}$ is defined as $\prod_{i=l}^{m} M_i = M_l M_{l+1} \ldots M_m$ if $m \geq l$ and 1 otherwise. Similar to Eq. (17), we can compute the progress made during an odd indexed epoch. Recall that the only difference will be that the odd indexed epochs see the permutations in the order $(\sigma_1, \sigma_2, \ldots, \sigma_n)$ instead of $(\sigma_n, \sigma_{n-1}, \ldots, \sigma_1)$. After doing the computation, we get the following equation:

$$
x_0^{2k} = \left(\prod_{i=1}^{n}(I - S_{n-i+1})\right)x_0^{2k-1} + \sum_{i=1}^{n}\left(\prod_{j=1}^{n-i}(I - S_{n-j+1})\right)t_i.
$$

Combining the results above, we can get the total progress made after the pair of epoch $2k - 1$ and $2k$:

$$
\begin{aligned}
x_0^{2k+1} &= \left(\prod_{i=1}^{n}(I - S_i)\right)x_0^{2k} + \sum_{i=1}^{n}\left(\prod_{j=1}^{n-i}(I - S_j)\right)t_{n+1-i} \\
&= \left(\prod_{i=1}^{n}(I - S_i)\right)\left(\prod_{i=1}^{n}(I - S_{n-i+1})\right)x_0^{2k-1} + \left(\prod_{j=1}^{n}(I - S_j)\right)\sum_{i=1}^{n}\left(\prod_{j=1}^{n-i}(I - S_{n+1-j})\right)t_i \\
&\quad + \sum_{i=1}^{n}\left(\prod_{j=1}^{n-i}(I - S_j)\right)t_{n+1-i}.
\end{aligned} \tag{18}
$$

In the sum above, the first term will have an exponential decay, hence we need to control the next two terms. We denote the sum of the terms as $z$ (see the definition below) and we will control its norm later in this proof.

$$
\begin{aligned}
z &:= \left(\prod_{j=1}^{n}(I - S_j)\right)\sum_{i=1}^{n}\left(\prod_{j=1}^{n-i}(I - S_{n+1-j})\right)t_i + \sum_{i=1}^{n}\left(\prod_{j=1}^{n-i}(I - S_j)\right)t_{n+1-i} \\
&= \sum_{i=1}^{n}\left(\prod_{j=1}^{n}(I - S_j)\right)\left(\prod_{j=1}^{n-i}(I - S_{n+1-j})\right)t_i + \sum_{i=1}^{n}\left(\prod_{j=1}^{n-i}(I - S_j)\right)t_{n+1-i}.
\end{aligned}
$$

To see where the iterates end up after $K$ epochs, we simply set $2k = K$ in Eq. 18 and then keep applying the equation recursively to preceding epochs. Then, we get

$$
\begin{aligned}
x_n^K = x_0^{K+1} &= \left(\prod_{i=1}^{n}(I - S_i)\right)\left(\prod_{i=1}^{n}(I - S_{n-i+1})\right)x_0^{K-1} + z \\
&= \left(\left(\prod_{i=1}^{n}(I - S_i)\right)\left(\prod_{i=1}^{n}(I - S_{n-i+1})\right)\right)^2 x_0^{K-3} + \left(\prod_{i=1}^{n}(I - S_i)\right)\left(\prod_{i=1}^{n}(I - S_{n-i+1})\right)z + z \\
&\vdots
\end{aligned}
$$

$$= \left( \left( \prod_{i=1}^{n} (\boldsymbol{I} - \boldsymbol{S}_i) \right) \left( \prod_{i=1}^{n} (\boldsymbol{I} - \boldsymbol{S}_{n-i+1}) \right) \right)^{K/2} \boldsymbol{x}_0^1 + \sum_{k=0}^{\frac{K}{2}-1} \left( \left( \prod_{i=1}^{n} (\boldsymbol{I} - \boldsymbol{S}_i) \right) \left( \prod_{i=1}^{n} (\boldsymbol{I} - \boldsymbol{S}_{n-i+1}) \right) \right)^{k} \boldsymbol{z}.$$

Taking squared norms and expectations on both sides, we get

$$\mathbb{E}[\|\boldsymbol{x}_n^K\|^2] = \mathbb{E}\left[ \left\| \left( \left( \prod_{i=1}^{n} (\boldsymbol{I} - \boldsymbol{S}_i) \right) \left( \prod_{i=1}^{n} (\boldsymbol{I} - \boldsymbol{S}_{n-i+1}) \right) \right)^{K/2} \boldsymbol{x}_0^1 \right. \right.$$
$$\left. \left. + \sum_{k=0}^{\frac{K}{2}-1} \left( \left( \prod_{i=1}^{n} (\boldsymbol{I} - \boldsymbol{S}_i) \right) \left( \prod_{i=1}^{n} (\boldsymbol{I} - \boldsymbol{S}_{n-i+1}) \right) \right)^{k} \boldsymbol{z} \right\|^2 \right]$$

$$\leq 2\mathbb{E}\left[ \left\| \left( \left( \prod_{i=1}^{n} (\boldsymbol{I} - \boldsymbol{S}_i) \right) \left( \prod_{i=1}^{n} (\boldsymbol{I} - \boldsymbol{S}_{n-i+1}) \right) \right)^{K/2} \boldsymbol{x}_0^1 \right\|^2 \right]$$
$$+ 2\mathbb{E}\left[ \left\| \sum_{k=0}^{\frac{K}{2}-1} \left( \left( \prod_{i=1}^{n} (\boldsymbol{I} - \boldsymbol{S}_i) \right) \left( \prod_{i=1}^{n} (\boldsymbol{I} - \boldsymbol{S}_{n-i+1}) \right) \right)^{k} \boldsymbol{z} \right\|^2 \right]$$
$$\text{(Since } (a+b)^2 \leq 2a^2 + 2b^2)$$

$$\leq 2\mathbb{E}\left[ \left\| \left( \left( \prod_{i=1}^{n} (\boldsymbol{I} - \boldsymbol{S}_i) \right) \left( \prod_{i=1}^{n} (\boldsymbol{I} - \boldsymbol{S}_{n-i+1}) \right) \right)^{K/2} \boldsymbol{x}_0^1 \right\|^2 \right]$$
$$+ 2\mathbb{E}\left[ \left( \|\boldsymbol{z}\| \sum_{k=0}^{\frac{K}{2}-1} \left\| \prod_{i=1}^{n} (\boldsymbol{I} - \boldsymbol{S}_i) \prod_{i=1}^{n} (\boldsymbol{I} - \boldsymbol{S}_{n-i+1}) \right\|^{k} \right)^2 \right].$$

We assumed that the functions $f_i$ have $L$-Lipschitz gradients (Assumption 2). This translates to $\boldsymbol{A}_i$ having maximum eigenvalue less than $L$. Hence, if $\alpha \leq 1/L$, we get that $\boldsymbol{I} - \alpha \boldsymbol{A}_i$ is positive semi-definite with maximum eigenvalue bounded by 1. Hence, $\|\boldsymbol{I} - \boldsymbol{S}_i\| \leq 1$. Using this and the fact that for matrices $\boldsymbol{M}_1$ and $\boldsymbol{M}_2$, $\|\boldsymbol{M}_1 \boldsymbol{M}_2\| \leq \|\boldsymbol{M}_1\|\|\boldsymbol{M}_2\|$, we get that

$$\mathbb{E}\left[\|\boldsymbol{x}_n^K\|^2\right] \leq 2\mathbb{E}\left[ \left\| \left( \left( \prod_{i=1}^{n} (\boldsymbol{I} - \boldsymbol{S}_i) \right) \left( \prod_{i=1}^{n} (\boldsymbol{I} - \boldsymbol{S}_{n-i+1}) \right) \right)^{K/2} \boldsymbol{x}_0^1 \right\|^2 \right]$$
$$+ 2\mathbb{E}\left[ \left( \|\boldsymbol{z}\| \sum_{k=0}^{\frac{K}{2}-1} \left( \prod_{i=1}^{n} \|\boldsymbol{I} - \boldsymbol{S}_i\| \prod_{i=1}^{n} \|\boldsymbol{I} - \boldsymbol{S}_{n-i+1}\| \right)^{k} \right)^2 \right]$$

$$\leq 2\mathbb{E}\left[ \left\| \left( \left( \prod_{i=1}^{n} (\boldsymbol{I} - \boldsymbol{S}_i) \right) \left( \prod_{i=1}^{n} (\boldsymbol{I} - \boldsymbol{S}_{n-i+1}) \right) \right)^{K/2} \boldsymbol{x}_0^1 \right\|^2 \right] + 2\mathbb{E}\left[ \left( \|\boldsymbol{z}\| \sum_{k=0}^{\frac{K}{2}-1} 1 \right)^2 \right]$$

$$= 2\mathbb{E}\left[ \left\| \left( \left( \prod_{i=1}^{n} (\boldsymbol{I} - \boldsymbol{S}_i) \right) \left( \prod_{i=1}^{n} (\boldsymbol{I} - \boldsymbol{S}_{n-i+1}) \right) \right)^{K/2} \boldsymbol{x}_0^1 \right\|^2 \right] + \frac{K^2}{2} \mathbb{E}\left[ \|\boldsymbol{z}\|^2 \right].$$

We handle the two terms above separately. For the first term, we have the following bound:

**Lemma 3.** *If* $\alpha \leq \frac{1}{8\kappa L} \min\left\{2, \frac{\sqrt{\kappa}}{n}\right\}$, *then*

$$\left\| \prod_{i=1}^{n} (\boldsymbol{I} - \boldsymbol{S}_i) \prod_{i=1}^{n} (\boldsymbol{I} - \boldsymbol{S}_{n-i+1}) \right\| \leq 1 - \alpha n \mu$$

Note that $K \geq 80\kappa^{3/2} \log(nK) \max\left\{1, \frac{\sqrt{\kappa}}{n}\right\} \implies \alpha \leq \frac{1}{8\kappa L} \min\left\{2, \frac{\sqrt{\kappa}}{n}\right\}$.

We also have the following lemma that bounds the expected squared norm of $\boldsymbol{z}$.

**Lemma 4.** *If $\alpha \leq \frac{1}{L}$, then*

$$\mathbb{E}\left[\|\boldsymbol{z}\|^2\right] \leq 2n^2\alpha^4 L^2(G^*)^2 + 170n^5\alpha^6 L^4 G^2 \log n,$$

*where $G^* = \max_i \|\boldsymbol{b}_i\|$, and the expectation is taken over the randomness of $\boldsymbol{z}$.*

Using these lemmas, we get that

$$\mathbb{E}\left[\left\|\boldsymbol{x}_n^K\right\|^2\right] \leq 2\left(1 - n\alpha\mu\right)^{K/2}\left\|\boldsymbol{x}_0^1\right\|^2 + K^2 n^2 \alpha^4 L^2 G^2 + 85 K^2 n^5 \alpha^6 L^4 G^2 \log n$$

$$\leq 2e^{-\frac{1}{2}n\alpha\mu K}\left\|\boldsymbol{x}_0^1\right\|^2 + K^2 n^2 \alpha^4 L^2 G^2 + 85 K^2 n^5 \alpha^6 L^4 G^2 \log n.$$

Substituting $\alpha = \frac{10 \log nK}{\mu nK}$ gives us the result.

### E.1    PROOF OF LEMMA 3

We define $(\widetilde{\boldsymbol{S}}_1, \ldots, \widetilde{\boldsymbol{S}}_n) := (\boldsymbol{S}_1, \ldots, \boldsymbol{S}_n)$ and $(\widetilde{\boldsymbol{S}}_{n+1}, \ldots, \widetilde{\boldsymbol{S}}_{2n}) := (\boldsymbol{S}_n, \ldots, \boldsymbol{S}_1)$. Then,

$$\left\|\prod_{i=1}^n (\boldsymbol{I} - \boldsymbol{S}_i)\prod_{i=1}^n (\boldsymbol{I} - \boldsymbol{S}_{n-i+1})\right\| = \left\|\prod_{i=1}^{2n}(\boldsymbol{I} - \widetilde{\boldsymbol{S}}_i)\right\|$$

$$= \left\|\boldsymbol{I} - \sum_{i=1}^{2n}\widetilde{\boldsymbol{S}}_i + \sum_{i<j}\widetilde{\boldsymbol{S}}_i\widetilde{\boldsymbol{S}}_j - \ldots\right\|$$

$$\leq \left\|\boldsymbol{I} - \sum_{i=1}^{2n}\widetilde{\boldsymbol{S}}_i + \sum_{i<j}\widetilde{\boldsymbol{S}}_i\widetilde{\boldsymbol{S}}_j\right\| + \left\|\sum_{i<j<k}\widetilde{\boldsymbol{S}}_i\widetilde{\boldsymbol{S}}_j\widetilde{\boldsymbol{S}}_k\right\| + \ldots$$

$$\leq \left\|\boldsymbol{I} - \sum_{i=1}^{2n}\widetilde{\boldsymbol{S}}_i + \sum_{i<j}\widetilde{\boldsymbol{S}}_i\widetilde{\boldsymbol{S}}_j\right\| + \sum_{i<j<k}\left\|\widetilde{\boldsymbol{S}}_i\right\|\left\|\widetilde{\boldsymbol{S}}_j\right\|\left\|\widetilde{\boldsymbol{S}}_k\right\| + \ldots.$$

Note that

$$\sum_{i<j}\widetilde{\boldsymbol{S}}_i\widetilde{\boldsymbol{S}}_j = 2\sum_{i\neq j}\boldsymbol{S}_i\boldsymbol{S}_j + \sum_{i=1}^n \boldsymbol{S}_i^2$$

$$= 2\left(\sum_{i=1}^n \boldsymbol{S}_i\right)\left(\sum_{i=1}^n \boldsymbol{S}_i\right) - \sum_{i=1}^n \boldsymbol{S}_i^2.$$

Substituting this and noting that $\sum_{i=1}^{2n}\widetilde{\boldsymbol{S}}_i = 2\sum_{i=1}^n \boldsymbol{S}_i$, we get

$$\left\|\prod_{i=1}^n (\boldsymbol{I} - \boldsymbol{S}_i)\prod_{i=1}^n (\boldsymbol{I} - \boldsymbol{S}_{n-i+1})\right\| \leq \left\|\boldsymbol{I} - 2\sum_{i=1}^n \boldsymbol{S}_i + 2\left(\sum_{i=1}^n \boldsymbol{S}_i\right)\left(\sum_{i=1}^n \boldsymbol{S}_i\right) - \sum_{i=1}^n \boldsymbol{S}_i^2\right\|$$

$$+ \sum_{i<j<k}\left\|\widetilde{\boldsymbol{S}}_i\right\|\left\|\widetilde{\boldsymbol{S}}_j\right\|\left\|\widetilde{\boldsymbol{S}}_k\right\| + \ldots$$

$$\leq \left\|\boldsymbol{I} - 2\sum_{i=1}^n \boldsymbol{S}_i + 2\left(\sum_{i=1}^n \boldsymbol{S}_i\right)\left(\sum_{i=1}^n \boldsymbol{S}_i\right)\right\| + \left\|\sum_{i=1}^n \boldsymbol{S}_i^2\right\|$$

$$+ \sum_{i<j<k}\left\|\widetilde{\boldsymbol{S}}_i\right\|\left\|\widetilde{\boldsymbol{S}}_j\right\|\left\|\widetilde{\boldsymbol{S}}_k\right\| + \ldots.$$

Let us denote $T := \sum_{i=1}^{n} S_i$. Then, we know by Assumptions 2 and 3 that $T$ has eigenvalues in $[n\alpha\mu, n\alpha L]$. Then, as long as $\alpha \leq \frac{1}{4nL}$, we get that

$$\left\| I - 2\sum_{i=1}^{n} S_i + 2\left(\sum_{i=1}^{n} S_i\right)\left(\sum_{i=1}^{n} S_i\right) \right\| = \|I - 2T + 2T^2\|$$

$$\leq 1 - \frac{3}{2}n\alpha\mu.$$

Substituting this, we get

$$\left\| \prod_{i=1}^{n}(I - S_i) \prod_{i=1}^{n}(I - S_{n-i+1}) \right\| \leq \left(1 - \frac{3}{2}n\alpha\mu\right) + \left\| \sum_{i=1}^{n} S_i^2 \right\| + \sum_{i<j<k} \left\|\widetilde{S}_i\right\| \left\|\widetilde{S}_j\right\| \left\|\widetilde{S}_k\right\| + \dots.$$

By Assumption 2, we know that $\|A_i\| \leq L$. Hence, $\|\widetilde{\mathbf{S}}_i\| \leq \alpha L$. Hence,

$$\left\| \prod_{i=1}^{n}(I - S_i) \prod_{i=1}^{n}(I - S_{n-i+1}) \right\| \leq \left(1 - \frac{3}{2}n\alpha\mu\right) + n\alpha^2 L^2 + \left(\binom{2n}{3}\alpha^3 L^3 + \binom{2n}{4}\alpha^4 L^4 + \dots\right)$$

$$\leq 1 - \frac{3}{2}n\alpha\mu + n\alpha^2 L^2 + \sum_{i=3}^{2n}(2n\alpha L)^i$$

$$\leq 1 - \frac{3}{2}n\alpha\mu + n\alpha^2 L^2 + \frac{8n^3\alpha^3 L^3}{1 - 2n\alpha L}$$

$$\leq 1 - \frac{3}{2}n\alpha\mu + n\alpha^2 L^2 + 16n^3\alpha^3 L^3. \qquad \text{(Since } \alpha \leq 1/4nL.\text{)}$$

Finally, as long as $\alpha \leq \frac{1}{4\kappa L}$ and $\alpha \leq \frac{1}{8nL\sqrt{\kappa}}$,

$$\left\| \prod_{i=1}^{n}(I - S_i) \prod_{i=1}^{n}(I - S_{n-i+1}) \right\| \leq 1 - n\alpha\mu.$$

### E.2 PROOF OF LEMMA 4

We start off by computing the first order expansion of $z$. We have the following lemma for this:

**Lemma 5.**

$$z = \sum_{i=1}^{n} S_i t_i + \sum_{j=n+1}^{2n-1} \sum_{l=1}^{j-1} \left(\prod_{p=1}^{l-1}(I - \widetilde{S}_p)\right) \widetilde{S}_l \widetilde{S}_j \left(\sum_{i=1}^{2n-j} t_i\right) + \sum_{j=1}^{n-1} \sum_{l=1}^{j-1} \left(\prod_{p=1}^{l-1}(I - S_p)\right) S_l S_j \left(\sum_{i=1}^{n-j} t_{n+1-i}\right),$$

where $(\widetilde{S}_1, \dots, \widetilde{S}_n) := (S_1, \dots, S_n)$ and $(\widetilde{S}_{n+1}, \dots, \widetilde{S}_{2n}) := (S_n, \dots, S_1)$.

The proof of this lemma is quite algebraic and hence has been pushed to the end, in Appendix E.4.

The strategy is to bound $\|S_i t_i\|$, $\|\sum_{i=1}^{2n-j} t_i\|$, and $\|\sum_{i=1}^{n-j} t_{n+1-i}\|$. Hence, we apply Lemma 5 and use triangle inequality:

$$\mathbb{E}\left[\|z\|^2\right] = \mathbb{E}\left[\left\| \sum_{i=1}^{n} S_i t_i + \sum_{j=n+1}^{2n-1} \sum_{k=1}^{j-1} \left(\prod_{p=1}^{l-1}(I - \widetilde{S}_p)\right) \widetilde{S}_l \widetilde{S}_j \left(\sum_{i=1}^{2n-j} t_i\right)\right.\right.$$

$$\left.\left. + \sum_{j=1}^{n-1} \sum_{l=1}^{j-1} \left(\prod_{p=1}^{l-1}(I - S_p)\right) S_l S_j \left(\sum_{i=1}^{n-j} t_{n+1-i}\right) \right\|^2\right]$$

$$\leq \mathbb{E}\left[\left(\sum_{i=1}^{n} \|S_i\|\|t_i\| + \sum_{j=n+1}^{2n-1} \sum_{l=1}^{j-1} \left(\prod_{p=1}^{l-1}\|I - \widetilde{S}_p\|\right) \|\widetilde{S}_l\|\|\widetilde{S}_j\| \left(\sum_{i=1}^{2n-j} \|t_i\|\right)\right.\right.$$

$$+ \sum_{j=1}^{n-1} \sum_{l=1}^{j-1} \left( \prod_{p=1}^{l-1} \|\boldsymbol{I} - \boldsymbol{S}_p\| \right) \|\boldsymbol{S}_l\| \|\boldsymbol{S}_j\| \left( \sum_{i=1}^{n-j} \|\boldsymbol{t}_{n+1-i}\| \right) \right)^2 \right].$$

Now, we recall that $\|\boldsymbol{S}_i\| \leq \alpha L$ and $\|\boldsymbol{t}_i\| \leq \alpha G$. Because $\alpha \leq 1/L$, we also get that $\|\boldsymbol{I} - \boldsymbol{S}_i\| \leq 1$. Using these,

$$\mathbb{E}[\|\boldsymbol{z}\|^2] \leq \mathbb{E}\left[ \left( \sum_{i=1}^{n} \alpha^2 LG + \sum_{j=n+1}^{2n-1} \sum_{l=1}^{j-1} \left( \prod_{p=1}^{l-1} 1 \right) \alpha L \alpha L \left\| \sum_{i=1}^{2n-j} \boldsymbol{t}_i \right\| \right.\right.$$
$$\left.\left. + \sum_{j=1}^{n-1} \sum_{l=1}^{j-1} \left( \prod_{p=1}^{l-1} 1 \right) \alpha L \alpha L \left\| \sum_{i=1}^{n-j} \boldsymbol{t}_{n+1-i} \right\| \right)^2 \right]$$

$$\leq \mathbb{E}\left[ \left( n\alpha^2 LG + 2n\alpha^2 L^2 \sum_{j=n+1}^{2n-1} \left\| \sum_{i=1}^{2n-j} \boldsymbol{t}_i \right\| + n\alpha^2 L^2 \sum_{j=1}^{n-1} \left\| \sum_{i=1}^{n-j} \boldsymbol{t}_{n+1-i} \right\| \right)^2 \right]$$

$$= n^2 \alpha^4 L^2 \mathbb{E}\left[ \left( G + 2L \sum_{j=n+1}^{2n-1} \left\| \sum_{i=1}^{2n-j} \boldsymbol{t}_i \right\| + L \sum_{j=1}^{n-1} \left\| \sum_{i=1}^{n-j} \boldsymbol{t}_{n+1-i} \right\| \right)^2 \right]$$

$$\leq 2n^2 \alpha^4 L^2 \left( G^2 + L^2 \mathbb{E}\left[ \left( 2 \sum_{j=n+1}^{2n-1} \left\| \sum_{i=1}^{2n-j} \boldsymbol{t}_i \right\| + \sum_{j=1}^{n-1} \left\| \sum_{i=1}^{n-j} \boldsymbol{t}_{n+1-i} \right\| \right)^2 \right] \right).$$
$$\text{(Since } (a+b)^2 \leq 2a^2 + 2b^2)$$

Using Hoeffding-Serfling inequality for bounded random vectors (Schneider, 2016, Theorem 2), we get the following lemma

**Lemma 6.** *For all $j, l \in [1, n]$ we have that*

$$\mathbb{E}\left[ \left\| \sum_{i=1}^{j} \boldsymbol{t}_i \right\|^2 \right] \leq 18 j \alpha^2 (G^*)^2 \log(n)$$

$$\mathbb{E}\left[ \left\| \sum_{i=1}^{j} \boldsymbol{t}_i \right\| \left\| \sum_{i=1}^{l} \boldsymbol{t}_i \right\| \right] \leq 18 \sqrt{jl} \alpha^2 (G^*)^2 \log(n),$$

*where $G^* = \max_i \|\boldsymbol{b}_i\|$, and the expectation is taken over the randomness of $\boldsymbol{t}_i$.*

Writing out the expansion of $\left( 2\sum_{j=n+1}^{2n-1} \left\| \sum_{i=1}^{2n-j} \boldsymbol{t}_i \right\| + \sum_{j=1}^{n-1} \left\| \sum_{i=1}^{n-j} \boldsymbol{t}_{n+1-i} \right\| \right)^2$ and using the lemma above on the individual terms, we get

$$\mathbb{E}\left[ \|\boldsymbol{z}\|^2 \right] \leq 2n^2 \alpha^4 L^2 G^2 + 2n^2 \alpha^4 L^4 (90\alpha^2 n^3 G^2 \log n).$$

### E.3 PROOF OF LEMMA 6

This proof is similar to the one in (Ahn et al., 2020). Define $G^* := \max_i \|\boldsymbol{b}_i\|$. We use the following theorem adapted to our setting

**Theorem 7.** [(Schneider, 2016, Theorem 2)] *With probability at least $1 - \frac{\delta}{n}$,*

$$\left\| \sum_{i=1}^{j} \boldsymbol{t}_i \right\| \leq \alpha G^* \sqrt{8j \left( 1 - \frac{j-1}{n} \right) \log \frac{2n}{\delta}}.$$

Then taking a union bound over $j = 1, \ldots, n$, we get that with probability at least $1 - \delta$,

$$\forall j \in [1, n] : \left\| \sum_{i=1}^{j} \boldsymbol{t}_i \right\| \leq \alpha G^* \sqrt{8j \left(1 - \frac{j-1}{n}\right) \log \frac{2n}{\delta}} \leq \alpha G^* \sqrt{8j \log \frac{2n}{\delta}}.$$

Then, for the complementary event (which happens with probability $\delta$), we use the fact that $\|\boldsymbol{t}_i\| = \|\alpha \boldsymbol{b}_{\sigma_i}\| \leq \alpha G^*$ to get the following:

$$\forall j \in [1, n] : \left\| \sum_{i=1}^{j} \boldsymbol{t}_i \right\| \leq \sum_{i=1}^{j} \|\boldsymbol{t}_i\| \leq \alpha G^* j.$$

Now, choose $\delta = 1/n$. Then, we get that

$$\mathbb{E}\left[ \left\| \sum_{i=1}^{j} \boldsymbol{t}_i \right\|^2 \right] \leq \left(1 - \frac{1}{n}\right) 8j\alpha^2 G^2 \log(2n^2) + \frac{1}{n}(\alpha G^* j)^2$$

$$\leq 18j\alpha^2 (G^*)^2 \log n.$$

Similarly, we can also get

$$\mathbb{E}\left[ \left\| \sum_{i=1}^{j} \boldsymbol{t}_i \right\| \left\| \sum_{i=1}^{l} \boldsymbol{t}_i \right\| \right] \leq 18\sqrt{jl}\alpha^2 (G^*)^2 \log(n).$$

### E.4 PROOF OF LEMMA 5

As in the lemma's statement, define $(\widetilde{\boldsymbol{S}}_1, \ldots, \widetilde{\boldsymbol{S}}_{2n})$ as $(\widetilde{\boldsymbol{S}}_1, \ldots, \widetilde{\boldsymbol{S}}_n) = (\boldsymbol{S}_1, \ldots, \boldsymbol{S}_n)$ and $(\widetilde{\boldsymbol{S}}_{n+1}, \ldots, \widetilde{\boldsymbol{S}}_{2n}) = (\boldsymbol{S}_n, \ldots, \boldsymbol{S}_1)$. As a reminder, the term $\boldsymbol{z}$ is defined as follows:

$$\boldsymbol{z} := \sum_{i=1}^{n} \left( \prod_{j=1}^{n} (\boldsymbol{I} - \boldsymbol{S}_j) \right) \left( \prod_{j=1}^{n-i} (\boldsymbol{I} - \boldsymbol{S}_{n+1-j}) \right) \boldsymbol{t}_i + \sum_{i=1}^{n} \left( \prod_{j=1}^{n-i} (\boldsymbol{I} - \boldsymbol{S}_j) \right) \boldsymbol{t}_{n+1-i}. \tag{19}$$

First, we analyze the first term in $\boldsymbol{z}$. Towards that end, we start by expanding $\left( \prod_{j=1}^{n} (\boldsymbol{I} - \boldsymbol{S}_j) \right) \left( \prod_{j=1}^{n-i} (\boldsymbol{I} - \boldsymbol{S}_{n+1-j}) \right)$:

$$\left( \prod_{j=1}^{n} (\boldsymbol{I} - \boldsymbol{S}_j) \right) \left( \prod_{j=1}^{n-i} (\boldsymbol{I} - \boldsymbol{S}_{n+1-j}) \right) = \left( \prod_{j=1}^{2n-i} (\boldsymbol{I} - \widetilde{\boldsymbol{S}}_j) \right) \tag{20}$$

$$= \left( \prod_{j=1}^{2n-i-1} (\boldsymbol{I} - \widetilde{\boldsymbol{S}}_j) \right) \left( \boldsymbol{I} - \widetilde{\boldsymbol{S}}_{2n-i} \right)$$

$$= \left( \prod_{j=1}^{2n-i-1} (\boldsymbol{I} - \widetilde{\boldsymbol{S}}_j) \right) - \left( \prod_{j=1}^{2n-i-1} (\boldsymbol{I} - \widetilde{\boldsymbol{S}}_j) \right) \widetilde{\boldsymbol{S}}_{2n-i}.$$

Similarly, we expand the term $\left( \prod_{j=1}^{2n-i-1} (\boldsymbol{I} - \widetilde{\boldsymbol{S}}_j) \right)$ and then recursively keep doing it to get the following:

$$\left( \prod_{j=1}^{n} (\boldsymbol{I} - \boldsymbol{S}_j) \right) \left( \prod_{j=1}^{n-i} (\boldsymbol{I} - \boldsymbol{S}_{n+1-j}) \right) = \left( \prod_{j=1}^{2n-i-1} (\boldsymbol{I} - \widetilde{\boldsymbol{S}}_j) \right) - \left( \prod_{j=1}^{2n-i-1} (\boldsymbol{I} - \widetilde{\boldsymbol{S}}_j) \right) \widetilde{\boldsymbol{S}}_{2n-i}$$

$$= \left( \prod_{j=1}^{2n-i-2} (\boldsymbol{I} - \widetilde{\boldsymbol{S}}_j) \right) - \left( \prod_{j=1}^{2n-i-2} (\boldsymbol{I} - \widetilde{\boldsymbol{S}}_j) \right) \widetilde{\boldsymbol{S}}_{2n-i-1} - \left( \prod_{j=1}^{2n-i-1} (\boldsymbol{I} - \widetilde{\boldsymbol{S}}_j) \right) \widetilde{\boldsymbol{S}}_{2n-i}$$

$$\begin{aligned}
&= \quad \vdots \\
&= (\boldsymbol{I} - \widetilde{\boldsymbol{S}}_1) - (\boldsymbol{I} - \widetilde{\boldsymbol{S}}_1)\widetilde{\boldsymbol{S}}_2 - \ldots \\
&\quad - \left( \prod_{j=1}^{2n-i-2}(\boldsymbol{I} - \widetilde{\boldsymbol{S}}_j) \right) \widetilde{\boldsymbol{S}}_{2n-i-1} - \left( \prod_{j=1}^{2n-i-1}(\boldsymbol{I} - \widetilde{\boldsymbol{S}}_j) \right) \widetilde{\boldsymbol{S}}_{2n-i} \\
&= \boldsymbol{I} - \sum_{j=1}^{2n-i} \left( \prod_{l=1}^{j-1}(\boldsymbol{I} - \widetilde{\boldsymbol{S}}_l) \right) \widetilde{\boldsymbol{S}}_j.
\end{aligned}$$

Note that the term $\prod_{l=1}^{j-1}(\boldsymbol{I} - \widetilde{\boldsymbol{S}}_l)$ above is similar to the RHS of Eq. (20). Hence, we repeat the process again on this term to get the following

$$\begin{aligned}
\left( \prod_{j=1}^{n}(\boldsymbol{I} - \boldsymbol{S}_j) \right) \left( \prod_{j=1}^{n-i}(\boldsymbol{I} - \boldsymbol{S}_{n+1-j}) \right) &= \boldsymbol{I} - \sum_{j=1}^{2n-i} \left( \boldsymbol{I} - \sum_{l=1}^{j-1} \left( \prod_{p=1}^{l-1}(\boldsymbol{I} - \widetilde{\boldsymbol{S}}_p) \right) \widetilde{\boldsymbol{S}}_l \right) \widetilde{\boldsymbol{S}}_j \\
&= \boldsymbol{I} - \sum_{j=1}^{2n-i} \widetilde{\boldsymbol{S}}_j + \sum_{j=1}^{2n-i} \sum_{l=1}^{j-1} \left( \prod_{p=1}^{l-1}(\boldsymbol{I} - \widetilde{\boldsymbol{S}}_p) \right) \widetilde{\boldsymbol{S}}_l \widetilde{\boldsymbol{S}}_j. \quad (21)
\end{aligned}$$

Using this in the first term $\sum_{i=1}^{n} \left( \prod_{j=1}^{n}(\boldsymbol{I} - \boldsymbol{S}_j) \right) \left( \prod_{j=1}^{n-i}(\boldsymbol{I} - \boldsymbol{S}_{n+1-j}) \right) \boldsymbol{t}_i$ (in Eq. (19)), we get

$$\begin{aligned}
\sum_{i=1}^{n} \left( \prod_{j=1}^{n}(\boldsymbol{I} - \boldsymbol{S}_j) \right) \left( \prod_{j=1}^{n-i}(\boldsymbol{I} - \boldsymbol{S}_{n+1-j}) \right) \boldsymbol{t}_i &= \sum_{i=1}^{n} \left( \boldsymbol{I} - \sum_{j=1}^{2n-i} \widetilde{\boldsymbol{S}}_j + \sum_{j=1}^{2n-i} \sum_{l=1}^{j-1} \left( \prod_{p=1}^{l-1}(\boldsymbol{I} - \widetilde{\boldsymbol{S}}_p) \right) \widetilde{\boldsymbol{S}}_l \widetilde{\boldsymbol{S}}_j \right) \boldsymbol{t}_i \\
&= \sum_{i=1}^{n} \boldsymbol{t}_i - \sum_{i=1}^{n} \sum_{j=1}^{2n-i} \widetilde{\boldsymbol{S}}_j \boldsymbol{t}_i + \sum_{i=1}^{n} \sum_{j=1}^{2n-i} \sum_{l=1}^{j-1} \left( \prod_{p=1}^{l-1}(\boldsymbol{I} - \widetilde{\boldsymbol{S}}_p) \right) \widetilde{\boldsymbol{S}}_l \widetilde{\boldsymbol{S}}_j \boldsymbol{t}_i.
\end{aligned}$$

Now, we use the fact that $\sum_{i=1}^{n} \boldsymbol{b}_i = \boldsymbol{0}$ (Eq. (16)) to get that $\sum_{i=1}^{n} \boldsymbol{t}_i = \boldsymbol{0}$. Then,

$$\sum_{i=1}^{n} \left( \prod_{j=1}^{n}(\boldsymbol{I} - \boldsymbol{S}_j) \right) \left( \prod_{j=1}^{n-i}(\boldsymbol{I} - \boldsymbol{S}_{n+1-j}) \right) \boldsymbol{t}_i = -\sum_{i=1}^{n} \sum_{j=1}^{2n-i} \widetilde{\boldsymbol{S}}_j \boldsymbol{t}_i + \sum_{i=1}^{n} \sum_{j=1}^{2n-i} \sum_{l=1}^{j-1} \left( \prod_{p=1}^{l-1}(\boldsymbol{I} - \widetilde{\boldsymbol{S}}_p) \right) \widetilde{\boldsymbol{S}}_l \widetilde{\boldsymbol{S}}_j \boldsymbol{t}_i$$

For convenience, we define $\widetilde{\boldsymbol{M}}_j := \sum_{l=1}^{j-1} \left( \prod_{p=1}^{l-1}(\boldsymbol{I} - \widetilde{\boldsymbol{S}}_p) \right) \widetilde{\boldsymbol{S}}_l \widetilde{\boldsymbol{S}}_j$. Then,

$$\begin{aligned}
\sum_{i=1}^{n} \left( \prod_{j=1}^{n}(\boldsymbol{I} - \boldsymbol{S}_j) \right) \left( \prod_{j=1}^{n-i}(\boldsymbol{I} - \boldsymbol{S}_{n+1-j}) \right) \boldsymbol{t}_i &= -\sum_{i=1}^{n} \sum_{j=1}^{2n-i} \widetilde{\boldsymbol{S}}_j \boldsymbol{t}_i + \sum_{i=1}^{n} \sum_{j=1}^{2n-i} \widetilde{\boldsymbol{M}}_j \boldsymbol{t}_i \\
&= -\sum_{i=1}^{n} \sum_{j=1}^{2n-i} \widetilde{\boldsymbol{S}}_j \boldsymbol{t}_i + \sum_{i=1}^{n} \sum_{j=1}^{n} \widetilde{\boldsymbol{M}}_j \boldsymbol{t}_i + \sum_{i=1}^{n} \sum_{j=n+1}^{2n-i} \widetilde{\boldsymbol{M}}_j \boldsymbol{t}_i \\
&= -\sum_{i=1}^{n} \sum_{j=1}^{2n-i} \widetilde{\boldsymbol{S}}_j \boldsymbol{t}_i + \sum_{j=1}^{n} \widetilde{\boldsymbol{M}}_j \sum_{i=1}^{n} \boldsymbol{t}_i + \sum_{i=1}^{n} \sum_{j=n+1}^{2n-i} \widetilde{\boldsymbol{M}}_j \boldsymbol{t}_i \\
&= -\sum_{i=1}^{n} \sum_{j=1}^{2n-i} \widetilde{\boldsymbol{S}}_j \boldsymbol{t}_i + \sum_{i=1}^{n} \sum_{j=n+1}^{2n-i} \widetilde{\boldsymbol{M}}_j \boldsymbol{t}_i. \qquad \text{(Since } \sum_{i=1}^{n} \boldsymbol{t}_i = \boldsymbol{0}\text{)}
\end{aligned}$$

Note that $\sum_{i=1}^{n} \sum_{j=n+1}^{2n-i} \widetilde{\boldsymbol{M}}_j \boldsymbol{t}_i = \sum_{j=n+1}^{2n-1} \widetilde{\boldsymbol{M}}_j \left( \sum_{i=1}^{2n-j} \boldsymbol{t}_i \right)$. Hence,

$$\sum_{i=1}^{n} \left( \prod_{j=1}^{n}(\boldsymbol{I} - \boldsymbol{S}_j) \right) \left( \prod_{j=1}^{n-i}(\boldsymbol{I} - \boldsymbol{S}_{n+1-j}) \right) \boldsymbol{t}_i = -\sum_{i=1}^{n} \sum_{j=1}^{2n-i} \widetilde{\boldsymbol{S}}_j \boldsymbol{t}_i + \sum_{j=n+1}^{2n-1} \widetilde{\boldsymbol{M}}_j \left( \sum_{i=1}^{2n-j} \boldsymbol{t}_i \right)$$

$$
\begin{aligned}
&= -\sum_{i=1}^{n}\sum_{j=1}^{n}\widetilde{\boldsymbol{S}}_j \boldsymbol{t}_i - \sum_{i=1}^{n}\sum_{j=n+1}^{2n-i}\widetilde{\boldsymbol{S}}_j \boldsymbol{t}_i + \sum_{j=n+1}^{2n-1}\widetilde{\boldsymbol{M}}_j \left(\sum_{i=1}^{2n-j}\boldsymbol{t}_i\right)\\
&= -\sum_{j=1}^{n}\widetilde{\boldsymbol{S}}_j \sum_{i=1}^{n}\boldsymbol{t}_i - \sum_{i=1}^{n}\sum_{j=n+1}^{2n-i}\widetilde{\boldsymbol{S}}_j \boldsymbol{t}_i + \sum_{j=n+1}^{2n-1}\widetilde{\boldsymbol{M}}_j \left(\sum_{i=1}^{2n-j}\boldsymbol{t}_i\right)\\
&= -\sum_{i=1}^{n}\sum_{j=n+1}^{2n-i}\widetilde{\boldsymbol{S}}_j \boldsymbol{t}_i + \sum_{j=n+1}^{2n-1}\widetilde{\boldsymbol{M}}_j \left(\sum_{i=1}^{2n-j}\boldsymbol{t}_i\right) \qquad \text{(Since } \sum_{i=1}^{n}\boldsymbol{t}_i = \boldsymbol{0}\text{)}\\
&= -\sum_{i=1}^{n}\sum_{j=n+1}^{2n-i}\widetilde{\boldsymbol{S}}_j \boldsymbol{t}_i + \sum_{j=n+1}^{2n-1}\sum_{l=1}^{j-1}\left(\prod_{p=1}^{l-1}(\boldsymbol{I}-\widetilde{\boldsymbol{S}}_p)\right)\widetilde{\boldsymbol{S}}_l\widetilde{\boldsymbol{S}}_j \left(\sum_{i=1}^{2n-j}\boldsymbol{t}_i\right)\\
&= -\sum_{i=1}^{n}\sum_{j=i+1}^{n}\boldsymbol{S}_j \boldsymbol{t}_i + \sum_{j=n+1}^{2n-1}\sum_{l=1}^{j-1}\left(\prod_{p=1}^{l-1}(\boldsymbol{I}-\widetilde{\boldsymbol{S}}_p)\right)\widetilde{\boldsymbol{S}}_l\widetilde{\boldsymbol{S}}_j \left(\sum_{i=1}^{2n-j}\boldsymbol{t}_i\right). \quad (22)
\end{aligned}
$$

Next we analyze the second term in $\boldsymbol{z}$. For this, we start by expanding $\prod_{j=1}^{n-i}(\boldsymbol{I}-\boldsymbol{S}_j)$ in a similar way as Eq. (21)

$$
\prod_{j=1}^{n-i}(\boldsymbol{I}-\boldsymbol{S}_j) = \boldsymbol{I} - \sum_{j=1}^{n-i}\boldsymbol{S}_j + \sum_{j=1}^{n-i}\sum_{l=1}^{j-1}\left(\prod_{p=1}^{l-1}(\boldsymbol{I}-\boldsymbol{S}_p)\right)\boldsymbol{S}_l\boldsymbol{S}_j \qquad (23)
$$

Using this, we get

$$
\begin{aligned}
\sum_{i=1}^{n}\left(\prod_{j=1}^{n-i}(\boldsymbol{I}-\boldsymbol{S}_j)\right)\boldsymbol{t}_{n+1-i} &= \sum_{i=1}^{n}\left(\boldsymbol{I} - \sum_{j=1}^{n-i}\boldsymbol{S}_j + \sum_{j=1}^{n-i}\sum_{l=1}^{j-1}\left(\prod_{p=1}^{l-1}(\boldsymbol{I}-\boldsymbol{S}_p)\right)\boldsymbol{S}_l\boldsymbol{S}_j\right)\boldsymbol{t}_{n+1-i}\\
&= \sum_{i=1}^{n}\boldsymbol{t}_{n+1-i} - \sum_{i=1}^{n}\sum_{j=1}^{n-i}\boldsymbol{S}_j\boldsymbol{t}_{n+1-i} + \sum_{i=1}^{n}\sum_{j=1}^{n-i}\sum_{l=1}^{j-1}\left(\prod_{p=1}^{l-1}(\boldsymbol{I}-\boldsymbol{S}_p)\right)\boldsymbol{S}_l\boldsymbol{S}_j\boldsymbol{t}_{n+1-i}\\
&= -\sum_{i=1}^{n}\sum_{j=1}^{n-i}\boldsymbol{S}_j\boldsymbol{t}_{n+1-i} + \sum_{i=1}^{n}\sum_{j=1}^{n-i}\sum_{l=1}^{j-1}\left(\prod_{p=1}^{l-1}(\boldsymbol{I}-\boldsymbol{S}_p)\right)\boldsymbol{S}_l\boldsymbol{S}_j\boldsymbol{t}_{n+1-i},
\end{aligned}
$$

where we used the fact that $\sum_{i=1}^{n}\boldsymbol{t}_i = \boldsymbol{0}$ in the last equality. For convenience, we define $\boldsymbol{M}_j := \sum_{l=1}^{j-1}\left(\prod_{p=1}^{l-1}(\boldsymbol{I}-\boldsymbol{S}_p)\right)\boldsymbol{S}_l\boldsymbol{S}_j$. Then,

$$
\sum_{i=1}^{n}\left(\prod_{j=1}^{n-i}(\boldsymbol{I}-\boldsymbol{S}_j)\right)\boldsymbol{t}_{n+1-i} = -\sum_{i=1}^{n}\sum_{j=1}^{n-i}\boldsymbol{S}_j\boldsymbol{t}_{n+1-i} + \sum_{i=1}^{n}\sum_{j=1}^{n-i}\boldsymbol{M}_j\boldsymbol{t}_{n+1-i}.
$$

Since $\sum_{i=1}^{n}\sum_{j=1}^{n-i}\boldsymbol{M}_j\boldsymbol{t}_{n+1-i} = \sum_{j=1}^{n-1}\boldsymbol{M}_j\left(\sum_{i=1}^{n-j}\boldsymbol{t}_{n+1-i}\right)$, we get

$$
\begin{aligned}
\sum_{i=1}^{n}\left(\prod_{j=1}^{n-i}(\boldsymbol{I}-\boldsymbol{S}_j)\right)\boldsymbol{t}_{n+1-i} &= -\sum_{i=1}^{n}\sum_{j=1}^{n-i}\boldsymbol{S}_j\boldsymbol{t}_{n+1-i} + \sum_{j=1}^{n-1}\boldsymbol{M}_j\left(\sum_{i=1}^{n-j}\boldsymbol{t}_{n+1-i}\right)\\
&= -\sum_{l=1}^{n}\sum_{j=1}^{l-1}\boldsymbol{S}_j\boldsymbol{t}_l + \sum_{j=1}^{n-1}\boldsymbol{M}_j\left(\sum_{i=1}^{n-j}\boldsymbol{t}_{n+1-i}\right)\\
&= -\sum_{i=1}^{n}\sum_{j=1}^{i-1}\boldsymbol{S}_j\boldsymbol{t}_i + \sum_{j=1}^{n-1}\boldsymbol{M}_j\left(\sum_{i=1}^{n-j}\boldsymbol{t}_{n+1-i}\right)\\
&= -\sum_{i=1}^{n}\sum_{j=1}^{i-1}\boldsymbol{S}_j\boldsymbol{t}_i + \sum_{j=1}^{n-1}\sum_{l=1}^{j-1}\left(\prod_{p=1}^{l-1}(\boldsymbol{I}-\boldsymbol{S}_p)\right)\boldsymbol{S}_l\boldsymbol{S}_j\left(\sum_{i=1}^{n-j}\boldsymbol{t}_{n+1-i}\right).
\end{aligned}
$$
$$(24)$$

Finally, substituting Eq. (22) and (24) in the definition of $\boldsymbol{z}$ (Eq. (19)), we get

$$
\begin{aligned}
\boldsymbol{z} = &-\sum_{i=1}^{n}\sum_{j=i+1}^{n}\boldsymbol{S}_j\boldsymbol{t}_i + \sum_{j=n+1}^{2n-1}\sum_{l=1}^{j-1}\left(\prod_{p=1}^{l-1}(\boldsymbol{I}-\widetilde{\boldsymbol{S}}_p)\right)\widetilde{\boldsymbol{S}}_l\widetilde{\boldsymbol{S}}_j\left(\sum_{i=1}^{2n-j}\boldsymbol{t}_i\right) \\
&-\sum_{i=1}^{n}\sum_{j=1}^{i-1}\boldsymbol{S}_j\boldsymbol{t}_i + \sum_{j=1}^{n-1}\sum_{l=1}^{j-1}\left(\prod_{p=1}^{l-1}(\boldsymbol{I}-\boldsymbol{S}_p)\right)\boldsymbol{S}_l\boldsymbol{S}_j\left(\sum_{i=1}^{n-j}\boldsymbol{t}_{n+1-i}\right) \\
= &-\sum_{i=1}^{n}\left(\sum_{j=1}^{i-1}\boldsymbol{S}_j + \sum_{j=i+1}^{n}\boldsymbol{S}_j\right)\boldsymbol{t}_i \\
&+ \sum_{j=n+1}^{2n-1}\sum_{l=1}^{j-1}\left(\prod_{p=1}^{l-1}(\boldsymbol{I}-\widetilde{\boldsymbol{S}}_p)\right)\widetilde{\boldsymbol{S}}_l\widetilde{\boldsymbol{S}}_j\left(\sum_{i=1}^{2n-j}\boldsymbol{t}_i\right) + \sum_{j=1}^{n-1}\sum_{l=1}^{j-1}\left(\prod_{p=1}^{l-1}(\boldsymbol{I}-\boldsymbol{S}_p)\right)\boldsymbol{S}_l\boldsymbol{S}_j\left(\sum_{i=1}^{n-j}\boldsymbol{t}_{n+1-i}\right) \\
= &-\sum_{i=1}^{n}\left(\sum_{j=1}^{n}\boldsymbol{S}_j\right)\boldsymbol{t}_i + \sum_{i=1}^{n}\boldsymbol{S}_i\boldsymbol{t}_i \\
&+ \sum_{j=n+1}^{2n-1}\sum_{l=1}^{j-1}\left(\prod_{p=1}^{l-1}(\boldsymbol{I}-\widetilde{\boldsymbol{S}}_p)\right)\widetilde{\boldsymbol{S}}_l\widetilde{\boldsymbol{S}}_j\left(\sum_{i=1}^{2n-j}\boldsymbol{t}_i\right) + \sum_{j=1}^{n-1}\sum_{l=1}^{j-1}\left(\prod_{p=1}^{l-1}(\boldsymbol{I}-\boldsymbol{S}_p)\right)\boldsymbol{S}_l\boldsymbol{S}_j\left(\sum_{i=1}^{n-j}\boldsymbol{t}_{n+1-i}\right) \\
= &-\left(\sum_{j=1}^{n}\boldsymbol{S}_j\right)\sum_{i=1}^{n}\boldsymbol{t}_i + \sum_{i=1}^{n}\boldsymbol{S}_i\boldsymbol{t}_i \\
&+ \sum_{j=n+1}^{2n-1}\sum_{l=1}^{j-1}\left(\prod_{p=1}^{l-1}(\boldsymbol{I}-\widetilde{\boldsymbol{S}}_p)\right)\widetilde{\boldsymbol{S}}_l\widetilde{\boldsymbol{S}}_j\left(\sum_{i=1}^{2n-j}\boldsymbol{t}_i\right) + \sum_{j=1}^{n-1}\sum_{l=1}^{j-1}\left(\prod_{p=1}^{l-1}(\boldsymbol{I}-\boldsymbol{S}_p)\right)\boldsymbol{S}_l\boldsymbol{S}_j\left(\sum_{i=1}^{n-j}\boldsymbol{t}_{n+1-i}\right) \\
= &\sum_{i=1}^{n}\boldsymbol{S}_i\boldsymbol{t}_i + \sum_{j=n+1}^{2n-1}\sum_{l=1}^{j-1}\left(\prod_{p=1}^{l-1}(\boldsymbol{I}-\widetilde{\boldsymbol{S}}_p)\right)\widetilde{\boldsymbol{S}}_l\widetilde{\boldsymbol{S}}_j\left(\sum_{i=1}^{2n-j}\boldsymbol{t}_i\right) + \sum_{j=1}^{n-1}\sum_{l=1}^{j-1}\left(\prod_{p=1}^{l-1}(\boldsymbol{I}-\boldsymbol{S}_p)\right)\boldsymbol{S}_l\boldsymbol{S}_j\left(\sum_{i=1}^{n-j}\boldsymbol{t}_{n+1-i}\right),
\end{aligned}
$$

where we used the fact that $\sum_{i=1}^{n}\boldsymbol{t}_i = \boldsymbol{0}$ in the last equality.

## F  PROOF OF THEOREM 5

The proof is similar to that of Theorem 4, except for that here we leverage the independence of random permutations in every other epoch. The setup is also the same as Theorem 4, but we explain it again here nevertheless, for the completeness of this proof.

Let $F(\boldsymbol{x}) := \frac{1}{n}\sum_{i=1}^{n} f_i(\boldsymbol{x})$ be such that its minimizer it at the origin. This can be assumed without loss of generality because we can shift the coordinates appropriately, similar to (Safran & Shamir, 2019). Since the $f_i$ are convex quadratics, we can write them as $f_i(\boldsymbol{x}) = \frac{1}{2}\boldsymbol{x}^\top \boldsymbol{A}_i\boldsymbol{x} - \boldsymbol{b}_i^\top\boldsymbol{x} + c_i$, where $\boldsymbol{A}_i$ are symmetric, positive-semidefinite matrices. We can omit the constants $c_i$ because they do not affect the minimizer or the gradients. Because we assume that the minimizer of $F(\boldsymbol{x})$ is at the origin, we get that

$$
\sum_{i=1}^{n}\boldsymbol{b}_i = \boldsymbol{0}. \tag{25}
$$

Let $\sigma^k = (\sigma_1^k, \sigma_2^k, \ldots, \sigma_n^k)$ be the random permutation of $(1, 2, \ldots, n)$ sampled in epoch $2k-1$. Then epoch $2k-1$ sees the $n$ functions in the reverse sequence:

$$
\left(\frac{1}{2}\boldsymbol{x}^\top\boldsymbol{A}_{\sigma_1^k}\boldsymbol{x} - \boldsymbol{b}_{\sigma_1^k}^\top\boldsymbol{x}, \frac{1}{2}\boldsymbol{x}^\top\boldsymbol{A}_{\sigma_2^k}\boldsymbol{x} - \boldsymbol{b}_{\sigma_2^k}^\top\boldsymbol{x}, \ldots, \frac{1}{2}\boldsymbol{x}^\top\boldsymbol{A}_{\sigma_n^k}\boldsymbol{x} - \boldsymbol{b}_{\sigma_n^k}^\top\boldsymbol{x}\right),
$$

whereas epoch $2k$ sees the $n$ functions in the reverse sequence:

$$
\left(\frac{1}{2}\boldsymbol{x}^\top\boldsymbol{A}_{\sigma_n^k}\boldsymbol{x} - \boldsymbol{b}_{\sigma_n^k}^\top\boldsymbol{x}, \frac{1}{2}\boldsymbol{x}^\top\boldsymbol{A}_{\sigma_{n-1}^k}\boldsymbol{x} - \boldsymbol{b}_{\sigma_{n-1}^k}^\top\boldsymbol{x}, \ldots, \frac{1}{2}\boldsymbol{x}^\top\boldsymbol{A}_{\sigma_1^k}\boldsymbol{x} - \boldsymbol{b}_{\sigma_1^k}^\top\boldsymbol{x}\right).
$$

We define $\boldsymbol{S}_i^k := \alpha \boldsymbol{A}_{\sigma_i^k}$ and $\boldsymbol{t}_i^k = \alpha \boldsymbol{b}_{\sigma_i^k}$ for convenience of notation. We start off by computing the progress made duting an even indexed epoch. Since the even epochs use the reverse permutation of $\sigma^k$, we get

$$
\begin{aligned}
\boldsymbol{x}_0^{2k+1} &= \boldsymbol{x}_n^{2k} \\
&= \boldsymbol{x}_{n-1}^{2k} - \alpha \left( \boldsymbol{A}_{\sigma_1^k} \boldsymbol{x}_{n-1}^{2k} - \boldsymbol{b}_{\sigma_1^k} \right) \qquad (f_{\sigma_1^k} \text{ used at the last iteration of epoch } 2k.) \\
&= \left( \boldsymbol{I} - \alpha \boldsymbol{A}_{\sigma_1^k} \right) \boldsymbol{x}_{n-1}^{2k} + \alpha \boldsymbol{b}_{\sigma_1^k} \\
&= (\boldsymbol{I} - \boldsymbol{S}_1^k) \boldsymbol{x}_{n-1}^{2k} + \boldsymbol{t}_1^k.
\end{aligned}
$$

We recursively apply the same procedure as above to the whole epoch to get the following

$$
\begin{aligned}
\boldsymbol{x}_0^{2k+1} &= (\boldsymbol{I} - \boldsymbol{S}_1^k) \boldsymbol{x}_{n-1}^{2k} + \boldsymbol{t}_1^k \\
&= (\boldsymbol{I} - \boldsymbol{S}_1^k) \left( (\boldsymbol{I} - \boldsymbol{S}_2) \boldsymbol{x}_{n-2}^{2k} + \boldsymbol{t}_2^k \right) + \boldsymbol{t}_1^k \\
&= (\boldsymbol{I} - \boldsymbol{S}_1^k)(\boldsymbol{I} - \boldsymbol{S}_2^k) \boldsymbol{x}_{n-2}^{2k} + (\boldsymbol{I} - \boldsymbol{S}_1) \boldsymbol{t}_2^k + \boldsymbol{t}_1^k \\
&\quad \vdots \\
&= \left( \prod_{i=1}^n (\boldsymbol{I} - \boldsymbol{S}_i^k) \right) \boldsymbol{x}_0^{2k} + \sum_{i=1}^n \left( \prod_{j=1}^{n-i} (\boldsymbol{I} - \boldsymbol{S}_j^k) \right) \boldsymbol{t}_{n+1-i}^k,
\end{aligned}
\tag{26}
$$

where the product of matrices $\{\boldsymbol{M}_i\}$ is defined as $\prod_{i=l}^m \boldsymbol{M}_i = \boldsymbol{M}_l \boldsymbol{M}_{l+1} \ldots \boldsymbol{M}_m$ if $m \geq l$ and 1 otherwise. Similar to Eq. (26), we can compute the progress made during an odd indexed epoch. Recall that the only difference will be that the odd indexed epochs see the permutations in the order $(\sigma_1^k, \sigma_2^k, \ldots, \sigma_n^k)$ instead of $(\sigma_n^k, \sigma_{n-1}^k, \ldots, \sigma_1^k)$. After doing the computation, we get the following equation:

$$
\boldsymbol{x}_0^{2k} = \left( \prod_{i=1}^n (\boldsymbol{I} - \boldsymbol{S}_{n-i+1}^k) \right) \boldsymbol{x}_0^{2k-1} + \sum_{i=1}^n \left( \prod_{j=1}^{n-i} (\boldsymbol{I} - \boldsymbol{S}_{n-j+1}^k) \right) \boldsymbol{t}_i^k.
$$

Combining the results above, we can get the total progress made after the pair of epoch $2k - 1$ and $2k$:

$$
\begin{aligned}
\boldsymbol{x}_0^{2k+1} &= \left( \prod_{i=1}^n (\boldsymbol{I} - \boldsymbol{S}_i^k) \right) \boldsymbol{x}_0^{2k} + \sum_{i=1}^n \left( \prod_{j=1}^{n-i} (\boldsymbol{I} - \boldsymbol{S}_j^k) \right) \boldsymbol{t}_{n+1-i}^k \\
&= \left( \prod_{i=1}^n (\boldsymbol{I} - \boldsymbol{S}_i^k) \right) \left( \prod_{i=1}^n (\boldsymbol{I} - \boldsymbol{S}_{n-i+1}^k) \right) \boldsymbol{x}_0^{2k-1} + \left( \prod_{j=1}^n (\boldsymbol{I} - \boldsymbol{S}_j^k) \right) \sum_{i=1}^n \left( \prod_{j=1}^{n-i} (\boldsymbol{I} - \boldsymbol{S}_{n+1-j}^k) \right) \boldsymbol{t}_i^k \\
&\quad + \sum_{i=1}^n \left( \prod_{j=1}^{n-i} (\boldsymbol{I} - \boldsymbol{S}_j^k) \right) \boldsymbol{t}_{n+1-i}^k.
\end{aligned}
\tag{27}
$$

In the sum above, the first term will have an exponential decay, hence we need to control the next two terms. Similar to Theorem 4, we denote the sum of the terms as $\boldsymbol{z}^k$:

$$
\begin{aligned}
\boldsymbol{z}^k &:= \left( \prod_{j=1}^n (\boldsymbol{I} - \boldsymbol{S}_j^k) \right) \sum_{i=1}^n \left( \prod_{j=1}^{n-i} (\boldsymbol{I} - \boldsymbol{S}_{n+1-j}^k) \right) \boldsymbol{t}_i^k + \sum_{i=1}^n \left( \prod_{j=1}^{n-i} (\boldsymbol{I} - \boldsymbol{S}_j^k) \right) \boldsymbol{t}_{n+1-i}^k \\
&= \sum_{i=1}^n \left( \prod_{j=1}^n (\boldsymbol{I} - \boldsymbol{S}_j^k) \right) \left( \prod_{j=1}^{n-i} (\boldsymbol{I} - \boldsymbol{S}_{n+1-j}^k) \right) \boldsymbol{t}_i^k + \sum_{i=1}^n \left( \prod_{j=1}^{n-i} (\boldsymbol{I} - \boldsymbol{S}_j^k) \right) \boldsymbol{t}_{n+1-i}^k.
\end{aligned}
$$

To see where the iterates end up after $K$ epochs, we simply set $2k = K$ in Eq. 27 and then keep applying the equation recursively to preceding epochs. Then, we get

$$
\boldsymbol{x}_n^K = \boldsymbol{x}_0^{K+1} = \left( \prod_{i=1}^n (\boldsymbol{I} - \boldsymbol{S}_i^{\frac{K}{2}}) \right) \left( \prod_{i=1}^n (\boldsymbol{I} - \boldsymbol{S}_{n-i+1}^{\frac{K}{2}}) \right) \boldsymbol{x}_0^{K-1} + \boldsymbol{z}^{\frac{K}{2}}
$$

$$
= \left( \left( \prod_{i=1}^{n} (\boldsymbol{I} - \boldsymbol{S}_i^{\frac{K}{2}}) \right) \left( \prod_{i=1}^{n} (\boldsymbol{I} - \boldsymbol{S}_{n-i+1}^{\frac{K}{2}}) \right) \right) \left( \left( \prod_{i=1}^{n} (\boldsymbol{I} - \boldsymbol{S}_i^{\frac{K}{2}-1}) \right) \left( \prod_{i=1}^{n} (\boldsymbol{I} - \boldsymbol{S}_{n-i+1}^{\frac{K}{2}-1}) \right) \right) \boldsymbol{x}_0^{K-3}
$$

$$
+ \left( \prod_{i=1}^{n} (\boldsymbol{I} - \boldsymbol{S}_i^{\frac{K}{2}}) \right) \left( \prod_{i=1}^{n} (\boldsymbol{I} - \boldsymbol{S}_{n-i+1}^{\frac{K}{2}}) \right) \boldsymbol{z}^{\frac{K}{2}-1} + \boldsymbol{z}^{\frac{K}{2}}
$$

$$
\vdots
$$

$$
= \left( \prod_{k=0}^{\frac{K}{2}-1} \left( \prod_{i=1}^{n} (\boldsymbol{I} - \boldsymbol{S}_i^{\frac{K}{2}-k}) \right) \left( \prod_{i=1}^{n} (\boldsymbol{I} - \boldsymbol{S}_{n-i+1}^{\frac{K}{2}-k}) \right) \right) \boldsymbol{x}_0^1
$$

$$
+ \sum_{k=0}^{\frac{K}{2}-1} \left( \prod_{l=0}^{k-1} \left( \prod_{i=1}^{n} (\boldsymbol{I} - \boldsymbol{S}_i^{\frac{K}{2}-l}) \right) \left( \prod_{i=1}^{n} (\boldsymbol{I} - \boldsymbol{S}_{n-i+1}^{\frac{K}{2}-l}) \right) \right) \boldsymbol{z}^{\frac{K}{2}-k}.
$$

Taking squared norms and expectations on both sides, we get

$$
\mathbb{E}[\|\boldsymbol{x}_n^K\|^2] = \mathbb{E} \left[ \left\| \left( \prod_{k=0}^{\frac{K}{2}-1} \left( \prod_{i=1}^{n} (\boldsymbol{I} - \boldsymbol{S}_i^{\frac{K}{2}-k}) \right) \left( \prod_{i=1}^{n} (\boldsymbol{I} - \boldsymbol{S}_{n-i+1}^{\frac{K}{2}-k}) \right) \right) \boldsymbol{x}_0^1 \right. \right.
$$

$$
+ \left. \left. \sum_{k=0}^{\frac{K}{2}-1} \left( \prod_{l=0}^{k-1} \left( \prod_{i=1}^{n} (\boldsymbol{I} - \boldsymbol{S}_i^{\frac{K}{2}-l}) \right) \left( \prod_{i=1}^{n} (\boldsymbol{I} - \boldsymbol{S}_{n-i+1}^{\frac{K}{2}-l}) \right) \right) \boldsymbol{z}^{\frac{K}{2}-k} \right\|^2 \right]
$$

$$
\leq 2\mathbb{E} \left[ \left\| \left( \prod_{k=0}^{\frac{K}{2}-1} \left( \prod_{i=1}^{n} (\boldsymbol{I} - \boldsymbol{S}_i^{\frac{K}{2}-k}) \right) \left( \prod_{i=1}^{n} (\boldsymbol{I} - \boldsymbol{S}_{n-i+1}^{\frac{K}{2}-k}) \right) \right) \boldsymbol{x}_0^1 \right\|^2 \right]
$$

$$
+ 2\mathbb{E} \left[ \left\| \sum_{k=0}^{\frac{K}{2}-1} \left( \prod_{l=0}^{k-1} \left( \prod_{i=1}^{n} (\boldsymbol{I} - \boldsymbol{S}_i^{\frac{K}{2}-l}) \right) \left( \prod_{i=1}^{n} (\boldsymbol{I} - \boldsymbol{S}_{n-i+1}^{\frac{K}{2}-l}) \right) \right) \boldsymbol{z}^{\frac{K}{2}-k} \right\|^2 \right], \quad (28)
$$

where we used the fact that $(a+b)^2 \leq 2a^2 + 2b^2$. Next, we expand the second term above to get

$$
\mathbb{E}[\|\boldsymbol{x}_n^K\|^2] \leq 2\mathbb{E} \left[ \left\| \left( \prod_{k=0}^{\frac{K}{2}-1} \left( \prod_{i=1}^{n} (\boldsymbol{I} - \boldsymbol{S}_i^{\frac{K}{2}-k}) \right) \left( \prod_{i=1}^{n} (\boldsymbol{I} - \boldsymbol{S}_{n-i+1}^{\frac{K}{2}-k}) \right) \right) \boldsymbol{x}_0^1 \right\|^2 \right]
$$

$$
+ 2 \sum_{k=0}^{\frac{K}{2}-1} \mathbb{E} \left[ \left\| \left( \prod_{l=0}^{k-1} \left( \prod_{i=1}^{n} (\boldsymbol{I} - \boldsymbol{S}_i^{\frac{K}{2}-l}) \right) \left( \prod_{i=1}^{n} (\boldsymbol{I} - \boldsymbol{S}_{n-i+1}^{\frac{K}{2}-l}) \right) \right) \boldsymbol{z}^{\frac{K}{2}-k} \right\|^2 \right]
$$

$$
+ 4 \sum_{0 \leq k' < k \leq \frac{K}{2}-1} \mathbb{E} \left[ \left\langle \left( \prod_{l=0}^{k-1} \left( \prod_{i=1}^{n} (\boldsymbol{I} - \boldsymbol{S}_i^{\frac{K}{2}-l}) \right) \left( \prod_{i=1}^{n} (\boldsymbol{I} - \boldsymbol{S}_{n-i+1}^{\frac{K}{2}-l}) \right) \right) \boldsymbol{z}^{\frac{K}{2}-k}, \right. \right.
$$

$$
\left. \left. \left( \prod_{l=0}^{k'-1} \left( \prod_{i=1}^{n} (\boldsymbol{I} - \boldsymbol{S}_i^{\frac{K}{2}-l}) \right) \left( \prod_{i=1}^{n} (\boldsymbol{I} - \boldsymbol{S}_{n-i+1}^{\frac{K}{2}-l}) \right) \right) \boldsymbol{z}^{\frac{K}{2}-k'} \right\rangle \right]. \quad (29)
$$

We handle each of the three terms separately. Let $\boldsymbol{N}_k := \left( \prod_{i=1}^{n} (\boldsymbol{I} - \boldsymbol{S}_i^{\frac{K}{2}-k}) \right) \left( \prod_{i=1}^{n} (\boldsymbol{I} - \boldsymbol{S}_{n-i+1}^{\frac{K}{2}-k}) \right)$. The first term can be written as:

$$
\mathbb{E} \left[ \left\| \left( \prod_{k=0}^{\frac{K}{2}-1} \left( \prod_{i=1}^{n} (\boldsymbol{I} - \boldsymbol{S}_i^{\frac{K}{2}-k}) \right) \left( \prod_{i=1}^{n} (\boldsymbol{I} - \boldsymbol{S}_{n-i+1}^{\frac{K}{2}-k}) \right) \right) \boldsymbol{x}_0^1 \right\|^2 \right] = \mathbb{E} \left[ \left\| \left( \prod_{k=0}^{\frac{K}{2}-1} \boldsymbol{N}_k \right) \boldsymbol{x}_0^1 \right\|^2 \right]
$$

$$= \mathbb{E}\left[(\boldsymbol{x}_0^1)^\top \left(\prod_{k=0}^{\frac{K}{2}-1} \boldsymbol{N}_k\right)^\top \left(\prod_{k=0}^{\frac{K}{2}-1} \boldsymbol{N}_k\right) \boldsymbol{x}_0^1\right]$$

$$= \mathbb{E}\left[(\boldsymbol{x}_0^1)^\top \left(\prod_{k=1}^{\frac{K}{2}-1} \boldsymbol{N}_k\right)^\top \boldsymbol{N}_0^\top \boldsymbol{N}_0 \left(\prod_{k=1}^{\frac{K}{2}-1} \boldsymbol{N}_k\right) \boldsymbol{x}_0^1\right]$$

$$= \mathbb{E}\left[(\boldsymbol{x}_0^1)^\top \left(\prod_{k=1}^{\frac{K}{2}-1} \boldsymbol{N}_k\right)^\top \mathbb{E}\left[\boldsymbol{N}_0^\top \boldsymbol{N}_0\right] \left(\prod_{k=1}^{\frac{K}{2}-1} \boldsymbol{N}_k\right) \boldsymbol{x}_0^1\right],$$

where, in the last line, we used the fact that the permutations in every epoch are independent of the permutations in other epochs. Next, we have the following lemma that bounds the spectral norm of $\mathbb{E}\left[\boldsymbol{N}_k^\top \boldsymbol{N}_k\right]$ for any $k \in [0, \frac{K}{2}-1]$:

**Lemma 7.** *For any $0 \le \alpha \le \frac{3}{16nL}\min\{1, \sqrt{\frac{n}{\kappa}}\}$,*

$$\left\|\mathbb{E}[\boldsymbol{N}_k^\top \boldsymbol{N}_k]\right\| \le 1 - \alpha n\mu$$

Note that $K \ge 55\kappa \log(nK) \max\left\{1, \sqrt{\frac{n}{\kappa}}\right\} \implies \alpha \le \frac{3}{16nL}\min\{1, \frac{n}{\kappa}\}$, and thus this lemma. Using it, we get

$$\mathbb{E}\left[\left\|\left(\prod_{k=0}^{\frac{K}{2}-1}\left(\prod_{i=1}^{n}(\boldsymbol{I}-\boldsymbol{S}_i^{\frac{K}{2}-k})\right)\left(\prod_{i=1}^{n}(\boldsymbol{I}-\boldsymbol{S}_{n-i+1}^{\frac{K}{2}-k})\right)\right)\boldsymbol{x}_0^1\right\|^2\right]$$

$$= \mathbb{E}\left[(\boldsymbol{x}_0^1)^\top \left(\prod_{k=1}^{\frac{K}{2}-1} \boldsymbol{N}_k\right)^\top \mathbb{E}\left[\boldsymbol{N}_0^\top \boldsymbol{N}_0\right] \left(\prod_{k=1}^{\frac{K}{2}-1} \boldsymbol{N}_k\right) \boldsymbol{x}_0^1\right]$$

$$\le (1-n\alpha\mu)\mathbb{E}\left[(\boldsymbol{x}_0^1)^\top \left(\prod_{k=1}^{\frac{K}{2}-1} \boldsymbol{N}_k\right)^\top \left(\prod_{k=1}^{\frac{K}{2}-1} \boldsymbol{N}_k\right) \boldsymbol{x}_0^1\right]$$

$$= (1-n\alpha\mu)\mathbb{E}\left[(\boldsymbol{x}_0^1)^\top \left(\prod_{k=2}^{\frac{K}{2}-1} \boldsymbol{N}_k\right)^\top \mathbb{E}\left[\boldsymbol{N}_1^\top \boldsymbol{N}_1\right] \left(\prod_{k=2}^{\frac{K}{2}-1} \boldsymbol{N}_k\right) \boldsymbol{x}_0^1\right]$$

$$\vdots$$

$$\le (1-n\alpha\mu)^{K/2}\|\boldsymbol{x}_0^1\|^2. \tag{30}$$

Next we look at the second term in Ineq. (29):

$$\mathbb{E}\left[\left\|\left(\prod_{l=0}^{k-1}\left(\prod_{i=1}^{n}(\boldsymbol{I}-\boldsymbol{S}_i^{\frac{K}{2}-l})\right)\left(\prod_{i=1}^{n}(\boldsymbol{I}-\boldsymbol{S}_{n-i+1}^{\frac{K}{2}-l})\right)\right)\boldsymbol{z}^{\frac{K}{2}-k}\right\|^2\right]$$

$$\le \mathbb{E}\left[\left(\prod_{l=0}^{k-1}\left(\prod_{i=1}^{n}\|\boldsymbol{I}-\boldsymbol{S}_i^{\frac{K}{2}-l}\|^2\right)\left(\prod_{i=1}^{n}\|\boldsymbol{I}-\boldsymbol{S}_{n-i+1}^{\frac{K}{2}-l}\|^2\right)\right)\|\boldsymbol{z}^{\frac{K}{2}-k}\|^2\right]$$

$$\le \mathbb{E}\left[\left\|\boldsymbol{z}^{\frac{K}{2}-k}\right\|^2\right],$$

where in the last step we used that $\|\boldsymbol{I}-\boldsymbol{S}_i^k\| \le 1$ for all $i, k$. To see why this is true, recall that $\boldsymbol{S}_i^k = \alpha\boldsymbol{A}_{\sigma_i^k}$. Further by Assumption 2, $\|\boldsymbol{A}_{\sigma_i^k}\| \le L$ and hence as long as $\alpha \le 1/L$, we have that $\|\boldsymbol{I}-\boldsymbol{S}_i^k\| \le 1$.

Next, note that for any $k$ we can apply Lemma 4 on $\mathbb{E}[\|z^{\frac{K}{2}-k}\|^2]$. Hence, we get the following bound on the second term:

$$2\sum_{k=0}^{\frac{K}{2}-1} \mathbb{E}\left[\left\|\left(\prod_{l=0}^{k-1}\left(\prod_{i=1}^{n}(I-S_i^{\frac{K}{2}-l})\right)\left(\prod_{i=1}^{n}(I-S_{n-i+1}^{\frac{K}{2}-l})\right)\right)z^{\frac{K}{2}-k}\right\|^2\right]$$

$$\leq 2\frac{K}{2}(2n^2\alpha^4 L^2(G^*)^2 + 170n^5\alpha^6 L^4(G^*)^2\log n). \tag{31}$$

Finally, we focus on the third term in Ineq. (29). We have the following lemma that gives an upper bound for it:

**Lemma 8.** *Let* $\alpha \leq \frac{1}{2nL}$ *and* $n > 6$. *Then for* $0 \leq k' < k \leq \frac{K}{2}-1$,

$$\mathbb{E}\left[\left\langle\left(\prod_{l=0}^{k-1}\left(\prod_{i=1}^{n}(I-S_i^{\frac{K}{2}-l})\right)\left(\prod_{i=1}^{n}(I-S_{n-i+1}^{\frac{K}{2}-l})\right)\right)z^{\frac{K}{2}-k},\right.\right.$$

$$\left.\left.\left(\prod_{l=0}^{k'-1}\left(\prod_{i=1}^{n}(I-S_i^{\frac{K}{2}-l})\right)\left(\prod_{i=1}^{n}(I-S_{n-i+1}^{\frac{K}{2}-l})\right)\right)z^{\frac{K}{2}-k'}\right\rangle\right]$$

$$\leq 1000n^2\alpha^4 L^2(G^*)^2 + 2000n^6\alpha^7 L^5(G^*)^2\log n.$$

Using Lemma 8, and inequalities (30) and (31) in Ineq. (29), we get

$$\mathbb{E}[\|x_n^K\|^2] \leq 2(1-n\alpha\mu)^{K/2}\|x_0^1\|^2 + 2n^2 K\alpha^4 L^2(G^*)^2 + 170n^5 K\alpha^6 L^4(G^*)^2\log n$$
$$+ 1000n^2 K^2\alpha^4 L^2(G^*)^2 + 2000n^6 K^2\alpha^7 L^5(G^*)^2\log n$$
$$\leq 2(1-n\alpha\mu)^{K/2}\|x_0^1\|^2 + 1002n^2 K\alpha^4 L^2(G^*)^2 + 2170n^6 K^2\alpha^7 L^4(G^*)^2\log n.$$

Substituting $\alpha = \frac{10\log nK}{\mu nK}$ gives us the desired result.

## F.1 PROOF OF LEMMA 7

Recall that we have defined $N_k := \left(\prod_{i=1}^{n}(I-S_i^{\frac{K}{2}-k})\right)\left(\prod_{i=1}^{n}(I-S_{n-i+1}^{\frac{K}{2}-k})\right)$. Hence,

$$N_k^\top = \left(\left(\prod_{i=1}^{n}(I-S_i^{\frac{K}{2}-k})\right)\left(\prod_{i=1}^{n}(I-S_{n-i+1}^{\frac{K}{2}-k})\right)\right)^\top$$

$$= \left(\prod_{i=1}^{n}(I-S_{n-i+1}^{\frac{K}{2}-k})\right)^\top\left(\prod_{i=1}^{n}(I-S_i^{\frac{K}{2}-k})\right)^\top$$

$$= \left(\prod_{i=1}^{n}(I-S_i^{\frac{K}{2}-k})^\top\right)\left(\prod_{i=1}^{n}(I-S_{n-i+1}^{\frac{K}{2}-k})^\top\right)$$

$$= \left(\prod_{i=1}^{n}(I-S_i^{\frac{K}{2}-k})\right)\left(\prod_{i=1}^{n}(I-S_{n-i+1}^{\frac{K}{2}-k})\right)$$

$$= N_k,$$

where we used the fact that $S_i^{\frac{K}{2}-k}$ are symmetric. Hence $N_k$ is symmetric. Then,

$$\|\mathbb{E}[N_k^\top N_k]\| = \max_{x:\|x\|=1}\mathbb{E}[x^\top N_k^\top N_k x]$$

$$= \max_{x:\|x\|=1}\mathbb{E}[x^\top N_k N_k x].$$

Next, note that $\|N_k\| \leq 1$ as long as $\alpha \leq \frac{1}{nL}$. This combined with the fact that $N_k$ is symmetric gives us that $x^\top N_k N_k x \leq x^\top N_k x$. Hence, we get

$$\|\mathbb{E}[N_k^\top N_k]\| \leq \max_{x:\|x\|=1}\mathbb{E}[x^\top N_k x]$$

$$
\begin{aligned}
&= \|\mathbb{E}[\boldsymbol{N}_k]\| \\
&= \left\| \mathbb{E}\left[ \left( \prod_{i=1}^{n}(\boldsymbol{I} - \boldsymbol{S}_i^{\frac{K}{2}-k}) \right) \left( \prod_{i=1}^{n}(\boldsymbol{I} - \boldsymbol{S}_{n-i+1}^{\frac{K}{2}-k}) \right) \right] \right\| \\
&= \left\| \mathbb{E}\left[ \left( \prod_{i=1}^{n}(\boldsymbol{I} - \boldsymbol{S}_i^{\frac{K}{2}-k}) \right) \left( \prod_{i=1}^{n}(\boldsymbol{I} - \boldsymbol{S}_i^{\frac{K}{2}-k}) \right)^{\top} \right] \right\|.
\end{aligned}
$$

To complete the proof, we apply the following lemma from Ahn et al. (2020):

**Lemma 9** (Lemma 6, Ahn et al. (2020)). *For any $0 \le \alpha \le \frac{3}{16nL}\min\{1, \frac{n}{\kappa}\}$ and $k \in [K]$,*

$$
\left\| \mathbb{E}\left[ \left( \prod_{i=1}^{n}(\boldsymbol{I} - \boldsymbol{S}_i^{\frac{K}{2}-k}) \right) \left( \prod_{i=1}^{n}(\boldsymbol{I} - \boldsymbol{S}_i^{\frac{K}{2}-k}) \right)^{\top} \right] \right\| \le 1 - \alpha n \mu.
$$

**Remark 2.** *To avoid confusion, we remark here that, the term '$\boldsymbol{S}_k$' in the paper Ahn et al. (2020) is the same as the term '$\prod_{i=1}^{n}(\boldsymbol{I} - \boldsymbol{S}_i^k)$' in our paper, and hence the original lemma statement in their paper looks different from what is written above.*

### F.2 PROOF OF LEMMA 8

We begin by decomposing the product into product of independent terms, similar to proof of Lemma 8 in Ahn et al. (2020). However, after that we diverge from their proof since we use FLIPFLOP specific analysis.

$$
\begin{aligned}
&\mathbb{E}\left[ \left\langle \left( \prod_{l=0}^{k-1} \left( \prod_{i=1}^{n}(\boldsymbol{I} - \boldsymbol{S}_i^{\frac{K}{2}-l}) \right) \left( \prod_{i=1}^{n}(\boldsymbol{I} - \boldsymbol{S}_{n-i+1}^{\frac{K}{2}-l}) \right) \right) \boldsymbol{z}^{\frac{K}{2}-k}, \right. \right. \\
&\left. \left. \left( \prod_{l=0}^{k'-1} \left( \prod_{i=1}^{n}(\boldsymbol{I} - \boldsymbol{S}_i^{\frac{K}{2}-l}) \right) \left( \prod_{i=1}^{n}(\boldsymbol{I} - \boldsymbol{S}_{n-i+1}^{\frac{K}{2}-l}) \right) \right) \boldsymbol{z}^{\frac{K}{2}-k'} \right\rangle \right] \\
&= \mathbb{E}\left[ \left( \boldsymbol{z}^{\frac{K}{2}-k} \right)^{\top} \left( \prod_{l=k'}^{k+1} \left( \prod_{i=1}^{n}(\boldsymbol{I} - \boldsymbol{S}_i^{\frac{K}{2}-l}) \right) \left( \prod_{i=1}^{n}(\boldsymbol{I} - \boldsymbol{S}_{n-i+1}^{\frac{K}{2}-l}) \right) \right)^{\top} \right. \\
&\left. \left( \prod_{l=0}^{k'} \left( \prod_{i=1}^{n}(\boldsymbol{I} - \boldsymbol{S}_i^{\frac{K}{2}-l}) \right) \left( \prod_{i=1}^{n}(\boldsymbol{I} - \boldsymbol{S}_{n-i+1}^{\frac{K}{2}-l}) \right) \right)^{\top} \right. \\
&\left. \left( \prod_{l=0}^{k'-1} \left( \prod_{i=1}^{n}(\boldsymbol{I} - \boldsymbol{S}_i^{\frac{K}{2}-l}) \right) \left( \prod_{i=1}^{n}(\boldsymbol{I} - \boldsymbol{S}_{n-i+1}^{\frac{K}{2}-l}) \right) \right) \boldsymbol{z}^{\frac{K}{2}-k'} \right].
\end{aligned}
$$

Since $k' < k$, we get that $\left( \boldsymbol{z}^{\frac{K}{2}-k} \right)^{\top}, \left( \prod_{l=k'}^{k+1} \left( \prod_{i=1}^{n}(\boldsymbol{I} - \boldsymbol{S}_i^{\frac{K}{2}-l}) \right) \left( \prod_{i=1}^{n}(\boldsymbol{I} - \boldsymbol{S}_{n-i+1}^{\frac{K}{2}-l}) \right) \right)^{\top}$ and

$$
\left( \prod_{l=0}^{k'} \left( \prod_{i=1}^{n}(\boldsymbol{I} - \boldsymbol{S}_i^{\frac{K}{2}-l}) \right) \left( \prod_{i=1}^{n}(\boldsymbol{I} - \boldsymbol{S}_{n-i+1}^{\frac{K}{2}-l}) \right) \right)^{\top} \left( \prod_{l=0}^{k'-1} \left( \prod_{i=1}^{n}(\boldsymbol{I} - \boldsymbol{S}_i^{\frac{K}{2}-l}) \right) \left( \prod_{i=1}^{n}(\boldsymbol{I} - \boldsymbol{S}_{n-i+1}^{\frac{K}{2}-l}) \right) \right) \boldsymbol{z}^{\frac{K}{2}-k'}
$$

are independent. Hence, we can write the expectation as product of expectations:

$$
\begin{aligned}
&\mathbb{E}\left[ \left\langle \left( \prod_{l=0}^{k-1} \left( \prod_{i=1}^{n}(\boldsymbol{I} - \boldsymbol{S}_i^{\frac{K}{2}-l}) \right) \left( \prod_{i=1}^{n}(\boldsymbol{I} - \boldsymbol{S}_{n-i+1}^{\frac{K}{2}-l}) \right) \right) \boldsymbol{z}^{\frac{K}{2}-k}, \right. \right. \\
&\left. \left. \left( \prod_{l=0}^{k'-1} \left( \prod_{i=1}^{n}(\boldsymbol{I} - \boldsymbol{S}_i^{\frac{K}{2}-l}) \right) \left( \prod_{i=1}^{n}(\boldsymbol{I} - \boldsymbol{S}_{n-i+1}^{\frac{K}{2}-l}) \right) \right) \boldsymbol{z}^{\frac{K}{2}-k'} \right\rangle \right]
\end{aligned}
$$

$$= \mathbb{E}\left[\left(\boldsymbol{z}^{\frac{K}{2}-k}\right)^\top\right] \mathbb{E}\left[\left(\prod_{l=k'}^{k+1}\left(\prod_{i=1}^{n}(\boldsymbol{I}-\boldsymbol{S}_i^{\frac{K}{2}-l})\right)\left(\prod_{i=1}^{n}(\boldsymbol{I}-\boldsymbol{S}_{n-i+1}^{\frac{K}{2}-l})\right)\right)^\top\right]$$

$$\mathbb{E}\left[\left(\prod_{l=0}^{k'}\left(\prod_{i=1}^{n}(\boldsymbol{I}-\boldsymbol{S}_i^{\frac{K}{2}-l})\right)\left(\prod_{i=1}^{n}(\boldsymbol{I}-\boldsymbol{S}_{n-i+1}^{\frac{K}{2}-l})\right)\right)^\top\right.$$

$$\left.\left(\prod_{l=0}^{k'-1}\left(\prod_{i=1}^{n}(\boldsymbol{I}-\boldsymbol{S}_i^{\frac{K}{2}-l})\right)\left(\prod_{i=1}^{n}(\boldsymbol{I}-\boldsymbol{S}_{n-i+1}^{\frac{K}{2}-l})\right)\right)\boldsymbol{z}^{\frac{K}{2}-k'}\right].$$

Applying the Cauchy-Schwarz inequality on the decomposition above, we get

$$\mathbb{E}\left[\left\langle\left(\prod_{l=0}^{k-1}\left(\prod_{i=1}^{n}(\boldsymbol{I}-\boldsymbol{S}_i^{\frac{K}{2}-l})\right)\left(\prod_{i=1}^{n}(\boldsymbol{I}-\boldsymbol{S}_{n-i+1}^{\frac{K}{2}-l})\right)\right)\boldsymbol{z}^{\frac{K}{2}-k},\right.\right.$$

$$\left.\left.\left(\prod_{l=0}^{k'-1}\left(\prod_{i=1}^{n}(\boldsymbol{I}-\boldsymbol{S}_i^{\frac{K}{2}-l})\right)\left(\prod_{i=1}^{n}(\boldsymbol{I}-\boldsymbol{S}_{n-i+1}^{\frac{K}{2}-l})\right)\right)\boldsymbol{z}^{\frac{K}{2}-k'}\right\rangle\right]$$

$$\leq \left\|\mathbb{E}\left[\boldsymbol{z}^{\frac{K}{2}-k}\right]\right\|\left\|\mathbb{E}\left[\prod_{l=k'}^{k+1}\left(\prod_{i=1}^{n}(\boldsymbol{I}-\boldsymbol{S}_i^{\frac{K}{2}-l})\right)\left(\prod_{i=1}^{n}(\boldsymbol{I}-\boldsymbol{S}_{n-i+1}^{\frac{K}{2}-l})\right)\right]\right\|$$

$$\left\|\mathbb{E}\left[\left(\prod_{l=0}^{k'}\left(\prod_{i=1}^{n}(\boldsymbol{I}-\boldsymbol{S}_i^{\frac{K}{2}-l})\right)\left(\prod_{i=1}^{n}(\boldsymbol{I}-\boldsymbol{S}_{n-i+1}^{\frac{K}{2}-l})\right)\right)^\top\right.\right.$$

$$\left.\left.\left(\prod_{l=0}^{k'-1}\left(\prod_{i=1}^{n}(\boldsymbol{I}-\boldsymbol{S}_i^{\frac{K}{2}-l})\right)\left(\prod_{i=1}^{n}(\boldsymbol{I}-\boldsymbol{S}_{n-i+1}^{\frac{K}{2}-l})\right)\right)\boldsymbol{z}^{\frac{K}{2}-k'}\right]\right\|$$

$$\leq \left\|\mathbb{E}\left[\boldsymbol{z}^{\frac{K}{2}-k}\right]\right\|\left\|\mathbb{E}\left[\prod_{l=k'}^{k+1}\left(\prod_{i=1}^{n}\|\boldsymbol{I}-\boldsymbol{S}_i^{\frac{K}{2}-l}\|\right)\left(\prod_{i=1}^{n}\|\boldsymbol{I}-\boldsymbol{S}_{n-i+1}^{\frac{K}{2}-l}\|\right)\right]\right\|$$

$$\left\|\mathbb{E}\left[\left(\prod_{l=0}^{k'}\left(\prod_{i=1}^{n}(\boldsymbol{I}-\boldsymbol{S}_i^{\frac{K}{2}-l})\right)\left(\prod_{i=1}^{n}(\boldsymbol{I}-\boldsymbol{S}_{n-i+1}^{\frac{K}{2}-l})\right)\right)^\top\right.\right.$$

$$\left.\left.\left(\prod_{l=0}^{k'-1}\left(\prod_{i=1}^{n}(\boldsymbol{I}-\boldsymbol{S}_i^{\frac{K}{2}-l})\right)\left(\prod_{i=1}^{n}(\boldsymbol{I}-\boldsymbol{S}_{n-i+1}^{\frac{K}{2}-l})\right)\right)\boldsymbol{z}^{\frac{K}{2}-k'}\right]\right\|$$

$$\leq \left\|\mathbb{E}\left[\boldsymbol{z}^{\frac{K}{2}-k}\right]\right\|$$

$$\left\|\mathbb{E}\left[\left(\prod_{l=0}^{k'}\left(\prod_{i=1}^{n}(\boldsymbol{I}-\boldsymbol{S}_i^{\frac{K}{2}-l})\right)\left(\prod_{i=1}^{n}(\boldsymbol{I}-\boldsymbol{S}_{n-i+1}^{\frac{K}{2}-l})\right)\right)^\top\right.\right.$$

$$\left.\left.\left(\prod_{l=0}^{k'-1}\left(\prod_{i=1}^{n}(\boldsymbol{I}-\boldsymbol{S}_i^{\frac{K}{2}-l})\right)\left(\prod_{i=1}^{n}(\boldsymbol{I}-\boldsymbol{S}_{n-i+1}^{\frac{K}{2}-l})\right)\right)\boldsymbol{z}^{\frac{K}{2}-k'}\right]\right\|,$$

where in the last step we used that $\|\boldsymbol{I}-\boldsymbol{S}_i^k\| \leq 1$ for all $i, k$. To see why this is true, recall that $\boldsymbol{S}_i^k = \alpha\boldsymbol{A}_{\sigma_i^k}$. Further by Assumption 2, $\|\boldsymbol{A}_{\sigma_i^k}\| \leq L$ and hence as long as $\alpha \leq 1/L$, we have $\|\boldsymbol{I}-\boldsymbol{S}_i^k\| \leq 1$.

For the two terms in the product above, we have the following lemma:

**Lemma 10.** *If $n > 6$ and $\alpha \leq \frac{1}{nL}$, then*

$$\|\mathbb{E}[\boldsymbol{z}^{\frac{K}{2}-k}]\| \leq 28n\alpha^2 LG^* + 9\alpha^5 L^4 n^4 G^*\sqrt{2n\log n}.$$

**Lemma 11.** *If $n > 6$ and $\alpha \leq \frac{1}{2nL}$, then*

$$
\left\| \mathbb{E}\left[ \left( \prod_{l=0}^{k'} \left( \prod_{i=1}^{n} (\boldsymbol{I} - \boldsymbol{S}_i^{\frac{K}{2}-l}) \right) \left( \prod_{i=1}^{n} (\boldsymbol{I} - \boldsymbol{S}_{n-i+1}^{\frac{K}{2}-l}) \right) \right)^{\top} \right.\right.
$$

$$
\left.\left. \left( \prod_{l=0}^{k'-1} \left( \prod_{i=1}^{n} (\boldsymbol{I} - \boldsymbol{S}_i^{\frac{K}{2}-l}) \right) \left( \prod_{i=1}^{n} (\boldsymbol{I} - \boldsymbol{S}_{n-i+1}^{\frac{K}{2}-l}) \right) \right) \boldsymbol{z}^{\frac{K}{2}-k'} \right] \right\|
$$

$$
\leq 32n\alpha^2 L G^* + 24\alpha^5 L^4 n^4 G^* \sqrt{2n \log n}.
$$

Finally, using these lemma we get

$$
\mathbb{E}\left[ \left\langle \left( \prod_{l=0}^{k-1} \left( \prod_{i=1}^{n} (\boldsymbol{I} - \boldsymbol{S}_i^{\frac{K}{2}-l}) \right) \left( \prod_{i=1}^{n} (\boldsymbol{I} - \boldsymbol{S}_{n-i+1}^{\frac{K}{2}-l}) \right) \right) \boldsymbol{z}^{\frac{K}{2}-k}, \right.\right.
$$

$$
\left.\left. \left( \prod_{l=0}^{k'-1} \left( \prod_{i=1}^{n} (\boldsymbol{I} - \boldsymbol{S}_i^{\frac{K}{2}-l}) \right) \left( \prod_{i=1}^{n} (\boldsymbol{I} - \boldsymbol{S}_{n-i+1}^{\frac{K}{2}-l}) \right) \right) \boldsymbol{z}^{\frac{K}{2}-k'} \right\rangle \right]
$$

$$
\leq \left( 28n\alpha^2 L G^* + 9\alpha^5 L^4 n^4 G^* \sqrt{2n \log n} \right) \cdot \left( 32n\alpha^2 L G^* + 24\alpha^5 L^4 n^4 G^* \sqrt{2n \log n} \right)
$$

$$
\leq 896n^2\alpha^4 L^2 (G^*)^2 + 960\alpha^7 L^5 n^5 (G^*)^2 \sqrt{2n \log n} + 432\alpha^{10} L^8 n^9 (G^*)^2 \log n
$$

$$
\leq 1000n^2\alpha^4 L^2 (G^*)^2 + 2000\alpha^7 L^5 n^6 (G^*)^2 \log n.
$$

### F.3 PROOF OF LEMMA 10

Since we are dealing with just a single epoch, we will skip the superscript. Using Lemma 5, we get

$$
\|\mathbb{E}[\boldsymbol{z}]\| = \left\| \mathbb{E}\left[ \sum_{i=1}^{n} \boldsymbol{S}_i \boldsymbol{t}_i \right] + \mathbb{E}\left[ \sum_{j=n+1}^{2n-1} \sum_{l=1}^{j-1} \left( \prod_{p=1}^{l-1} (\boldsymbol{I} - \widetilde{\boldsymbol{S}}_p) \right) \widetilde{\boldsymbol{S}}_l \widetilde{\boldsymbol{S}}_j \left( \sum_{i=1}^{2n-j} \boldsymbol{t}_i \right) \right] \right.
$$

$$
\left. + \mathbb{E}\left[ \sum_{j=1}^{n-1} \sum_{l=1}^{j-1} \left( \prod_{p=1}^{l-1} (\boldsymbol{I} - \boldsymbol{S}_p) \right) \boldsymbol{S}_l \boldsymbol{S}_j \left( \sum_{i=1}^{n-j} \boldsymbol{t}_{n+1-i} \right) \right] \right\|
$$

$$
\leq \sum_{i=1}^{n} \mathbb{E}\left[ \|\boldsymbol{S}_i\| \|\boldsymbol{t}_i\| \right] + \sum_{j=n+1}^{2n-1} \sum_{l=1}^{j-1} \left\| \mathbb{E}\left[ \left( \prod_{p=1}^{l-1} (\boldsymbol{I} - \widetilde{\boldsymbol{S}}_p) \right) \widetilde{\boldsymbol{S}}_l \widetilde{\boldsymbol{S}}_j \left( \sum_{i=1}^{2n-j} \boldsymbol{t}_i \right) \right] \right\|
$$

$$
+ \sum_{j=1}^{n-1} \sum_{l=1}^{j-1} \left\| \mathbb{E}\left[ \left( \prod_{p=1}^{l-1} (\boldsymbol{I} - \boldsymbol{S}_p) \right) \boldsymbol{S}_l \boldsymbol{S}_j \left( \sum_{i=1}^{n-j} \boldsymbol{t}_{n+1-i} \right) \right] \right\|. \tag{32}
$$

Define $G^* := \max_i \|\boldsymbol{b}_i\|$. Then $\|\boldsymbol{S}_i\| \|\boldsymbol{t}_i\| = \|\alpha \boldsymbol{A}_{\sigma_i^k}\| \|\alpha \boldsymbol{b}_{\sigma_i^k}\| \leq \alpha^2 L G^*$ and hence

$$
\sum_{i=1}^{n} \mathbb{E}\left[ \|\boldsymbol{S}_i\| \|\boldsymbol{t}_i\| \right] \leq n\alpha^2 L G^*. \tag{33}
$$

Next we bound the other two terms. Using Eq. (23), we get that for any $l < j$,

$$
\mathbb{E}\left[ \left( \prod_{p=1}^{l-1} (\boldsymbol{I} - \boldsymbol{S}_p) \right) \boldsymbol{S}_l \boldsymbol{S}_j \left( \sum_{i=1}^{n-j} \boldsymbol{t}_{n+1-i} \right) \right]
$$

$$
= \mathbb{E}\left[ \left( \boldsymbol{I} - \sum_{p=1}^{l-1} \boldsymbol{S}_p + \sum_{p=1}^{l-1} \sum_{q=1}^{p-1} \left( \prod_{r=1}^{q-1} (\boldsymbol{I} - \boldsymbol{S}_q) \right) \boldsymbol{S}_r \boldsymbol{S}_p \right) \boldsymbol{S}_l \boldsymbol{S}_j \left( \sum_{i=1}^{n-j} \boldsymbol{t}_{n+1-i} \right) \right]
$$

$$= \sum_{i=1}^{n-j} \mathbb{E}\left[\boldsymbol{S}_l \boldsymbol{S}_j \boldsymbol{t}_{n+1-i}\right] - \mathbb{E}\left[\sum_{p=1}^{l-1} \boldsymbol{S}_p \boldsymbol{S}_l \boldsymbol{S}_j \left(\sum_{i=1}^{n-j} \boldsymbol{t}_{n+1-i}\right)\right]$$

$$+ \mathbb{E}\left[\sum_{p=1}^{l-1}\sum_{q=1}^{p-1}\left(\prod_{r=1}^{q-1}(\boldsymbol{I}-\boldsymbol{S}_q)\right)\boldsymbol{S}_r \boldsymbol{S}_p \boldsymbol{S}_l \boldsymbol{S}_j \left(\sum_{i=1}^{n-j} \boldsymbol{t}_{n+1-i}\right)\right]$$

$$= \sum_{i>j,l} \mathbb{E}\left[\boldsymbol{S}_l \boldsymbol{S}_j \boldsymbol{t}_i\right] - \sum_{p<l,j<i} \mathbb{E}\left[\boldsymbol{S}_p \boldsymbol{S}_l \boldsymbol{S}_j \boldsymbol{t}_i\right] + \sum_{p=1}^{l-1}\sum_{q=1}^{p-1} \mathbb{E}\left[\left(\prod_{r=1}^{q-1}(\boldsymbol{I}-\boldsymbol{S}_q)\right)\boldsymbol{S}_r \boldsymbol{S}_p \boldsymbol{S}_l \boldsymbol{S}_j \left(\sum_{i=1}^{n-j}\boldsymbol{t}_{n+1-i}\right)\right]$$

$$= \sum_{i>j,l} \mathbb{E}\left[\boldsymbol{S}_l \boldsymbol{S}_j \mathbb{E}[\boldsymbol{t}_i|\boldsymbol{S}_l,\boldsymbol{S}_j]\right] - \sum_{p<l,j<i} \mathbb{E}\left[\boldsymbol{S}_p \boldsymbol{S}_l \boldsymbol{S}_j \mathbb{E}[\boldsymbol{t}_i|\boldsymbol{S}_l,\boldsymbol{S}_j,\boldsymbol{S}_p]\right]$$

$$+ \sum_{p=1}^{l-1}\sum_{q=1}^{p-1} \mathbb{E}\left[\left(\prod_{r=1}^{q-1}(\boldsymbol{I}-\boldsymbol{S}_q)\right)\boldsymbol{S}_r \boldsymbol{S}_p \boldsymbol{S}_l \boldsymbol{S}_j \left(\sum_{i=1}^{n-j}\boldsymbol{t}_{n+1-i}\right)\right].$$

Since $\sum_{i=1}^{n} \boldsymbol{t}_i = 0$, and we use uniform random permutations, $\mathbb{E}[\boldsymbol{t}_i|\boldsymbol{S}_l,\boldsymbol{S}_j] = \sum_{\substack{\boldsymbol{t}_i \neq \boldsymbol{t}_l, \\ \boldsymbol{t}_i \neq \boldsymbol{t}_j}} \frac{\boldsymbol{t}_i}{n-2} = \frac{-\boldsymbol{t}_j - \boldsymbol{t}_l}{n-2}$. Similarly, $\mathbb{E}[\boldsymbol{t}_i|\boldsymbol{S}_l,\boldsymbol{S}_j,\boldsymbol{S}_p] = \frac{-\boldsymbol{t}_j - \boldsymbol{t}_l - \boldsymbol{t}_p}{n-3}$. Hence,

$$\left\|\mathbb{E}\left[\left(\prod_{p=1}^{l-1}(\boldsymbol{I}-\boldsymbol{S}_p)\right)\boldsymbol{S}_l \boldsymbol{S}_j \left(\sum_{i=1}^{n-j}\boldsymbol{t}_{n+1-i}\right)\right]\right\|$$

$$\leq \sum_{i>j,l} \left\|\mathbb{E}\left[\boldsymbol{S}_l \boldsymbol{S}_j \mathbb{E}[\boldsymbol{t}_i|\boldsymbol{S}_l,\boldsymbol{S}_j]\right]\right\| + \sum_{p<l,j<i} \left\|\mathbb{E}\left[\boldsymbol{S}_p \boldsymbol{S}_l \boldsymbol{S}_j \mathbb{E}[\boldsymbol{t}_i|\boldsymbol{S}_l,\boldsymbol{S}_j,\boldsymbol{S}_p]\right]\right\|$$

$$+ \sum_{p=1}^{l-1}\sum_{q=1}^{p-1} \mathbb{E}\left[\left(\prod_{r=1}^{q-1}\|\boldsymbol{I}-\boldsymbol{S}_q\|\right)\|\boldsymbol{S}_r\|\|\boldsymbol{S}_p\|\|\boldsymbol{S}_l\|\|\boldsymbol{S}_j\|\left\|\sum_{i=1}^{n-j}\boldsymbol{t}_{n+1-i}\right\|\right]$$

$$\leq \sum_{i>j,l} \mathbb{E}\left[\|\boldsymbol{S}_l\|\|\boldsymbol{S}_j\|\frac{\|\boldsymbol{t}_l + \boldsymbol{t}_j\|}{n-2}\right] + \sum_{p<l,j<i} \mathbb{E}\left[\|\boldsymbol{S}_p\|\|\boldsymbol{S}_l\|\|\boldsymbol{S}_j\|\frac{\|\boldsymbol{t}_l + \boldsymbol{t}_j + \boldsymbol{t}_p\|}{n-3}\right]$$

$$+ \sum_{p=1}^{l-1}\sum_{q=1}^{p-1} \mathbb{E}\left[\left(\prod_{r=1}^{q-1}\|\boldsymbol{I}-\boldsymbol{S}_q\|\right)\|\boldsymbol{S}_r\|\|\boldsymbol{S}_p\|\|\boldsymbol{S}_l\|\|\boldsymbol{S}_j\|\left\|\sum_{i=1}^{n-j}\boldsymbol{t}_{n+1-i}\right\|\right]$$

$$\leq \frac{2n}{n-2}\alpha^3 L^2 G^* + \frac{3n^2}{2(n-3)}\alpha^4 L^3 G^* + \alpha^4 L^4 \sum_{p=1}^{l-1}\sum_{q=1}^{p-1} \mathbb{E}\left[\left\|\sum_{i=1}^{n-j}\boldsymbol{t}_{n+1-i}\right\|\right]$$

$$\text{(Since } \|S_i\| \leq \alpha L, \|t_i\| \leq \alpha G^*\text{)}$$

$$\leq 4\alpha^3 L^2 G^* + 3n\alpha^4 L^3 G^* + \alpha^5 L^4 n^2 G^* \sqrt{18n \log n}, \tag{34}$$

where we used Lemma 6 and the assumption that $n > 6$ in the last step.

The third term in Ineq. (32) can be handled similarly. For any $l, j$:

$$\mathbb{E}\left[\left(\prod_{p=1}^{l-1}(\boldsymbol{I}-\widetilde{\boldsymbol{S}}_p)\right)\widetilde{\boldsymbol{S}}_l \widetilde{\boldsymbol{S}}_j \left(\sum_{i=1}^{2n-j}\boldsymbol{t}_i\right)\right]$$

$$= \mathbb{E}\left[\left(\boldsymbol{I} - \sum_{p=1}^{l-1}\widetilde{\boldsymbol{S}}_p + \sum_{p=1}^{l-1}\sum_{q=1}^{p-1}\left(\prod_{r=1}^{q-1}(\boldsymbol{I}-\widetilde{\boldsymbol{S}}_q)\right)\widetilde{\boldsymbol{S}}_r \widetilde{\boldsymbol{S}}_p\right)\widetilde{\boldsymbol{S}}_l \widetilde{\boldsymbol{S}}_j \left(\sum_{i=1}^{2n-j}\boldsymbol{t}_i\right)\right]$$

$$= \sum_{i=1}^{2n-j} \mathbb{E}\left[\widetilde{\boldsymbol{S}}_l \widetilde{\boldsymbol{S}}_j \boldsymbol{t}_i\right] - \sum_{p=1}^{l-1}\sum_{i=1}^{2n-j} \mathbb{E}\left[\widetilde{\boldsymbol{S}}_p \widetilde{\boldsymbol{S}}_l \widetilde{\boldsymbol{S}}_j \boldsymbol{t}_i\right] + \mathbb{E}\left[\sum_{p=1}^{l-1}\sum_{q=1}^{p-1}\left(\prod_{r=1}^{q-1}(\boldsymbol{I}-\widetilde{\boldsymbol{S}}_q)\right)\widetilde{\boldsymbol{S}}_r \widetilde{\boldsymbol{S}}_p \widetilde{\boldsymbol{S}}_l \widetilde{\boldsymbol{S}}_j \left(\sum_{i=1}^{2n-j}\boldsymbol{t}_i\right)\right].$$

Now, it is easy to see that $i \neq j$ and $i \neq 2n - j + 1$. Then, if $i = l$ or $i = 2n - l + 1$ we use for that case the fact that $\|\mathbb{E}[\widetilde{\boldsymbol{S}}_l \widetilde{\boldsymbol{S}}_j \boldsymbol{t}_i]\| \leq \alpha^3 L^2 G^*$. For all other $i$, we can again use that $\mathbb{E}[\boldsymbol{t}_i|\widetilde{\boldsymbol{S}}_l,\widetilde{\boldsymbol{S}}_j] = \frac{-\boldsymbol{t}_l - \boldsymbol{t}_j}{n-2}$ if $l \leq n$ or $\mathbb{E}[\boldsymbol{t}_i|\widetilde{\boldsymbol{S}}_l,\widetilde{\boldsymbol{S}}_j] = \frac{-\boldsymbol{t}_{2n-l+1} - \boldsymbol{t}_j}{n-2}$ otherwise. Similarly, we can bound

$\left\|\mathbb{E}\left[\widetilde{\boldsymbol{S}}_p\widetilde{\boldsymbol{S}}_l\widetilde{\boldsymbol{S}}_j\boldsymbol{t}_i\right]\right\|$. Proceeding in a way similar to how we derived Ineq. (34), we get

$$
\left\|\mathbb{E}\left[\left(\prod_{p=1}^{l-1}(\boldsymbol{I}-\widetilde{\boldsymbol{S}}_p)\right)\widetilde{\boldsymbol{S}}_l\widetilde{\boldsymbol{S}}_j\left(\sum_{i=1}^{2n-j}\boldsymbol{t}_i\right)\right]\right\|
$$

$$
\leq \sum_{i=1}^{2n-j}\left\|\mathbb{E}\left[\widetilde{\boldsymbol{S}}_l\widetilde{\boldsymbol{S}}_j\boldsymbol{t}_i\right]\right\| + \sum_{p=1}^{l-1}\sum_{i=1}^{2n-j}\left\|\mathbb{E}\left[\widetilde{\boldsymbol{S}}_p\widetilde{\boldsymbol{S}}_l\widetilde{\boldsymbol{S}}_j\boldsymbol{t}_i\right]\right\|
$$

$$
+ \sum_{p=1}^{l-1}\sum_{q=1}^{p-1}\left\|\mathbb{E}\left[\left(\prod_{r=1}^{q-1}(\boldsymbol{I}-\widetilde{\boldsymbol{S}}_q)\right)\widetilde{\boldsymbol{S}}_r\widetilde{\boldsymbol{S}}_p\widetilde{\boldsymbol{S}}_l\widetilde{\boldsymbol{S}}_j\left(\sum_{i=1}^{2n-j}\boldsymbol{t}_i\right)\right]\right\|
$$

$$
\leq \alpha^3 L^2 G^* + \frac{2n}{n-2}\alpha^3 L^2 G^* + \alpha^4 L^3 G^* + n\alpha^4 L^3 G^* + \frac{3n^2}{2(n-3)}\alpha^4 L^3 G^*
$$

$$
+ \sum_{p=1}^{l-1}\sum_{q=1}^{p-1}\left\|\mathbb{E}\left[\left(\prod_{r=1}^{q-1}(\boldsymbol{I}-\widetilde{\boldsymbol{S}}_q)\right)\widetilde{\boldsymbol{S}}_r\widetilde{\boldsymbol{S}}_p\widetilde{\boldsymbol{S}}_l\widetilde{\boldsymbol{S}}_j\left(\sum_{i=1}^{2n-j}\boldsymbol{t}_i\right)\right]\right\|
$$

$$
\leq 5\alpha^3 L^2 G^* + 5n\alpha^4 L^3 G^*
$$

$$
+ \sum_{p=1}^{l-1}\sum_{q=1}^{p-1}\left\|\mathbb{E}\left[\left(\prod_{r=1}^{q-1}\|\boldsymbol{I}-\widetilde{\boldsymbol{S}}_q\|\right)\|\widetilde{\boldsymbol{S}}_r\|\|\widetilde{\boldsymbol{S}}_p\|\|\widetilde{\boldsymbol{S}}_l\|\|\widetilde{\boldsymbol{S}}_j\|\left(\sum_{i=1}^{2n-j}\boldsymbol{t}_i\right)\right]\right\|
$$

$$
\leq 5\alpha^3 L^2 G^* + 5n\alpha^4 L^3 G^* + \alpha^5 L^4 n^2 G^*\sqrt{18n\log n}. \tag{35}
$$

Substituting Ineq. (33), (34) and (35) into (32), we get

$$
\|\mathbb{E}[\boldsymbol{z}]\| \leq n\alpha^2 L G^* + 10n^2\alpha^3 L^2 G^* + 10n^3\alpha^4 L^3 G^* + 2\alpha^5 L^4 n^4 G^*\sqrt{18n\log n}
$$

$$
+ 4n^2\alpha^3 L^2 G^* + 3n^3\alpha^4 L^3 G^* + \alpha^5 L^4 n^4 G^*\sqrt{18n\log n}
$$

$$
\leq 28n\alpha^2 L G^* + 9\alpha^5 L^4 n^4 G^*\sqrt{2n\log n},
$$

where we used the assumption that $\alpha \leq \frac{1}{nL}$ in the last step.

## F.4 PROOF OF LEMMA 11

Define the matrix $\boldsymbol{M}$ as

$$
\boldsymbol{M} := \left(\prod_{l=0}^{k'-1}\left(\prod_{i=1}^{n}(\boldsymbol{I}-\boldsymbol{S}_i^{\frac{K}{2}-l})\right)\left(\prod_{i=1}^{n}(\boldsymbol{I}-\boldsymbol{S}_{n-i+1}^{\frac{K}{2}-l})\right)\right)^{\top}\left(\prod_{l=0}^{k'-1}\left(\prod_{i=1}^{n}(\boldsymbol{I}-\boldsymbol{S}_i^{\frac{K}{2}-l})\right)\left(\prod_{i=1}^{n}(\boldsymbol{I}-\boldsymbol{S}_{n-i+1}^{\frac{K}{2}-l})\right)\right).
$$

Since $\boldsymbol{M}$ is independent of $\left(\prod_{i=1}^{n}(\boldsymbol{I}-\boldsymbol{S}_i^{\frac{K}{2}-k'})\right)\left(\prod_{i=1}^{n}(\boldsymbol{I}-\boldsymbol{S}_{n-i+1}^{\frac{K}{2}-k'})\right)$ and $\boldsymbol{z}^{\frac{K}{2}-k'}$, using the tower rule, we get

$$
\mathbb{E}\left[\left(\prod_{l=0}^{k'}\left(\prod_{i=1}^{n}(\boldsymbol{I}-\boldsymbol{S}_i^{\frac{K}{2}-l})\right)\left(\prod_{i=1}^{n}(\boldsymbol{I}-\boldsymbol{S}_{n-i+1}^{\frac{K}{2}-l})\right)\right)^{\top}\right.
$$

$$
\left.\left(\prod_{l=0}^{k'-1}\left(\prod_{i=1}^{n}(\boldsymbol{I}-\boldsymbol{S}_i^{\frac{K}{2}-l})\right)\left(\prod_{i=1}^{n}(\boldsymbol{I}-\boldsymbol{S}_{n-i+1}^{\frac{K}{2}-l})\right)\right)\boldsymbol{z}^{\frac{K}{2}-k'}\right]
$$

$$
= \mathbb{E}\left[\left(\left(\prod_{i=1}^{n}(\boldsymbol{I}-\boldsymbol{S}_i^{\frac{K}{2}-k'})\right)\left(\prod_{i=1}^{n}(\boldsymbol{I}-\boldsymbol{S}_{n-i+1}^{\frac{K}{2}-k'})\right)\right)^{\top}\mathbb{E}[\boldsymbol{M}]\boldsymbol{z}^{\frac{K}{2}-k'}\right].
$$

We will now drop the superscript $\frac{K}{2}-k'$ for convenience. Hence, we need to control the following term:

$$
\mathbb{E}\left[\left(\left(\prod_{i=1}^{n}(\boldsymbol{I}-\boldsymbol{S}_i)\right)\left(\prod_{i=1}^{n}(\boldsymbol{I}-\boldsymbol{S}_{n-i+1})\right)\right)^{\top}\mathbb{E}[\boldsymbol{M}]\boldsymbol{z}\right]
$$

We define $(\widetilde{\boldsymbol{S}}_1, \ldots, \widetilde{\boldsymbol{S}}_{2n})$ as $(\widetilde{\boldsymbol{S}}_1, \ldots, \widetilde{\boldsymbol{S}}_n) = (\boldsymbol{S}_1, \ldots, \boldsymbol{S}_n)$ and $(\widetilde{\boldsymbol{S}}_{n+1}, \ldots, \widetilde{\boldsymbol{S}}_{2n}) = (\boldsymbol{S}_n, \ldots, \boldsymbol{S}_1)$. Then, we use Eq. (21) to get

$$\left( \left( \prod_{i=1}^{n}(\boldsymbol{I} - \boldsymbol{S}_i) \right) \left( \prod_{i=1}^{n}(\boldsymbol{I} - \boldsymbol{S}_{n-i+1}) \right) \right) = \prod_{i=1}^{2n}(\boldsymbol{I} - \widetilde{\boldsymbol{S}}_i)$$

$$= \boldsymbol{I} - \sum_{j=1}^{2n} \widetilde{\boldsymbol{S}}_j + \sum_{j=1}^{2n}\sum_{l=1}^{j-1} \left( \prod_{p=1}^{l-1}(\boldsymbol{I} - \widetilde{\boldsymbol{S}}_p) \right) \widetilde{\boldsymbol{S}}_l \widetilde{\boldsymbol{S}}_j.$$

Note that $\boldsymbol{I} - \sum_{j=1}^{2n} \widetilde{\boldsymbol{S}}_j$ is a constant matrix. Since $\widetilde{\boldsymbol{S}}_j = \alpha \boldsymbol{A}_{\sigma_j}$, we have that $\|\widetilde{\boldsymbol{S}}_j\| \leq \alpha L$ by Assumption 2. Hence, $\alpha \leq \frac{1}{2nL}$ then $\|\boldsymbol{I} - \sum_{j=1}^{2n} \widetilde{\boldsymbol{S}}_j\| \leq 1$. Further, $\alpha \leq 1/L$ implies that $\|\boldsymbol{I} - \boldsymbol{S}_i\| \leq 1$, which implies $\|\boldsymbol{M}\| \leq 1$. Hence,

$$\left\| \mathbb{E}\left[ \left( \left( \prod_{i=1}^{n}(\boldsymbol{I} - \boldsymbol{S}_i) \right) \left( \prod_{i=1}^{n}(\boldsymbol{I} - \boldsymbol{S}_{n-i+1}) \right) \right)^{\top} \mathbb{E}[\boldsymbol{M}]\boldsymbol{z} \right] \right\|$$

$$\leq \left\| \mathbb{E}\left[ \left( \boldsymbol{I} - \sum_{j=1}^{2n} \widetilde{\boldsymbol{S}}_j \right)^{\top} \mathbb{E}[\boldsymbol{M}]\boldsymbol{z} \right] \right\|$$

$$+ \left\| \mathbb{E}\left[ \left( \sum_{j=1}^{2n}\sum_{l=1}^{j-1} \left( \prod_{p=1}^{l-1}(\boldsymbol{I} - \widetilde{\boldsymbol{S}}_p) \right) \widetilde{\boldsymbol{S}}_l \widetilde{\boldsymbol{S}}_j \right)^{\top} \mathbb{E}[\boldsymbol{M}]\boldsymbol{z} \right] \right\|$$

$$= \left\| \left( \boldsymbol{I} - \sum_{j=1}^{2n} \widetilde{\boldsymbol{S}}_j \right)^{\top} \mathbb{E}[\boldsymbol{M}]\mathbb{E}[\boldsymbol{z}] \right\| + \left\| \mathbb{E}\left[ \left( \sum_{j=1}^{2n}\sum_{l=1}^{j-1} \left( \prod_{p=1}^{l-1}(\boldsymbol{I} - \widetilde{\boldsymbol{S}}_p) \right) \widetilde{\boldsymbol{S}}_l \widetilde{\boldsymbol{S}}_j \right)^{\top} \mathbb{E}[\boldsymbol{M}]\boldsymbol{z} \right] \right\|$$

$$\leq \|\mathbb{E}[\boldsymbol{z}]\| + \left\| \mathbb{E}\left[ \left( \sum_{j=1}^{2n}\sum_{l=1}^{j-1} \left( \prod_{p=1}^{l-1}(\boldsymbol{I} - \widetilde{\boldsymbol{S}}_p) \right) \widetilde{\boldsymbol{S}}_l \widetilde{\boldsymbol{S}}_j \right)^{\top} \mathbb{E}[\boldsymbol{M}]\boldsymbol{z} \right] \right\|.$$

We can apply Lemma 10 to bound $\|\mathbb{E}[\boldsymbol{z}]\|$. So, we focus on the other term. Using Lemma 5,

$$\left\| \mathbb{E}\left[ \left( \sum_{j=1}^{2n}\sum_{l=1}^{j-1} \left( \prod_{p=1}^{l-1}(\boldsymbol{I} - \widetilde{\boldsymbol{S}}_p) \right) \widetilde{\boldsymbol{S}}_l \widetilde{\boldsymbol{S}}_j \right)^{\top} \mathbb{E}[\boldsymbol{M}]\boldsymbol{z} \right] \right\|$$

$$\leq \left\| \mathbb{E}\left[ \left( \sum_{j=1}^{2n}\sum_{l=1}^{j-1} \left( \prod_{p=1}^{l-1}(\boldsymbol{I} - \widetilde{\boldsymbol{S}}_p) \right) \widetilde{\boldsymbol{S}}_l \widetilde{\boldsymbol{S}}_j \right)^{\top} \mathbb{E}[\boldsymbol{M}] \left( \sum_{i=1}^{n} \boldsymbol{S}_i \boldsymbol{t}_i \right) \right] \right\|$$

$$+ \left\| \mathbb{E}\left[ \left( \sum_{j=1}^{2n}\sum_{l=1}^{j-1} \left( \prod_{p=1}^{l-1}(\boldsymbol{I} - \widetilde{\boldsymbol{S}}_p) \right) \widetilde{\boldsymbol{S}}_l \widetilde{\boldsymbol{S}}_j \right)^{\top} \mathbb{E}[\boldsymbol{M}] \left( \sum_{j=n+1}^{2n-1}\sum_{l=1}^{j-1} \left( \prod_{p=1}^{l-1}(\boldsymbol{I} - \widetilde{\boldsymbol{S}}_p) \right) \widetilde{\boldsymbol{S}}_l \widetilde{\boldsymbol{S}}_j \left( \sum_{i=1}^{2n-j} \boldsymbol{t}_i \right) \right) \right] \right\|$$

$$+ \left\| \mathbb{E}\left[ \left( \sum_{j=1}^{2n}\sum_{l=1}^{j-1} \left( \prod_{p=1}^{l-1}(\boldsymbol{I} - \widetilde{\boldsymbol{S}}_p) \right) \widetilde{\boldsymbol{S}}_l \widetilde{\boldsymbol{S}}_j \right)^{\top} \mathbb{E}[\boldsymbol{M}] \left( \sum_{j=1}^{n-1}\sum_{l=1}^{j-1} \left( \prod_{p=1}^{l-1}(\boldsymbol{I} - \boldsymbol{S}_p) \right) \boldsymbol{S}_l \boldsymbol{S}_j \left( \sum_{i=1}^{n-j} \boldsymbol{t}_{n+1-i} \right) \right) \right] \right\|$$

$$= \left\| \mathbb{E}\left[ \left( \sum_{j=1}^{2n}\sum_{l=1}^{j-1} \left( \prod_{p=1}^{l-1}(\boldsymbol{I} - \widetilde{\boldsymbol{S}}_p) \right) \widetilde{\boldsymbol{S}}_l \widetilde{\boldsymbol{S}}_j \right)^{\top} \right] \mathbb{E}[\boldsymbol{M}] \left( \sum_{i=1}^{n} \boldsymbol{S}_i \boldsymbol{t}_i \right) \right\|$$

$$+ \left\| \mathbb{E}\left[ \left( \sum_{j=1}^{2n}\sum_{l=1}^{j-1} \left( \prod_{p=1}^{l-1}(\boldsymbol{I} - \widetilde{\boldsymbol{S}}_p) \right) \widetilde{\boldsymbol{S}}_l \widetilde{\boldsymbol{S}}_j \right)^{\top} \mathbb{E}[\boldsymbol{M}] \left( \sum_{j=n+1}^{2n-1}\sum_{l=1}^{j-1} \left( \prod_{p=1}^{l-1}(\boldsymbol{I} - \widetilde{\boldsymbol{S}}_p) \right) \widetilde{\boldsymbol{S}}_l \widetilde{\boldsymbol{S}}_j \left( \sum_{i=1}^{2n-j} \boldsymbol{t}_i \right) \right) \right] \right\|$$

$$+ \left\| \mathbb{E}\left[ \left( \sum_{j=1}^{2n}\sum_{l=1}^{j-1} \left(\prod_{p=1}^{l-1}(\boldsymbol{I}-\widetilde{\boldsymbol{S}}_p)\right) \widetilde{\boldsymbol{S}}_l\widetilde{\boldsymbol{S}}_j \right)^{\top} \mathbb{E}[\boldsymbol{M}] \left( \sum_{j=1}^{n-1}\sum_{l=1}^{j-1} \left(\prod_{p=1}^{l-1}(\boldsymbol{I}-\boldsymbol{S}_p)\right) \boldsymbol{S}_l\boldsymbol{S}_j \left(\sum_{i=1}^{n-j}\boldsymbol{t}_{n+1-i}\right) \right) \right] \right\|$$

$$\leq \mathbb{E}\left[ \sum_{j=1}^{2n}\sum_{l=1}^{j-1} \left(\prod_{p=1}^{l-1}\|\boldsymbol{I}-\widetilde{\boldsymbol{S}}_p\|\right) \|\widetilde{\boldsymbol{S}}_l\|\|\widetilde{\boldsymbol{S}}_j\| \right] \mathbb{E}[\|\boldsymbol{M}\|] \left\| \sum_{i=1}^{n}\boldsymbol{S}_i\boldsymbol{t}_i \right\|$$

$$+ \mathbb{E}\left[ \left( \sum_{j=1}^{2n}\sum_{l=1}^{j-1} \left(\prod_{p=1}^{l-1}\|\boldsymbol{I}-\widetilde{\boldsymbol{S}}_p\|\right) \|\widetilde{\boldsymbol{S}}_l\|\|\widetilde{\boldsymbol{S}}_j\| \right) \mathbb{E}[\|\boldsymbol{M}\|] \left( \sum_{j=n+1}^{2n-1}\sum_{l=1}^{j-1} \left(\prod_{p=1}^{l-1}\|\boldsymbol{I}-\widetilde{\boldsymbol{S}}_p\|\right) \|\widetilde{\boldsymbol{S}}_l\|\|\widetilde{\boldsymbol{S}}_j\| \left\| \sum_{i=1}^{2n-j}\boldsymbol{t}_i \right\| \right) \right]$$

$$+ \mathbb{E}\left[ \left( \sum_{j=1}^{2n}\sum_{l=1}^{j-1} \left(\prod_{p=1}^{l-1}\|\boldsymbol{I}-\widetilde{\boldsymbol{S}}_p\|\right) \|\widetilde{\boldsymbol{S}}_l\|\|\widetilde{\boldsymbol{S}}_j\| \right) \mathbb{E}[\|\boldsymbol{M}\|] \left( \sum_{j=1}^{n-1}\sum_{l=1}^{j-1} \left(\prod_{p=1}^{l-1}\|\boldsymbol{I}-\boldsymbol{S}_p\|\right) \|\boldsymbol{S}_l\|\|\boldsymbol{S}_j\| \left\| \sum_{i=1}^{n-j}\boldsymbol{t}_{n+1-i} \right\| \right) \right].$$

Now, we use that $\|\boldsymbol{M}\| \leq 1$, $\|\boldsymbol{I}-\widetilde{\boldsymbol{S}}_i\| \leq 1$, $\|\widetilde{\boldsymbol{S}}_i\| \leq \alpha L$ and $\|\widetilde{\boldsymbol{t}}_i\| \leq \alpha G^*$:

$$\left\| \mathbb{E}\left[ \left( \sum_{j=1}^{2n}\sum_{l=1}^{j-1} \left(\prod_{p=1}^{l-1}(\boldsymbol{I}-\widetilde{\boldsymbol{S}}_p)\right) \widetilde{\boldsymbol{S}}_l\widetilde{\boldsymbol{S}}_j \right)^{\top} \mathbb{E}[\boldsymbol{M}]\boldsymbol{z} \right] \right\|$$

$$\leq 4n^3\alpha^4 L^3 G^* + 4n^4\alpha^4 L^4 \mathbb{E}\left[ \left\| \sum_{i=1}^{2n-j}\boldsymbol{t}_i \right\| \right] + n^4\alpha^4 L^4 \mathbb{E}\left[ \left\| \sum_{i=1}^{n-j}\boldsymbol{t}_{n+1-i} \right\| \right].$$

Using Lemma 6, we get

$$\left\| \mathbb{E}\left[ \left( \sum_{j=1}^{2n}\sum_{l=1}^{j-1} \left(\prod_{p=1}^{l-1}(\boldsymbol{I}-\widetilde{\boldsymbol{S}}_p)\right) \widetilde{\boldsymbol{S}}_l\widetilde{\boldsymbol{S}}_j \right)^{\top} \mathbb{E}[\boldsymbol{M}]\boldsymbol{z} \right] \right\| \leq 4n^3\alpha^4 L^3 G^* + 15n^4\alpha^5 L^4 G^* \sqrt{2n\log n}.$$

Putting everything together,

$$\left\| \mathbb{E}\left[ \left( \prod_{l=0}^{k'} \left(\prod_{i=1}^{n}(\boldsymbol{I}-\boldsymbol{S}_i^{\frac{K}{2}-l})\right) \left(\prod_{i=1}^{n}(\boldsymbol{I}-\boldsymbol{S}_{n-i+1}^{\frac{K}{2}-l})\right) \right)^{\top} \right. \right.$$
$$\left. \left. \left( \prod_{l=0}^{k'-1} \left(\prod_{i=1}^{n}(\boldsymbol{I}-\boldsymbol{S}_i^{\frac{K}{2}-l})\right) \left(\prod_{i=1}^{n}(\boldsymbol{I}-\boldsymbol{S}_{n-i+1}^{\frac{K}{2}-l})\right) \right) \boldsymbol{z}^{\frac{K}{2}-k'} \right] \right\|$$
$$\leq (28n\alpha^2 L G^* + 9\alpha^5 L^4 n^4 G^* \sqrt{2n\log n}) + (4n^3\alpha^4 L^3 G^* + 15n^4\alpha^5 L^4 G^* \sqrt{2n\log n})$$
$$\leq 32n\alpha^2 L G^* + 24\alpha^5 L^4 n^4 G^* \sqrt{2n\log n}.$$

## G  PROOF OF THEOREM 6

*Proof.* We start off by defining the error term

$$\boldsymbol{r}^k = \left( \sum_{i=1}^{n}\nabla f_i\left(\boldsymbol{x}_{i-1}^{2k-1}\right) - \sum_{i=1}^{n}\nabla f_i\left(\boldsymbol{x}_0^{2k-1}\right) \right) + \left( \sum_{i=1}^{n}\nabla f_{n-i+1}\left(\boldsymbol{x}_{i-1}^{2k}\right) - \sum_{i=1}^{n}\nabla f_{n-i+1}\left(\boldsymbol{x}_0^{2k-1}\right) \right),$$

where $k \in [K/2]$. This captures the difference between true gradients that the algorithms observes, and the gradients that a full step of gradient descent would have seen.

For two consecutive epochs of FLIPFLOP WITH INCREMENTAL GD, we have the following inequality

$$\|\boldsymbol{x}_n^{2k} - \boldsymbol{x}^*\|^2 = \|\boldsymbol{x}_0^{2k-1} - \boldsymbol{x}^*\|^2 - 2\alpha \left\langle \boldsymbol{x}_0^{2k-1} - \boldsymbol{x}^*, \sum_{i=1}^{n}\nabla f_i\left(x_{i-1}^{2k-1}\right) + \sum_{i=1}^{n}\nabla f_{n-i+1}\left(x_{i-1}^{2k}\right) \right\rangle$$

$$+ \alpha^2 \left\| \sum_{i=1}^{n} \nabla f_i \left( x_{i-1}^{2k-1} \right) + \sum_{i=1}^{n} \nabla f_{n-i+1} \left( x_{i-1}^{2k} \right) \right\|^2$$

$$= \| \boldsymbol{x}_0^{2k-1} - \boldsymbol{x}^* \|^2 - 2\alpha \left\langle \boldsymbol{x}_0^{2k-1} - \boldsymbol{x}^*, 2n\nabla F(\boldsymbol{x}_0^{2k-1}) \right\rangle$$

$$- 2\alpha \left\langle \boldsymbol{x}_0^{2k-1} - \boldsymbol{x}^*, \boldsymbol{r}^k \right\rangle + \alpha^2 \left\| 2n\nabla F(\boldsymbol{x}_0^{2k-1}) + \boldsymbol{r}^k \right\|^2$$

$$\le \| \boldsymbol{x}_0^{2k-1} - \boldsymbol{x}^* \|^2 - 4n\alpha \left[ \frac{L\mu}{L+\mu} \| \boldsymbol{x}_0^{2k-1} - \boldsymbol{x}^* \|^2 + \frac{1}{L+\mu} \left\| \nabla F \left( \boldsymbol{x}_0^{2k-1} \right) \right\|^2 \right]$$

$$- 2\alpha \left\langle \boldsymbol{x}_0^{2k-1} - \boldsymbol{x}^*, \boldsymbol{r}^k \right\rangle + \alpha^2 \left\| 2n\nabla F(\boldsymbol{x}_0^{2k-1}) + \boldsymbol{r}^k \right\|^2$$

$$\le \| \boldsymbol{x}_0^{2k-1} - \boldsymbol{x}^* \|^2 - 4n\alpha \left[ \frac{L\mu}{L+\mu} \| \boldsymbol{x}_0^{2k-1} - \boldsymbol{x}^* \|^2 + \frac{1}{L+\mu} \left\| \nabla F \left( \boldsymbol{x}_0^{2k-1} \right) \right\|^2 \right]$$

$$- 2\alpha \left\langle \boldsymbol{x}_0^{2k-1} - \boldsymbol{x}^*, \boldsymbol{r}^k \right\rangle + 8\alpha^2 n^2 \left\| \nabla F(\boldsymbol{x}_0^{2k-1}) \right\|^2 + 2\alpha^2 \left\| \boldsymbol{r}^k \right\|^2$$

$$= \left( 1 - 4n\alpha \frac{L\mu}{L+\mu} \right) \| \boldsymbol{x}_0^{2k-1} - \boldsymbol{x}^* \|^2 - \left( 4n\alpha \frac{1}{L+\mu} - 8\alpha^2 n^2 \right) \left\| \nabla F(\boldsymbol{x}_0^{2k-1}) \right\|^2$$

$$- 2\alpha \left\langle \boldsymbol{x}_0^{2k-1} - \boldsymbol{x}^*, \boldsymbol{r}^k \right\rangle + 2\alpha^2 \left\| \boldsymbol{r}^k \right\|^2, \tag{36}$$

where the first inequality is due to Theorem 2.1.11 in Nesterov (2004) and the second one is simply $(a + b)^2 \le 2a^2 + 2b^2$.

What remains to be done is to bound the two terms with $\boldsymbol{r}^k$ dependence. Firstly, we give a bound on the norm of $\boldsymbol{r}^k$:

$$\| \boldsymbol{r}^k \| = \left\| \left( \sum_{i=1}^{n} \nabla f_i \left( \boldsymbol{x}_{i-1}^{2k-1} \right) - \sum_{i=1}^{n} \nabla f_i \left( \boldsymbol{x}_0^{2k-1} \right) \right) + \left( \sum_{i=1}^{n} \nabla f_{n-i+1} \left( \boldsymbol{x}_{i-1}^{2k} \right) - \sum_{i=1}^{n} \nabla f_{n-i+1} \left( \boldsymbol{x}_0^{2k-1} \right) \right) \right\|$$

$$\le \sum_{i=1}^{n} \left\| \nabla f_i \left( \boldsymbol{x}_{i-1}^{2k-1} \right) - \nabla f_i \left( \boldsymbol{x}_0^{2k-1} \right) \right\| + \sum_{i=1}^{n} \left\| \nabla f_{n-i+1} \left( \boldsymbol{x}_{i-1}^{2k} \right) - \nabla f_{n-i+1} \left( \boldsymbol{x}_0^{2k-1} \right) \right\|.$$

Next, we will use the smoothness assumption and bounded gradients property (Lemma 1).

$$\| \boldsymbol{r}^k \| \le L \sum_{i=1}^{n} \left\| \boldsymbol{x}_{i-1}^{2k-1} - \boldsymbol{x}_0^{2k-1} \right\| + L \sum_{i=1}^{n} \left\| \boldsymbol{x}_{i-1}^{2k} - \boldsymbol{x}_0^{2k-1} \right\|$$

$$\le LG\alpha \sum_{i=1}^{n} i + LG\alpha \sum_{i=1}^{n} (n + i)$$

$$= n(2n - 1)\alpha GL.$$

Hence,

$$\left\| \boldsymbol{r}^k \right\|^2 \le 4n^4 \alpha^2 G^2 L^2. \tag{37}$$

For the $\boldsymbol{r}^k$ term, we need a more careful bound. Since the Hessian is constant for quadratic functions, we use $\boldsymbol{H}_i$ to denote the Hessian matrix of function $f_i(\cdot)$. We start off by using the definition of $\boldsymbol{r}^k$:

$$\boldsymbol{r}^k = \left( \sum_{i=1}^{n} \nabla f_i \left( \boldsymbol{x}_{i-1}^{2k-1} \right) - \sum_{i=1}^{n} \nabla f_i \left( \boldsymbol{x}_0^{2k-1} \right) \right) + \left( \sum_{i=1}^{n} \nabla f_{n-i+1} \left( \boldsymbol{x}_{i-1}^{2k} \right) - \sum_{i=1}^{n} \nabla f_{n-i+1} \left( \boldsymbol{x}_0^{2k-1} \right) \right)$$

$$= \sum_{i=1}^{n} \left( \nabla f_i \left( \boldsymbol{x}_{i-1}^{2k-1} \right) - \nabla f_i \left( \boldsymbol{x}_0^{2k-1} \right) \right) + \sum_{i=1}^{n} \left( \nabla f_{n-i+1} \left( \boldsymbol{x}_{i-1}^{2k} \right) - \nabla f_{n-i+1} \left( \boldsymbol{x}_0^{2k-1} \right) \right)$$

$$= \sum_{i=1}^{n} \boldsymbol{H}_i \left( \boldsymbol{x}_{i-1}^{2k-1} - \boldsymbol{x}_0^{2k-1} \right) + \sum_{i=1}^{n} \boldsymbol{H}_{n-i+1} \left( \boldsymbol{x}_{i-1}^{2k} - \boldsymbol{x}_0^{2k-1} \right),$$

where we used the fact that for a quadratic function $f$ with Hessian $\boldsymbol{H}$, we have that $\nabla f(x) - \nabla f(y) = \boldsymbol{H}(x - y)$. After that, we express $\boldsymbol{x}_{i-1}^{2k-1} - \boldsymbol{x}_0^{2k-1}$ and $\boldsymbol{x}_{i-1}^{2k} - \boldsymbol{x}_0^{2k-1}$ as sum of gradient descent

steps:

$$
\boldsymbol{r}^k = \sum_{i=1}^{n} \boldsymbol{H}_i \left( \sum_{j=1}^{i-1} -\alpha \nabla f_j(\boldsymbol{x}_{j-1}^{2k-1}) \right) + \sum_{i=1}^{n} \boldsymbol{H}_{n-i+1} \left( \sum_{j=1}^{n} -\alpha \nabla f_j(\boldsymbol{x}_{j-1}^{2k-1}) + \sum_{j=1}^{i-1} -\alpha \nabla f_{n-j+1}(\boldsymbol{x}_{j-1}^{2k}) \right)
$$

$$
= -\alpha \sum_{i=1}^{n} \boldsymbol{H}_i \left( \sum_{j=1}^{i-1} \nabla f_j(\boldsymbol{x}_{j-1}^{2k-1}) \right) - \alpha \sum_{i=1}^{n} \boldsymbol{H}_{n-i+1} \left( \sum_{j=1}^{i-1} \nabla f_{n-j+1}(\boldsymbol{x}_{j-1}^{2k}) \right)
$$

$$
- \alpha \sum_{i=1}^{n} \boldsymbol{H}_{n-i+1} \left( \sum_{j=1}^{n} \nabla f_j(\boldsymbol{x}_{j-1}^{2k-1}) \right)
$$

$$
= -\alpha \sum_{i=1}^{n} \boldsymbol{H}_i \left( \sum_{j=1}^{i-1} \nabla f_j(\boldsymbol{x}_0^{2k-1}) \right) - \alpha \sum_{i=1}^{n} \boldsymbol{H}_{n-i+1} \left( \sum_{j=1}^{i-1} \nabla f_{n-j+1}(\boldsymbol{x}_0^{2k}) \right)
$$

$$
- \alpha \sum_{i=1}^{n} \boldsymbol{H}_{n-i+1} \left( \sum_{j=1}^{n} \nabla f_j(\boldsymbol{x}_0^{2k-1}) \right)
$$

$$
- \alpha \sum_{i=1}^{n} \boldsymbol{H}_i \left( \sum_{j=1}^{i-1} \nabla f_j(\boldsymbol{x}_{j-1}^{2k-1}) - f_j(\boldsymbol{x}_0^{2k-1}) \right)
$$

$$
- \alpha \sum_{i=1}^{n} \boldsymbol{H}_{n-i+1} \left( \sum_{j=1}^{i-1} \nabla f_{n-j+1}(\boldsymbol{x}_{j-1}^{2k}) - \nabla f_{n-j+1}(\boldsymbol{x}_0^{2k}) \right)
$$

$$
- \alpha \sum_{i=1}^{n} \boldsymbol{H}_{n-i+1} \left( \sum_{j=1}^{n} \nabla f_j(\boldsymbol{x}_{j-1}^{2k-1}) - \nabla f_j(\boldsymbol{x}_0^{2k-1}) \right)
$$

$$
= -2\alpha \sum_{i=1}^{n} \boldsymbol{H}_i \left( \sum_{j=1}^{n} \nabla f_j(\boldsymbol{x}_0^{2k-1}) \right) + \alpha \sum_{i=1}^{n} \boldsymbol{H}_i \nabla f_i(\boldsymbol{x}_0^{2k-1})
$$

$$
- \alpha \sum_{i=1}^{n} \boldsymbol{H}_i \left( \sum_{j=1}^{i-1} \nabla f_j(\boldsymbol{x}_{j-1}^{2k-1}) - f_j(\boldsymbol{x}_0^{2k-1}) \right)
$$

$$
- \alpha \sum_{i=1}^{n} \boldsymbol{H}_{n-i+1} \left( \sum_{j=1}^{i-1} \nabla f_{n-j+1}(\boldsymbol{x}_{j-1}^{2k}) - \nabla f_{n-j+1}(\boldsymbol{x}_0^{2k}) \right)
$$

$$
- \alpha \sum_{i=1}^{n} \boldsymbol{H}_i \left( \sum_{j=1}^{n} \nabla f_j(\boldsymbol{x}_{j-1}^{2k-1}) - \nabla f_j(\boldsymbol{x}_0^{2k-1}) \right)
$$

Next, we use the fact that $\sum_{j=1}^{n} \nabla f_j(\boldsymbol{x}) = n\nabla F(x)$. We will also again use the fact that for a quadratic function $f$ with Hessian $\boldsymbol{H}$, we have that $\nabla f(x) - \nabla f(y) = \boldsymbol{H}(x - y)$:

$$
\boldsymbol{r}^k = -2\alpha \sum_{i=1}^{n} \boldsymbol{H}_i(n\nabla F(\boldsymbol{x}_0^{2k-1})) + \alpha \sum_{i=1}^{n} \boldsymbol{H}_i(\nabla f_i(\boldsymbol{x}_0^{2k-1}) - \nabla f_i(\boldsymbol{x}^*)) + \alpha \sum_{i=1}^{n} \boldsymbol{H}_i \nabla f_i(\boldsymbol{x}^*)
$$

$$
- \alpha \sum_{i=1}^{n} \boldsymbol{H}_i \left( \sum_{j=1}^{i-1} \nabla f_j(\boldsymbol{x}_{j-1}^{2k-1}) - f_j(\boldsymbol{x}_0^{2k-1}) \right)
$$

$$
- \alpha \sum_{i=1}^{n} \boldsymbol{H}_{n-i+1} \left( \sum_{j=1}^{i-1} \nabla f_{n-j+1}(\boldsymbol{x}_{j-1}^{2k}) - \nabla f_{n-j+1}(\boldsymbol{x}_0^{2k}) \right)
$$

$$- \alpha \sum_{i=1}^{n} \boldsymbol{H}_i \left( \sum_{j=1}^{n} \nabla f_j(\boldsymbol{x}_{j-1}^{2k-1}) - \nabla f_j(\boldsymbol{x}_0^{2k-1}) \right)$$

$$= -2\alpha \left( \sum_{i=1}^{n} \boldsymbol{H}_i \right)^2 (\boldsymbol{x}_0^{2k-1} - \boldsymbol{x}^*) + \alpha \sum_{i=1}^{n} \boldsymbol{H}_i^2 (\boldsymbol{x}_0^{2k-1} - \boldsymbol{x}^*) + \alpha \sum_{i=1}^{n} \boldsymbol{H}_i \nabla f_i(\boldsymbol{x}^*)$$

$$- \alpha \sum_{i=1}^{n} \boldsymbol{H}_i \left( \sum_{j=1}^{i-1} \nabla f_j(\boldsymbol{x}_{j-1}^{2k-1}) - f_j(\boldsymbol{x}_0^{2k-1}) \right)$$

$$- \alpha \sum_{i=1}^{n} \boldsymbol{H}_{n-i+1} \left( \sum_{j=1}^{i-1} \nabla f_{n-j+1}(\boldsymbol{x}_{j-1}^{2k}) - \nabla f_{n-j+1}(\boldsymbol{x}_0^{2k}) \right)$$

$$- \alpha \sum_{i=1}^{n} \boldsymbol{H}_i \left( \sum_{j=1}^{n} \nabla f_j(\boldsymbol{x}_{j-1}^{2k-1}) - \nabla f_j(\boldsymbol{x}_0^{2k-1}) \right)$$

$$= \boldsymbol{a}^k + \boldsymbol{b}^k,$$

where the random variables $\boldsymbol{a}^k, \boldsymbol{b}^k$ as

$$\boldsymbol{a}^k := -2\alpha \left( \sum_{i=1}^{n} \boldsymbol{H}_i \right)^2 (\boldsymbol{x}_0^{2k-1} - \boldsymbol{x}^*) + \alpha \sum_{i=1}^{n} \boldsymbol{H}_i^2 (\boldsymbol{x}_0^{2k-1} - \boldsymbol{x}^*) + \alpha \sum_{i=1}^{n} \boldsymbol{H}_i \nabla f_i(\boldsymbol{x}^*), \text{ and}$$

$$\boldsymbol{b}^k := -\alpha \sum_{i=1}^{n} \boldsymbol{H}_i \left( \sum_{j=1}^{i-1} \nabla f_j(\boldsymbol{x}_{j-1}^{2k-1}) - f_j(\boldsymbol{x}_0^{2k-1}) \right)$$

$$- \alpha \sum_{i=1}^{n} \boldsymbol{H}_{n-i+1} \left( \sum_{j=1}^{i-1} \nabla f_{n-j+1}(\boldsymbol{x}_{j-1}^{2k}) - \nabla f_{n-j+1}(\boldsymbol{x}_0^{2k}) \right)$$

$$- \alpha \sum_{i=1}^{n} \boldsymbol{H}_i \left( \sum_{j=1}^{n} \nabla f_j(\boldsymbol{x}_{j-1}^{2k-1}) - \nabla f_j(\boldsymbol{x}_0^{2k-1}) \right).$$

Again, using smoothness assumption and bounded gradients property (Lemma 1) we get,

$$\|\boldsymbol{b}^k\| \le 3\alpha^2 L^2 G n^3. \tag{38}$$

Next, we decompose the inner product of $\boldsymbol{x}_0^{2k-1} - \boldsymbol{x}^*$ and $\mathbb{E}\left[\boldsymbol{r}^k\right]$ in Eq. (36):

$$-2\alpha \left\langle \boldsymbol{x}_0^{2k-1} - \boldsymbol{x}^*, \boldsymbol{r}^k \right\rangle = -2\alpha \left\langle \boldsymbol{x}_0^{2k-1} - \boldsymbol{x}^*, \boldsymbol{a}^k + \boldsymbol{b}^k \right\rangle$$

$$= -2\alpha \left\langle \boldsymbol{x}_0^{2k-1} - \boldsymbol{x}^*, \boldsymbol{a}^k \right\rangle - 2\alpha \left\langle \boldsymbol{x}_0^{2k-1} - \boldsymbol{x}^*, \boldsymbol{b}^k \right\rangle \tag{39}$$

For the first term in (39),

$$-2\alpha \left\langle \boldsymbol{x}_0^{2k-1} - \boldsymbol{x}^*, \boldsymbol{a}^k \right\rangle = 4\alpha^2 \left\langle \boldsymbol{x}_0^{2k-1} - \boldsymbol{x}^*, \left( \sum_{i=1}^{n} \boldsymbol{H}_i \right)^2 (\boldsymbol{x}_0^{2k-1} - \boldsymbol{x}^*) \right\rangle$$

$$- 2\alpha^2 \left\langle \boldsymbol{x}_0^{2k-1} - \boldsymbol{x}^*, \sum_{i=1}^{n} \boldsymbol{H}_i^2 (\boldsymbol{x}_0^{2k-1} - \boldsymbol{x}^*) \right\rangle$$

$$- 2\alpha^2 \left\langle \boldsymbol{x}_0^{2k-1} - \boldsymbol{x}^*, \sum_{i=1}^{n} \boldsymbol{H}_i \nabla f_i(\boldsymbol{x}^*) \right\rangle$$

$$\le 4\alpha^2 \left\langle \boldsymbol{x}_0^{2k-1} - \boldsymbol{x}^*, \left( \sum_{i=1}^{n} \boldsymbol{H}_i \right)^2 (\boldsymbol{x}_0^{2k-1} - \boldsymbol{x}^*) \right\rangle$$

$$- 2\alpha^2 \left\langle \boldsymbol{x}_0^{2k-1} - \boldsymbol{x}^*, \sum_{i=1}^n \boldsymbol{H}_i \nabla f_i(\boldsymbol{x}^*) \right\rangle$$

$$= 4\alpha^2 n^2 \|\nabla F(\boldsymbol{x}_0^{2k-1})\|^2 - 2\alpha^2 \left\langle \boldsymbol{x}_0^{2k-1} - \boldsymbol{x}^*, \sum_{i=1}^n \boldsymbol{H}_i \nabla f_i(\boldsymbol{x}^*) \right\rangle$$

$$\leq 4\alpha^2 n^2 \|\nabla F(\boldsymbol{x}_0^{2k-1})\|^2 + 2\alpha^2 n L G \|\boldsymbol{x}_0^{2k-1} - \boldsymbol{x}^*\|. \tag{40}$$

For the second term in (39), we use Cauchy-Schwarz and Ineq. (38)

$$-2\alpha \left\langle \boldsymbol{x}_0^{2k-1} - \boldsymbol{x}^*, \boldsymbol{b}^k \right\rangle \leq 6\alpha^3 L^2 G n^3 \|\boldsymbol{x}_0^{2k-1} - \boldsymbol{x}^*\|. \tag{41}$$

Substituting (40) and (41) back to (39), we get

$$-2\alpha \left\langle \boldsymbol{x}_0^{2k-1} - \boldsymbol{x}^*, \boldsymbol{r}^k \right\rangle \leq 4\alpha^2 n^2 \|\nabla F(\boldsymbol{x}_0^{2k-1})\|^2 + 2\alpha^2 n L G \|\boldsymbol{x}_0^{2k-1} - \boldsymbol{x}^*\|$$
$$+ 6\alpha^3 L^2 G n^3 \|\boldsymbol{x}_0^{2k-1} - \boldsymbol{x}^*\|. \tag{42}$$

Substituting (37) (42) back to (36), we finally get a recursion bound for one epoch:

$$\|\boldsymbol{x}_n^{2k} - \boldsymbol{x}^*\|^2 \leq \left(1 - 4n\alpha \frac{L\mu}{L+\mu}\right) \|\boldsymbol{x}_0^{2k-1} - \boldsymbol{x}^*\|^2 - \left(4n\alpha \frac{1}{L+\mu} - 8\alpha^2 n^2\right) \left\|\nabla F(\boldsymbol{x}_0^{2k-1})\right\|^2$$

$$- 2\alpha \left\langle \boldsymbol{x}_0^{2k-1} - \boldsymbol{x}^*, \boldsymbol{r}^k \right\rangle + 2\alpha^2 \left\|\boldsymbol{r}^k\right\|^2$$

$$\leq \left(1 - 4n\alpha \frac{L\mu}{L+\mu}\right) \|\boldsymbol{x}_0^{2k-1} - \boldsymbol{x}^*\|^2 - \left(4n\alpha \frac{1}{L+\mu} - 8\alpha^2 n^2\right) \left\|\nabla F(\boldsymbol{x}_0^{2k-1})\right\|^2$$

$$4\alpha^2 n^2 \|\nabla F(\boldsymbol{x}_0^{2k-1})\|^2 + 2\alpha^2 n L G \|\boldsymbol{x}_0^{2k-1} - \boldsymbol{x}^*\| + 6\alpha^3 L^2 G n^3 \|\boldsymbol{x}_0^{2k-1} - \boldsymbol{x}^*\|$$

$$+ 8\alpha^4 n^4 G^2 L^2$$

$$= \left(1 - 4n\alpha \frac{L\mu}{L+\mu}\right) \|\boldsymbol{x}_0^{2k-1} - \boldsymbol{x}^*\|^2 - \left(4n\alpha \frac{1}{L+\mu} - 12\alpha^2 n^2\right) \left\|\nabla F(\boldsymbol{x}_0^{2k-1})\right\|^2$$

$$+ 2\alpha^2 n L G (1 + 3\alpha L n^2) \|\boldsymbol{x}_0^{2k-1} - \boldsymbol{x}^*\| + 8\alpha^4 n^4 G^2 L^2.$$

Next, we use the fact that $2ab \leq \lambda a^2 + b^2/\lambda$ (for any $\lambda > 0$) on the term $2\alpha^2 n L G (1 + 3\alpha L n^2) \|\boldsymbol{x}_0^{2k-1} - \boldsymbol{x}^*\|$ to get that

$$2\alpha^2 n L G (1 + 3\alpha L n^2) \|\boldsymbol{x}_0^{2k-1} - \boldsymbol{x}^*\| \leq \left(\alpha^2 n L G (1 + 3\alpha L n^2)\right)^2 / (n\alpha\mu) + n\alpha\mu \|\boldsymbol{x}_0^{2k-1} - \boldsymbol{x}^*\|^2$$
$$= \mu^{-1} \alpha^3 n L^2 G^2 (1 + 3\alpha L n^2)^2 + n\alpha \|\boldsymbol{x}_0^{2k-1} - \boldsymbol{x}^*\|^2.$$

Substituting this back we get,

$$\|\boldsymbol{x}_n^{2k} - \boldsymbol{x}^*\|^2 \leq \left(1 - 4n\alpha \frac{L\mu}{L+\mu}\right) \|\boldsymbol{x}_0^{2k-1} - \boldsymbol{x}^*\|^2 - \left(4n\alpha \frac{1}{L+\mu} - 12\alpha^2 n^2\right) \left\|\nabla F(\boldsymbol{x}_0^{2k-1})\right\|^2$$

$$+ \mu^{-1} \alpha^3 n L^2 G^2 (1 + 3\alpha L n^2)^2 + n\alpha\mu \|\boldsymbol{x}_0^{2k-1} - \boldsymbol{x}^*\|^2 + 8\alpha^4 n^4 G^2 L^2$$

$$\leq \left(1 - 2n\alpha\mu + n\alpha\mu\right) \|\boldsymbol{x}_0^{2k-1} - \boldsymbol{x}^*\|^2 - \left(4n\alpha \frac{1}{L+\mu} - 12\alpha^2 n^2\right) \left\|\nabla F(\boldsymbol{x}_0^{2k-1})\right\|^2$$

$$+ \mu^{-1} \alpha^3 n L^2 G^2 (1 + 3\alpha L n^2)^2 + 8\alpha^4 n^4 G^2 L^2 \qquad \text{(Since } \mu \leq L\text{)}$$

$$= \left(1 - n\alpha\mu\right) \|\boldsymbol{x}_0^{2k-1} - \boldsymbol{x}^*\|^2 - \left(4n\alpha \frac{1}{L+\mu} - 12\alpha^2 n^2\right) \left\|\nabla F(\boldsymbol{x}_0^{2k-1})\right\|^2$$

$$+ \mu^{-1} \alpha^3 n L^2 G^2 (1 + 3\alpha L n^2)^2 + 8\alpha^4 n^4 G^2 L^2$$

Now, substituting the values of $\alpha$ and the bound on $K$, we get that $4n\alpha \frac{1}{L+\mu} - 12\alpha^2 n^2 \geq 0$ and hence,

$$\|\boldsymbol{x}_n^{2k} - \boldsymbol{x}^*\|^2 \leq (1 - n\alpha\mu) \|\boldsymbol{x}_0^{2k-1} - \boldsymbol{x}^*\|^2 + \mu^{-1} \alpha^3 n L^2 G^2 (1 + 3\alpha L n^2)^2 + 8\alpha^4 n^4 G^2 L^2$$

Now, iterating this for $K/2$ epoch pairs, we get

$$\|\boldsymbol{x}_n^K - \boldsymbol{x}^*\|^2 \leq (1 - n\alpha\mu)^{K/2} \|\boldsymbol{x}_0^1 - \boldsymbol{x}^*\|^2 + \frac{K}{2} \mu^{-1} \alpha^3 n L^2 G^2 (1 + 3\alpha L n^2)^2 + 4K\alpha^4 n^4 G^2 L^2$$

$$\leq e^{-n\alpha\mu K/2}\|\boldsymbol{x}_0^1 - \boldsymbol{x}^*\|^2 + \frac{K}{2}\mu^{-1}\alpha^3 nL^2G^2(1+3\alpha Ln^2)^2 + 4K\alpha^4 n^4 G^2 L^2$$

$$\leq e^{-n\alpha\mu K/2}\|\boldsymbol{x}_0^1 - \boldsymbol{x}^*\|^2 + K\mu^{-1}\alpha^3 nL^2G^2 + 9K\mu^{-1}\alpha^5 n^5 L^4 G^2 + 4K\alpha^4 n^4 G^2 L^2$$
$$\text{(Since } (1+a)^2 \leq 2+2a^2\text{)}$$

Substituting $\alpha = \frac{6\log nK}{\mu nK}$ gives us the desired result.

$\square$

## H  ADDITIONAL EXPERIMENTS

Although our theoretical guarantees for FLIPFLOP only hold for quadratic objectives, we conjecture that FLIPFLOP might be able to improve the convergence performance for other classes of functions, whose Hessians are smooth near their minimizers. To see this, we also ran some experiments on 1-dimensional logistic regression. As we can see in Figure 3, the convergence rates are very similar to those on quadratic functions. The data was synthetically generated such that the objective function becomes strongly convex and well conditioned near the minimizer. Note that logistic loss is not strongly convex on linearly separable data. Therefore, to make the loss strongly convex, we ensured that the data was not linearly separable. Essentially, the dataset was the following: all the datapoints were $z = \pm 1$, and their labels were $y = \mathbb{1}_{z>0}$ with probability $3/4$ and $y = \mathbb{1}_{z<0}$ with probability $1/4$. Framing this as an optimization problem, we have

$$\min_x F(x) := \mathbb{E}\left[-y\log(h(xz)) - (1-y)\log(1-h(xz))\right],$$

where $h(xz) = 1/(1+e^{-xz})$. Note that $x = -\log 3$ is the minimizer of this function, which is helpful because we can use it to compute the exact error. Similar to the experiment on quadratic functions, $n$ was set to $800$ and step size was set in the same regime as in Theorems 4, 5, and 6.

