# OpenReview forum: "Permutation-Based SGD: Is Random Optimal?"
_ICLR.cc/2022/Conference — ICLR 2022 Poster_

### Official Review · Reviewer_53pE · 2021-11-01

**Correctness:** 3
**Technical Novelty And Significance:** 3
**Empirical Novelty And Significance:** 1
**Recommendation:** 6
**Confidence:** 3

**Main Review:**

### Strengths
1. This paper provides some insights into permutation-based SGD for finite sum optimization. Theoretical contributions include exponential convergence rate for 1-d smooth functions, lower bound for general strongly-convex functions, and lower bound 1-d strongly-convex function with nonconvex component.
2. The paper proposes an algorithm FlipFlop, which is easy to implement and proved to have better performance than random permutations for quadratic functions. Empirical results also show FlipFlop performs better for logistic-like functions.
3. The paper is well organized and clearly organized.

### Weakness
1. Convergence rates of FlipFlop are only proved for quadratic functions. This is too restricted. I expect to see some discussions on the difficulty of generalizing the proof to more general functions.
2. Other restrictions include: fixed step size; dependence only in the number of iterations; numerical verification is only for a toy example

### Minor comments:
1. A different but related problem is the permutation in coordinate-wise SGD. A discussion on whether the proposed algorithm FlipFlop can be applied to coordinate-descent would be interesting.


**Summary Of The Paper:**

This paper studies theoretical properties of permutation-based fixed-step size SGD for finite sum optimization. The main theorems state that
* For 1-d Hessian-smooth functions, there exists permutations such that the convergence rate is exponential in the number of iterations $K$
* For higher dimension, there exists strongly-convex function such that the convergence rate is at best $O(1/K^3)$
* If some component in the strongly-convex objective function is nonconvex, the convergence rate has a lower bound $O(1/K^2)$

The paper then proposes the algorithm FlipFlop, where adjacent epochs use permutations of reverse order. The algorithm is proved to improve upon random permutation when all the component functions are quadratic. The results are corroborated with simulations.

**Summary Of The Review:**

In general, I recommend accepting the paper. Although the settings studied in the paper is limited, it provides a better understanding of permutation-based SGD and can motivate further studies.

---

> ### Author Response · Authors · 2021-11-21
> **Author Response to Reviewer 53pE**
>
> We thank you for your review, and providing constructive feedback. We have addressed your concerns below.
>
> ***
>
> **Convergence rates of FlipFlop are only proved for quadratic functions. This is too restricted. I expect to see some discussions on the difficulty of generalizing the proof to more general functions.**
>
>  **Response:** The analysis of FlipFlop leverages the fact that the Hessians of quadratic functions are constant. We think that the analysis of flipflop might be extended to strongly convex functions or even PŁ functions (which are non-convex in general), under some assumptions on the Lipschitz continuity of the Hessians, similar to how HaoChen\&Sra(2019) extended their analysis of quadratic functions to more general classes. A key take-away from FlipFlop is that we had to understand how permutation based SGD works specifically for quadratic functions, that is, we did a white-box analysis. In general, we feel that depending on the specific class of non-convex function (say deep neural networks), practitioners would have to think about permutation based SGD in a white-box fashion, to come up with better heuristics for shuffling.
>
> Due to the 9 page limit, we have currently added these to the appendix (Appendix A) of the paper, but will try to move it to the main paper in the next version, after restructuring the paper.
>
> ***
>
> **Other restrictions include: fixed step size; dependence only in the number of iterations**
>
> **Response:** Being the first paper (to the best of our knowledge) to theoretically analyze optimality of random permutations, we limited our scope to a specific, but common setting - strong convexity, constant step sizes, and only analyzed the dependence on the number of iterations. A more generic analysis that considers variable step sizes, and the dependence on factors like the condition number would be a very important direction of future work.
>
> ***
>
> **A different but related problem is the permutation in coordinate-wise SGD. A discussion on whether the proposed algorithm FlipFlop can be applied to coordinate-descent would be interesting.**
>
> **Response:** This is a very good point, and indeed we were recently informed about how a shuffling scheme similar to FlipFlop was being used in random coordinate descent for faster practical convergence (see page 231 in Nocedal\&Wright(2006)). For quadratic functions, even though the current analysis of FlipFlop would not go through for coordinate descent, we think that since the Hessian is constant, one can prove similar convergence rates, with a modified analysis.
>
> We have added this reference to the paper, along with the discussion on generalizing FlipFlop to non-quadratic functions.
>
> **References**
>
> Nocedal, Jorge, and Stephen Wright. Numerical optimization. Springer Science & Business Media, 2006.

---

### Official Review · Reviewer_jCFt · 2021-11-02

**Correctness:** 4
**Technical Novelty And Significance:** 4
**Empirical Novelty And Significance:** 3
**Recommendation:** 6
**Confidence:** 4

**Main Review:**

**Strengths**

This paper gives interesting theoretical results detailing precisely when random permutations are optimal for SGD under different settings for strongly-convex functions. The proof techniques are pretty clever. I also agree with the authors that Lemma 1 is indeed of independent interest and could facilitate future analysis on permutation-based algorithms in the strongly-convex setting. The FlipFlop technique they introduced is also very easy to implement with basically zero overhead, yet is able to speed up convergence both provably and in the synthetic experiments considered. The accelerated rate obtained are also quite impressive considering how simple the method is.

**Weaknesses**

Below are some nice-to-have's that are beyond the scope of this paper, but would significantly strengthen the paper if included:
- All analyses in this work are for either quadratic functions or strongly convex functions. It'd be nice if the authors have further insights for non-convex functions as well.
- Since FlipFlop is such a low-cost, easy to implement way to boost the performance in the strongly-convex case, trying them out on real-data benchmarks for (strongly-)convex functions or even some deep learning benchmarks would be very interesting to see. I am especially curious whether FlipFlop can bring "free speedup" in practice when the objective is non-convex.


**Typos and minor issues**
- Theorem 2: the initialization requirement in the proof should be stated more clearly in the theorem statement. The toy example given in the proof sketch does not work when initialized arbitrarily with arbitrary step size.
- page 13: "Similarly, we can find show that" -> "Similarly, we can show that"


**Summary Of The Paper:**

This work investigates the optimality of random permutation as a scan ordering for SGD. They found that for general strongly convex functions with Lipschitz Hessian, random permutations are optimal in high dimension but not optimal in 1-dimension. For general convex quadratics, random permutations are also not optimal. Finally, the authors introduced a new technique termed FlipFlop that works by reversing the permutation of the previous epoch at every even epoch, and at the odd epochs the algorithm just follows its original permutation, whether it be random or cyclic. FlipFlop has been proven to improve the convergence of random reshuffling, single shuffle, and incremental gradient descent on quadratic functions. Experiments on a 100-dimensional quadratic objective and 1-dimensional logistic regression demonstrate that FlipFlop indeed converges faster than random reshuffling.

**Summary Of The Review:**

Although the paper considers the optimality of permutation-based SGD only for strongly-convex functions, the theoretical results are interesting enough and the proposed trick FlipFlop seems to be a very promising approach to further accelerate random reshuffling. I am leaning towards accepting the paper, although additional experiments could further strengthen the paper.

---

> ### Author Response · Authors · 2021-11-21
> **Author Response to Reviewer jCFt**
>
> We thank you for your review, and providing constructive feedback. We have addressed your concerns below.
>
> ***
>
> **All analyses in this work are for either quadratic functions or strongly convex functions. It'd be nice if the authors have further insights for non-convex functions as well.**
>
>  **Response:** This is a good point. The analysis of FlipFlop leverages the fact that the Hessians of quadratic functions are constant. We think that the analysis of flipflop might be extended to PŁ functions (which are non-convex in general), under some assumptions on the Lipschitz continuity of the Hessians, similar to how HaoChen\&Sra(2019) extended their analysis of quadratic functions to more general classes. A key take-away from FlipFlop is that we had to understand how permutation based SGD works specifically for quadratic functions, that is, we did a white-box analysis. In general, we feel that depending on the specific class of non-convex function (say deep neural networks), practitioners would have to think about permutation based SGD in a white-box fashion, to come up with better heuristics for shuffling.
>
> In a concurrent work submitted to the same conference (https://openreview.net/pdf?id=7gWSJrP3opB), it is shown that by greedily sorting stale gradients, a permutation order can be found which converges faster for some deep learning tasks. Hence, there do exist better permutations than random, even for deep learning tasks. We could not find this paper on arXiv, so we have added a citation to the anonymous openreview version of this paper.
>
> Due to the 9 page limit, we have currently added this discussion to the appendix (Appendix A) of the paper, but will try to move it to the main paper in the next version, after restructuring the paper.
>
> ***
>
> **Since FlipFlop is such a low-cost, easy to implement way to boost the performance in the strongly-convex case, trying them out on real-data benchmarks....**
>
> **Response:** We had tried FlipFlop on CIFAR-10 (mentioned in Section 6.2 of our paper), and on your suggestion, we also ran FlipFlop on some strongly convex real-data benchmarks, but unfortunately we did not see a significant speedup. We will continue testing and fine-tuning on more real datasets, and will add the additional benchmark experiments to the final version of the paper.
>
> ***
>
> **Theorem 2: the initialization requirement in the proof should be stated more clearly in the theorem statement...**
>
> **Response:** Thank you for the suggestion, but the lower bound is actually invariant to initialization, as discussed below. We have also added the following discussion in the paper, please see Appendix B.4.
>
>
> The lower bound partitions the step size in three ranges -
>
> * In the step size ranges $\alpha \in [\frac{1}{2(n-1)KL},\frac{3}{nL}]$ and $\alpha \in [\frac{3}{nL}, \frac{1}{L}]$, the initializations are done at the minimizer and it is shown that any permutation based algorithm will still move away from the minimizer. The choice of initializing at the minimizer was solely for the convenience of analysis and calculations, and the proof should work for any other initialization as well.
>
> 	Furthermore, the effect of initializing at any random point will decay exponentially fast with epochs anyway. To see how, note that every epoch can be treated as $n$ steps of full gradient descent and some noise, and hence the full gradient descent part will essentially keep decreasing the effect of initialization exponentially, and what we would be left with is the noise in each epoch. Thus, it was more convenient for us to just assume initialization at the minimizer and only focus on the noise in each epoch.
> * The step size range $\alpha \notin [\frac{1}{2(n-1)KL}, \frac{1}{L}]$ can be divided into two parts, $\alpha \in [0,\frac{1}{2(n-1)KL}]$ and $\alpha \in [\frac{1}{L},\infty)$.
>
> 	For the range $\alpha \in [0,\frac{1}{2(n-1)KL}]$, we essentially show that the step size is too small to make any meaningful progress towards the minimizer. Hence, instead of initializing at $1$, initializing at any other arbitrary or random (non-zero) point will also give the same slow convergence rate.
>
> 	For the range $\alpha \in [\frac{1}{L},\infty)$, we show that the optimization algorithm will in fact diverge since the step size is too large. Hence, even here, any other arbitrary (non-zero) point of initialize will also give divergence.
>     \end{itemize}
>
> Yes, we agree that the toy example in the proof sketch does not work for arbitrarily small step size, but our motivation was to demonstrate the lower bound construction for the step size range of interest, which is neither "too small", nor "too large". We will clarify this in the next version of the paper, along with some other changes in the proof sketches, as suggested by Reviewer D2Ry.
>
> ***
> **Typo on Page 13**
>
> **Response:** Thank you for pointing out the typo. We have fixed it.

---

### Official Review · Reviewer_JbYX · 2021-11-03

**Correctness:** 4
**Technical Novelty And Significance:** 3
**Empirical Novelty And Significance:** 2
**Recommendation:** 6
**Confidence:** 2

**Details Of Ethics Concerns:**

The paper studies a theory problem (how permutation-order affects convergence in SGD), and does not pose obvious ethics concerns (as stated in the paper).

**Main Review:**

Strengths:
1. The paper studies SGD, the workhorse of optimization in machine learning.
2. The algorithms are simple and natural.
3. A natural twist (flipflopping, that is, reversing the permutation every other epoch) improves convergence on some subfamilies of strongly convex functions, beating known lower bounds without such twist.
4. The analysis does not appear to be complicated.
5. The paper is well written and easy to read and understand.

Regarding weaknesses, as admitted in the paper, the results are:
1. Not applicable to all convex function (for flipflopping), or non-convex functions.
2. Not applicable to other techniques such as variance reduction and momentum.
3. Not applicable when step size is not constant.
Those extensions will be left as future work, if possible.

**Summary Of The Paper:**

This paper is motivated by the observed phenomenon that, in stochastic gradient descent (SGD), without-replacement sampling (random permutation) gives faster convergence than with-replacement sampling.
The paper studies whether random permutations are optimal among permutation-based SGD, by considering different deterministic, random, or hybrid ways of generating permutations of input points.

Focusing on optimizing convex functions at a constant step size, the paper shows that:
1. there exist optimal permutations which converges exponentially faster than random permutations for 1-dimensional functions.
2. such improvement is not possible in higher dimensions or for strongly convex objectives, where random permutations are optimal.
3. by reversing the permutation every other epoch (flipflopping), convergence on quadratic functions improves for three permutation-based methods: Incremental Gradient Descent (deterministic), Random Reshuffle (random), and Single Shuffle (hybrid).

**Summary Of The Review:**

The research is well motivated, and a simple intuitive idea (flipflopping) improves convergence for certain (admittedly limited) settings.
This reviewer thinks the paper could be a contribution to this conference.

---

> ### Author Response · Authors · 2021-11-21
> **Author Response to Reviewer JbYX**
>
> Thank you for your review, and noting the paper's weaknesses. While the weaknesses are valid, we would like to say that since this is the first paper (to the best of our knowledge) to theoretically analyze the optimality of random permutations, we limited our scope to a specific, but common theoretical setting - strongly convex functions with constant step size.
>
> We have discussed your concerns below.  Due to the 9 page limit, we have currently added these to the appendix (Appendix A) of the paper, but will try to move it to the main paper in the next version, after restructuring the paper.
>
> ***
>
> **Not applicable to all convex function (for flipflopping), or non-convex functions.**
>
> **Response:** The analysis of FlipFlop leverages the fact that the Hessians of quadratic functions are constant. We think that the analysis of flipflop might be extended to general strongly convex functions or even PŁ functions (which are non-convex in general), under some assumptions on the Lipschitz continuity of the Hessians (similar to how HaoChen\&Sra(2019) extended their analysis of quadratic functions to more general classes). A key take-away from FlipFlop is that we had to understand how permutation based SGD works specifically for quadratic functions, that is, we did a white-box analysis. In general, we feel that depending on the specific class of non-convex function (say deep neural networks), practitioners would have to think about permutation based SGD in a white-box fashion, to come up with better heuristics for shuffling.
>
>    ***
>
> **Not applicable to other techniques such as variance reduction and momentum.**
>
> **Response:** One of the main motivations behind FlipFlop was to demonstrate that there can exist permutations better than random, for specific classes of functions. However, the analysis turned out to be quite lengthy, even without considering momentum, variance reduction or variable step sizes. Hence, while it would be very interesting to see how FlipFlop behaves under these modifications, it might be better to analyze other permutation-based techniques, that work on non-convex objectives like deep learning, and how those work with variance reduction, momentum etc.
>
> ***
> **Not applicable when step size is not constant.**
>
> **Response:** This is a very good suggestion, since all the existing lower bounds (to the best of our knowledge) work in the constant step size regime (Safran\&Shamir(2020), Rajput et al. (2020), and Safran\&Shamir(2021)). Thus, this would be a very important direction for future research.
>
> However, the case when step sizes are not constant can be tricky to prove lower bounds, since the step size could potentially depend on the permutation, and the current iterate. A more reasonable setting to prove lower bounds could be the case when the step sizes follow a schedule over epochs, similar to what happens in practice.
>
> ***
>
> We thank the reviewer for noting the potential future directions for research, and we agree that these could lead to very interesting future work.
>
> ***
>
> **References**
>
> Haochen, Jeff, and Suvrit Sra. "Random shuffling beats sgd after finite epochs." In International Conference on Machine Learning, pp. 2624-2633. PMLR, 2019.
>
> Safran, Itay, and Ohad Shamir. "How good is SGD with random shuffling?." In Conference on Learning Theory, pp. 3250-3284. PMLR, 2020.
>
> Rajput, Shashank, Anant Gupta, and Dimitris Papailiopoulos. "Closing the convergence gap of SGD without replacement." In International Conference on Machine Learning, pp. 7964-7973. PMLR, 2020.
>
> Safran, Itay, and Ohad Shamir. "Random Shuffling Beats SGD Only After Many Epochs on Ill-Conditioned Problems." arXiv preprint arXiv:2106.06880 (2021).

---

### Official Review · Reviewer_D2Ry · 2021-11-06

**Correctness:** 4
**Technical Novelty And Significance:** 4
**Empirical Novelty And Significance:** 2
**Recommendation:** 10
**Confidence:** 4

**Main Review:**

## Writing

The submission is well written with a carefully thought-out story. I'd like to congratulate the authors on their highly polished submission.

## Theory

I believe the theoretical claims in the submission are sound. I thoroughly checked the proofs for Theorems 1-3 and found only minor typographic issues.
I checked the remaining theorems (although less thoroughly) and did not discover any issues.
Some questions/comments for the authors are as follow:

**Theorem 1**: How did the authors conclude that permutations used to obtain the exponential rate are optimal?
I would have thought a minimax lower-bound for permutation-based SGD on 1-dimensional, Hessian-smooth functions to conclude (minimax) optimality and the paper does not cite such a result.

**Theorem 2**: the lower bound appears to be invariant to initialization, but only as long as the initialization is deterministic so that it can be observed by a resisting oracle.
This knowledge seems necessary in order to use a standard translation argument. Is this correct?
If so, do the authors think that a randomized bound is attainable?
This would more closely match practice, where $x_0$ is typically chosen at random.

**Section 6.1**: I don't think this proof sketch contributes significantly to the paper as it is written.
The intuition that FlipFlop approximately equalizes the weights applied to the $b_i$ terms in Eq. 7 is helpful and should be retained.
However, the number of approximations --- linearizing twice and then dropping higher order terms in the display at the bottom of Page 8 --- reduces my confidence in the final conclusion.
How do we know that Eq. 10 is still a good representation of the error after two epochs of FlipFlop given the sequence of approximations?
This space might be better used by Figure 3, which could be moved in to the main paper, and a discussion of potential difficulties extending the analysis to general convex functions.

**Lemma 1**: This is a nice result that I agree is of independent interest.
For example, analyses of some stochastic gradient methods (e.g. Adagrad) require that the iterates remain in a compact set.
Typically, constrained optimization with a compact constraint set is assumed;
however, perhaps restricting the analysis to "permutation Adagrad" and using a result analogous to Lemma 1 could do away with this unrealistic assumption and yield more practical theorems.


## Experiments

The experiments are small-scale and intended to confirm the theoretical developments.
As the methods experimented with are stochastic, they could be improved by including distribution information over multiple "restarts" or repeats.
For example, running 10 repeats with different seeds and then plotting the median + inter-quartile range would increase my confidence in the results.
Since this is not an empirical paper, I think the change is desirable rather than essential.

Figure 2:  I suggest including $n = 800 > K $ so that $1/(n^2 * K^2) \approx K^4$ and the high-order (in $K$) terms dominate as part of the figure caption, since this information is provided two pages later in Sec. 6.2.
The font sizes and line-widths should be increased so that the figure is legible when printed.
These same comments apply to Figure 3.

## Minor Comments

Page 2:
- "As we see in the following, the answer the the above is not straightforward, and depends heavily on the function class at hand." --- delete extra "the".

Appendix A.3:
- "In the previous subsection, we have show that..." -> "In the previous subsection, we have \*\*shown\*\* that..."

Appendix A.4:
- The `restatable` environment provided by `thmtools` is useful when proving lemmas which were stated earlier in the text.
- Bottom of page 17: you might want to be a bit careful limits of the last integral here, since there is no guarantee that $x_{i - 1} + (y_{0} - x_0)$.

Appendix B.1:
- For any $i \\in [n/2]$, let $p$ and $q$ be indices such that $\\sigma_p = f_i$ and $\\sigma_p = g_i$ --- I believe the second $\\sigma_p$ should be $\\sigma_q$.

Appendix B.2:
- Def.\ of $F(y)$: I believe the definition is missing a $1/n$ term, i.e. $F(y) = \\frac{1}{n} \\sum_{i=1}^n f_i(y)$.
- It is worth noting in the proof that the bound on $y_{n, j}$ can be applied recursively (over epochs) because it is independent of $\sigma_s$ and thus holds for all $K$.

Appendix C:
- "Similarly, for the other possible permutation..." --- I think you're missing $x_{0, k+1} = \\ldots$ in the equation at the end of this line.

**Summary Of The Paper:**

# === Update ===

I have decided to maintain my score. This is very strong submission in my opinion. A brief overview of my thoughts is the following:

**Topic**: This paper comes in the middle of a wave of interest in permutation SGD and similar variants. It is highly topical and a good fit for the conference.

**Theoretical Results**: It is easy focus on FlipFlop and its limited analysis (i.e. quadratics only), but there are many other highly interesting results in this submission. Theorems 1 and 2 show that one-dimensional functions are "easy" for permutation-based optimization, while quadratics in higher dimensions are hard. Yes, this result requires a fixed step-size, but it is also the first analysis of permutation SGD conducted on the level of specific permutations. Theorem 3 refines these results to show that the improvement in Theorem 1 is specific to quadratics. The proof technique for these results is elegant (continuity of the composition of updates + IVT) and, as a I noted in my review, leads to useful intermediate results.

**Experiments** The experiments are limited to synthetic data, but this is a theoretical paper with ~37 pages of appendices already. I think that an intensive evaluation of FlipFlop is beyond the scope of this submission. Moreover, since the analysis considers quadratics only, I think that the nice results on logistic regression in Appendix H are quite encouraging.

# ======

This submission analyzes the effects of permutation choice on the convergence of the random-reshuffling, shuffle-once, and incremental variants of stochastic gradient descent (SGD).
In particular, the goal of this work is to determine settings whether specific permutations can give faster convergence than random reshuffling and in what settings.
The authors show that, in the case of one-dimensional finite-sum functions with smooth Hessians, there exists a sequence of permutations for which SGD converges at a linear (i.e. exponential) rate.
However, they also prove that this phenomenon is specific to one-dimensional functions using a new, dimension-dependent lower bound for any permutation-based SGD method.
This lower bound is tight with the known convergence rate of random reshuffling, implying this method is optimal in the general setting.
Finally, the authors restrict themselves to finite-sums of convex quadratics and develop a simple heuristic permutation "schedule" giving provably faster convergence.
This approach, which is called FlipFlop, alternates between fresh permutations and the previous permutation, but in _reverse_ order.
Small experiments confirm that FlipFlop accelerates convergence of permutation-based SGD variants.


**Summary Of The Review:**

This is a very strong submission that contributes new theoretical insights to a hot topic in optimization for machine learning.
The lower-bound showing that random permutations are optimal in the general setting answers an open question about permutation-based variants of SGD, while the analysis of FlipFlop opens to the door to clever permutation schedules which improve convergence in specific settings.
I think, with additional empirical justification, FlipFlop could become another standard trick in stochastic optimization.
Additionally, the analysis in the appendix develops many small, but novel results which may also have use for other problems in optimization; see "Theory" for an example.
In addition to the novelty of the theoretical results, the paper is polished and the writing is well executed.

Given the above, I strongly advocate for this submission to be accepted.

---

> ### Author Response · Authors · 2021-11-21
> **Author response for Reviewer D2Ry**
>
> We are glad that you liked our paper and that you find it to be a very strong submission. We also thank you for your detailed review, constructive feedback, and also for thoroughly verifying our proofs. We have addressed your concerns below and updated our paper accordingly.
>
> ***
>
> **Theorem 1: How did the authors conclude that permutations used to obtain the exponential rate are optimal?...**
>
> **Response:** Thank you for pointing this out. Indeed the reviewer is correct that we do not provide a lower bound that matches the upper bound. We say that the rate is optimal since full gradient descent on smooth strongly convex function would converge at the rate $e^{-\theta(K)}$, ignoring the dependence on other factors (of course each step of full gradient descent would need to compute all the $n$ gradients and hence each step would be $n$ times slower than a step of SGD), which matches the rate in Theorem 1. In a sense, we treated full gradient descent as the optimal baseline, and hence loosely used the term "optimal". We have updated our paper and removed the word "optimal" from describing the permutations from Theorem 1, and instead only describe them as "exponentially faster than random", which also implies that the actual optimal permutation would be at least exponentially faster than random permutations.
>
> ***
>
> **Theorem 2: Initialization in the Lower Bound....**
>
> **Response:** This is a good point, and below we explain how the lower bound will indeed hold for any random generalization too. We have added the following discussion in the paper, please see Appendix B.4.
>
> The lower bound partitions the step size in three ranges:
>
> * In the step size ranges $\alpha \in [\frac{1}{2(n-1)KL},\frac{3}{nL}]$ and $\alpha \in [\frac{3}{nL}, \frac{1}{L}]$, the initializations are done at the minimizer and it is shown that any permutation-based algorithm will still move away from the minimizer. The choice of initializing at the minimizer was solely for the convenience of analysis and calculations, and the proof should work for any other initialization as well.
>
> 	Furthermore, the effect of initializing at any random point will decay exponentially fast with epochs anyway. To see how, note that every epoch can be treated as $n$ steps of full gradient descent and some noise, and hence the full gradient descent part will essentially keep decreasing the effect of initialization exponentially, and what we would be left with is the noise in each epoch. Thus, it was more convenient for us to just assume initialization at the minimizer and only focus on the noise in each epoch.
> * The step size range $\alpha \notin [\frac{1}{2(n-1)KL}, \frac{1}{L}]$ can be divided into two parts, $\alpha \in [0,\frac{1}{2(n-1)KL}]$ and $\alpha \in [\frac{1}{L},\infty)$.
>
> 	For the range $\alpha \in [0,\frac{1}{2(n-1)KL}]$, we essentially show that the step size is too small to make any meaningful progress towards the minimizer. Hence, instead of initializing at $1$, initializing at any other arbitrary or random (non-zero) point will also give the same slow convergence rate.
>
> 	For the range $\alpha \in [\frac{1}{L},\infty)$, we show that the optimization algorithm will in fact diverge since the step size is too large. Hence, even here, any other arbitrary (non-zero) point of initialize will also give divergence.
>
> ***
>
> **Section 6.1: Proof sketch not explanatory enough ...**
>
> **Response:** Thank you for the constructive feedback, we will try to re-write the entire proof sketch and make it more concise and precise in the next version of the paper.
>
> As for extending the analysis of FlipFlop to general convex functions, the main difficulty arises in controlling the Hessian. For quadratics, the Hessian is constant, and hence the calculations become simple. We think that with a Lipschitz-continuity based assumption on the Hessians, the analysis can be extended to more general families of strongly convex functions, and even PŁ functions (which are non-convex in general), similar to how HaoChen\&Sra(2019) extended their analysis of quadratic functions to more general classes.  A key take-away from FlipFlop is that we had to understand how permutation based SGD works specifically for quadratic functions, that is, we did a white-box analysis. In general, we feel that depending on the specific class of non-convex function (say deep neural networks), practitioners would have to think about permutation based SGD in a white-box fashion, to come up with better heuristics for shuffling.
>
> Due to the 9 page limit, we have currently added this to the appendix (Appendix A) of the paper, but will try to move it to the main paper in the next version, after rewriting the proof sketch and restructuring the paper.
>
> ***
>
> **Experiments**
>
> **Response:** Thank you for this valuable suggestion. We have added the plots with medians and inter-quartile ranges.
>
> ***
>
> **Minor Comments**
>
> **Response:** Thank you for pointing out the typos and the comments, we have fixed them.

---

> > ### Comment · Reviewer_D2Ry · 2021-11-28
> > **Author Response**
> >
> > Thank you for addressing the comments in my review.
> >
> > **Theorem 2: Initialization in the Lower Bound....**: Clearly the second two cases are easy, while the first case represents the interesting problem. I broadly agree with the authors' comments, which should probably be included in the main paper somewhere. Lower bounds for randomized algorithms are somewhat harder to obtain than deterministic ones, so this contribution should be pointed out.
> >
> > **Section 6.1: Proof sketch not explanatory enough ...**: Great. I'm looking forward to the updated version.
> >
> > I'd like to congratulate the authors again on a really nice piece of work.

---

> > > ### Author Response · Authors · 2021-12-06
> > > **Thank you for your review and additional comments**
> > >
> > > Thank you for your review and additional comments, we will incorporate your suggestions into the paper. Please let us know if you have any other concerns.

---

### Author Response · Authors · 2021-11-21
**Author Response for the Reviewers**

We thank all the reviewers for their time and effort in reviewing the paper, as well as for providing constructive feedback. We are encouraged by the overall positive scores (10 6 6 6). The reviewers agree that the paper provides novel theoretical contributions (D2Ry, jCFt) to the important topic of SGD (D2Ry, JbYX), that it has novel proof techniques (jCFt) and secondary results which might be of independent interest (D2Ry, jCFt), and that it motivates future research into clever permutations for faster convergence (D2Ry, 53pE). The reviewers also agree that the paper is well written (D2Ry, JbYX, 53pE).

We would like to say that being the first paper (to the best of our knowledge) to theoretically analyze the optimality of random permutations, we limited our scope to a specific, but common theoretical setting - strongly convex functions with constant step size. We think that future work can generalize the results of this paper to settings of non-convexity, variable step sizes, as well as techniques like variance reduction and momentum.

We have addressed the concerns of each reviewer below. The reviewers had some concerns in common, but we have responded to all of them individually. We have updated our paper according to the feedback by the reviewers, and will continue refining it and updating it based on any further discussion with the reviewers.

---

### Decision · Program_Chairs · 2022-01-20

**Decision:**

Accept (Poster)

**Comment:**

The paper considers the effect of permutations in SGD - exploring the question of can we go beyond random permutations (which themselves have shown to be better than with replacement sampling)? The paper studies these questions from multiple viewpoints - showing that there is a one dimensional function for which the optimal permutation can be exponentially better in terms of rate than random. Further they show that for the general high dimensional the gap between random and optimal is non-existent. Further they study a Flip-Flop algorithm which flips the permutation every alternate epoch and for convex quadratics they show that this technique can lead to improved convergence rates for multiple base permutation schemes.

Overall the results of the paper was found to be interesting across the reviewers. The reviewers agree that the paper is well written. The paper initiates the analysis in a new direction for optimization, i.e. how can we leverage permutations to further improve the convergence of GD and how much further can we go beyond random permutations.

The only weakness highlighted by the reviewers is the limitation of scope to quadratic functions for the FlipFlop algorithm - while this is a significant restriction, given the new line of enquiry opened by the paper this can be discounted.